# Positional Attention: Expressivity and Learnability of Algorithmic Computation

**Artur Back de Luca** [* 1]  **George Giapitzakis** [* 1]  **Shenghao Yang** [* 1]  **Petar Veličković** [2]  **Kimon Fountoulakis** [1]

## Abstract

There is a growing interest in the ability of neural networks to execute algorithmic tasks (e.g., arithmetic, summary statistics, and sorting). The goal of this work is to better understand the role of attention in Transformers for algorithmic execution. Its importance for algorithmic execution has been studied theoretically and empirically using parallel computational models. Notably, many parallel algorithms communicate between processors solely using positional information. Inspired by this observation, we investigate how Transformers can execute algorithms using positional attention, where attention weights depend exclusively on positional encodings. We prove that Transformers with positional attention (positional Transformers) maintain the same expressivity of parallel computational models, incurring a logarithmic depth cost relative to the input length. We analyze their in-distribution learnability and explore how parameter norms in positional attention affect sample complexity. Our results show that positional Transformers introduce a learning trade-off: while they exhibit better theoretical dependence on parameter norms, certain tasks may require more layers, which can, in turn, increase sample complexity. Finally, we empirically explore the out-of-distribution performance of positional Transformers and find that they perform well in tasks where their underlying algorithmic solution relies on positional information.

## 1. Introduction

Neural network models, particularly Transformers (Vaswani et al., 2017), have achieved remarkable success across various domains, especially in their ability to learn and execute algorithms (Veličković & Blundell, 2021). Notable applications range from basic arithmetic to solving problems on abstract data structures such as graphs and lists (Lee et al., 2024; Tang et al., 2020; Tay et al., 2020; Yan et al., 2020; Veličković et al., 2020; Veličković et al., 2022; Dudzik & Veličković, 2022; Ibarz et al., 2022; Cappart et al., 2023; Diao & Loynd, 2023; Bevilacqua et al., 2023; Rodionov & Prokhorenkova, 2023; Minder et al., 2023; Georgiev et al., 2023; Engelmayer et al., 2023a; Georgiev et al., 2024; Li et al., 2024). Due to this success, there is a growing interest in understanding neural networks as computational models that can solve complex algorithmic and computational tasks.

In this context, Pérez et al. (2021); Wei et al. (2022) show that Transformers are Turing Complete, and Giannou et al. (2023); Back De Luca & Fountoulakis (2024); Yang et al. (2024) illustrate how Transformers can effectively encode instructions in their parameters to solve linear algebra and graphs problems. Additionally, it has been shown that Transformers can perform computational tasks using far fewer layers than the number of sequential steps (Liu et al., 2023), indicating a connection between Transformers and parallel algorithms. Building on this, Sanford et al. (2024) demonstrates that Transformers can simulate the Massively Parallel Computation (MPC) model (Andoni et al., 2018), which is based on the MapReduce framework for large-scale data processing (Dean & Ghemawat, 2008).

While parallel computational models like MPC do not restrict how communication is established, we observe that in many parallel algorithms, the communication between processors is independent of the processed data. Instead, this communication relies solely on processor identification (see Appendix A.3 for examples). Motivated by this observation, we study a Transformer variant that reflects this data-agnostic communication property. Specifically, we investigate *positional attention*, where the attention weights are determined exclusively by fixed positional encodings.[1]

**Our contributions.** We examine Transformers with posi-

---
[*]Equal contribution [1]David R. Cheriton School of Computer Science, University of Waterloo, Waterloo, Ontario, Canada. [2]Google DeepMind, London, UK.. Correspondence to: Artur Back de Luca <abackdel@uwaterloo.ca>.

*Proceedings of the 42nd International Conference on Machine Learning*, Vancouver, Canada. PMLR 267, 2025. Copyright 2025 by the author(s).

---
[1]While positional encodings remain constant across layers, their attention weights can vary – unlike Graph Neural Networks (GNNs), which take a fixed connectivity structure as input. Furthermore, the tasks in this paper lack a predefined graph structure, necessitating an artificial graph for application to GNNs. For experimental comparisons, see Appendix D.

tional attention (positional Transformers) from both a theoretical and empirical perspective in the context of algorithmic computation. Our contributions are given below:

*Expressivity.* We prove that positional Transformers can simulate a single round of the Massively Parallel Computation (MPC) model with $O(\log n)$ layers, where $n$ is the input length. The expressivity result emerges from a connection between Transformers and parallel computational models. In particular, we utilize a proxy computational model, which only allows communication over a static network, bridging the simulation results between positional Transformers and MPC and, consequently, by Sanford et al. (2024), standard Transformers. Due to the restrictive nature of communication in positional Transformers and static networks, this equivalence with MPC is established with logarithmic cost.

*In-distribution generalization.* We provide a norm-based generalization bound for positional Transformers, which, combined with our expressivity result, implies that they can learn algorithmic tasks to any desired accuracy via empirical risk minimization. Our analysis reveals a trade-off between the number of layers and parameter norms: positional Transformers exhibit better dependence on parameter norms but may require more layers for certain tasks, potentially leading to higher overall theoretical complexity. This result is established using the Provably Approximately Correct (PAC) framework, where the complexity bounds of the hypothesis class are expressed as a function of the network parameter norms (Edelman et al., 2022).

*Out-of-distribution (OOD) generalization.* We empirically analyze the OOD performance of positional and standard Transformers. We observe that positional Transformers exhibit better OOD generalization for tasks where the underlying algorithmic solution relies solely on positional information. However, for tasks that do not meet this criterion, such as induction heads tasks, both models perform poorly. We *hypothesize* that the inherently dynamic nature of communication in such tasks affects the loss landscape, making it challenging to discover good parameters for the positional Transformer during training.

## 2. Related work

**Empirical:** Several studies focus on the experimental aspects of training neural networks to execute algorithms. Notable examples include algorithms operating on common data structures, such as lists (e.g., sorting, binary search) and graphs (e.g., shortest paths, connected components) (Yan et al., 2020; Tang et al., 2020; Veličković et al., 2022; Ibarz et al., 2022; Bevilacqua et al., 2023; Engelmayer et al., 2023b; Georgiev et al., 2023; Minder et al., 2023; Diao & Loynd, 2023). Additional research on combinatorial optimization tasks using graph neural networks and Trans-

formers offers further insights into algorithmic learning in structured domains (Vinyals et al., 2015; Prates et al., 2019; Joshi et al., 2019; Bai et al., 2020; Selsam & Bjørner, 2019; Karalias & Loukas, 2020; Gasse et al., 2019; Cappart et al., 2023; Veličković, 2023). Recent studies have also explored the broader algorithmic capabilities of Transformer-based models, shedding light on their potential for general algorithmic reasoning (Delétang et al., 2023; Butoi et al., 2025; Barbero et al., 2025). A parallel stream of research has focused on performing arithmetic operations (e.g. addition, multiplication) (Kaiser & Sutskever, 2016; Klindt, 2023; Shen et al., 2023; Lee et al., 2024; Ruoss et al., 2023). Specifically, Rodionov & Prokhorenkova (2023) leverages problem-specific information within a self-supervised training framework, whereas the other studies use intermediate labels to guide the learning process. For example, Engelmayer et al. (2023a) shows that using intermediate labels derived from parallel algorithms leads to improved performance. While some research, such as Kazemnejad et al. (2023), investigates the impact of different positional encodings across several tasks, to the best of our knowledge, no existing work examines the use of positional attention within the context of neural algorithmic reasoning. The closest formulation to the positional Transformer architecture presented in Section 4 is the position-based attention of Schmidt & Di Gangi (2023), but it is used in the context of neural machine translation.

**Theoretical:** From a theoretical perspective, the most closely related work to ours is Sanford et al. (2024), which presents simulation results for standard Transformers and the Massively Parallel Computation (MPC) model. Our work also presents simulation results demonstrating that positional Transformers can simulate MPC. The simulation proof involves introducing a proxy computational model with an alternative communication paradigm that can be shown to simulate MPC. Subsequently, we establish that the positional Transformer architecture can simulate this proxy model and, by extension, MPC itself. Other relevant studies demonstrate the expressive power of neural networks through simulation results. For example, Siegelmann & Sontag (1995) establishes the Turing completeness of recurrent neural networks (RNNs), while Hertrich & Skutella (2023) presents specific RNN constructions that solve the shortest paths problem and provide approximate solutions to the knapsack problem. Additionally, other simulation results focused on Transformers have shown their Turing completeness (Pérez et al., 2021; Wei et al., 2022) as well as demonstrated constructive solutions to linear algebra and graph-related problems (Giannou et al., 2023; Back De Luca & Fountoulakis, 2024; Yang et al., 2024).

**Self-attention**

$$X^{(0)} = X \oplus P$$

**Positional attention**

$$X^{(0)} = X$$

Figure 1: Diagram comparing the operations of self-attention in Transformers with positional attention. The figure illustrates a single attention head, but in multi-head attention, multiple sets of queries, keys, and values are processed in parallel and then combined. In Transformers, the model's input, $X^{(0)}$, is a combination of input values $X$ and positional encodings $P$. In positional attention, however, these components are processed separately. At layer $\ell$, the query ($Q^{(\ell)}$) and key ($K^{(\ell)}$) are derived solely from the positional encodings $P$, where $P$ remains fixed across layers. These are multiplied (denoted by Mul) and passed through a softmax function to produce the attention matrix $A^{(\ell)}$. As in self-attention, the value $V^{(\ell)}$ in positional attention is computed from the previous layer's input, $X^{(\ell-1)}$. The attention matrix $A^{(\ell)}$ and the value $V^{(\ell)}$ are then multiplied to form the weighted representation, which is linearly transformed into the output $O^{(\ell)}$. This output is passed to a Multilayer Perceptron (MLP) for further processing, as detailed in Section 4.

## 3. Preliminaries and notation

Let $\mathbb{N} = \{1, 2, \dots\}$ denote natural numbers and $[n] = \{1, 2, \dots, n\}$ for $n \in \mathbb{N}$. For a set $S$, let $\mathcal{P}(S)$ be its power set (i.e. the set containing all subsets of $S$). For a matrix $Z$, let $Z_{i,:}$ and $Z_{:,i}$ be the $i$-th row and column of $Z$, respectively. $\|\cdot\|_2$ denotes the spectral norm for matrices, and $\|\cdot\|_{p,q}$ denotes the $(p,q)$ matrix norm where the $p$-norm is over columns and $q$-norm over rows (e.g. $\|Z\|_{2,1} = \sum_i \|Z_{:,i}\|_2$). For vectors, $\|\cdot\|_p$ denotes the $\ell_p$ norm. For a sequence of vectors $X \in \mathbb{R}^{n \times d}$ we may use $X[i] = X_{i,:} \in \mathbb{R}^d$ for the $i$-th vector in the sequence.

**Definition 3.1** (Risk). Let $\mathcal{D}$ be a distribution over $\mathcal{X} \times \mathbb{R}$, and let $\ell : \mathbb{R} \times \mathbb{R} \to \mathbb{R}$ be a loss function. For a given class of functions $\mathcal{F} = \{f : \mathcal{X} \to \mathbb{R}\}$ and $f \in \mathcal{F}$, the *risk* (i.e. the generalization error) is defined as $\text{risk}(f; \mathcal{D}) = \mathbb{E}_{(x,y) \sim \mathcal{D}} [\ell(f(x), y)]$, and the *empirical risk* is $\widehat{\text{risk}}(f; (x^{(i)}, y^{(i)})_{i=1}^m) = \frac{1}{m} \sum_{i=1}^m \ell(f(x^{(i)}), y^{(i)})$.

## 4. The Positional Transformer architecture

We now define the *positional Transformer*, an adaptation of the Transformer model (Vaswani et al., 2017). This adaptation is motivated by the fact that communication between machines is independent of the data processed in many real-world parallel algorithms. In particular, we decouple the input values from the computation of attention weights. For an input $X \in \mathbb{R}^{n \times d_X}$, we define the $\ell^{\text{th}}$ layer as follows:

$$\text{F}^{(\ell)}(X) = \Phi^{(\ell)} \left( \left( \bigoplus_{h=1}^{H} A^{(\ell,h)} X W_V^{(\ell,h)} \right) W_O^{(\ell)} \oplus X \right). \tag{1}$$

The input is processed by $H$ attention heads, each associ-

ated with an attention weight matrix $A^{(\ell,h)} \in (0,1)^{n \times n}$ and a value matrix $W_V^{(\ell,h)} \in \mathbb{R}^{d_X \times d_V}$. Here, $\ell$ denotes the layer index and $h$ the head index, allowing a specific attention head within a layer to be identified as $(\ell, h)$. The outputs of these attention heads are concatenated and then transformed by an output matrix $W_O^\ell \in \mathbb{R}^{H \cdot d_V \times d_O}$. This result is concatenated with a residual connection of the input $X$ and then passed through a multilayer perceptron (MLP), represented as $\Phi^{(\ell)} : \mathbb{R}^{d_O + d_X} \to \mathbb{R}^{d_{\text{out}}}$.

Traditionally, in standard Transformers, the attention matrix $A^{(\ell,h)}$ is derived using self-attention.

$$A^{(\ell,h)}(X) = \text{softmax} \left( \left( X W_Q^{(\ell,h)} \right) \cdot \left( X W_K^{(\ell,h)} \right)^\top \right). \tag{2}$$

Here, the attention weights are derived by the input matrix $X$, using query and key matrices $W_Q^{(\ell,h)}, W_K^{(\ell,h)} \in \mathbb{R}^{d_X \times d_m}$, where $d_m$ is the embedding dimension.

In contrast, we utilize *positional attention*, where attention weights are learned solely using positional encodings $P$, which are constant across all layers. This distinction is also illustrated in Figure 1.

$$A^{(\ell,h)} = \text{softmax} \left( \left( P W_Q^{(\ell,h)} \right) \cdot \left( P W_K^{(\ell,h)} \right)^\top \right). \tag{3}$$

We utilize node positional encodings defined by a matrix $P \in \mathbb{R}^{n \times d_P}$ and whose attention weights are computed similarly to traditional attention, using query and key matrices but with input dimension $d_P$. The positional encodings are fixed across layers, indicated by the absence of the $\ell$ index.

Theoretically, we evaluate whether these changes reduce the expressive power of positional Transformers. We prove that

positional Transformers, like standard Transformers, can simulate the Massively Parallel Computation (MPC) model. This indicates that there is no loss in expressivity, though with an added logarithmic increase in depth relative to the input length, as discussed further in the next section.

## 5. Expressivity of Positional Transformers

A popular approach to illustrate the expressivity of an architecture is by establishing an equivalence with a computational model (Sanford et al., 2024; Pérez et al., 2021; Loukas, 2020). For positional Transformers, we will utilize the Massively Parallel Computation (MPC) model, which was first used by Sanford et al. (2024) to analyze standard Transformers. We begin by stating our main result.

**Theorem 5.1.** *Consider an instance of MPC* T *with* $R$ *rounds,* $N$ *machines with local memory* $s$. *Let* $\mathcal{M}$ *be a model following the architecture in Equation* (1) *with* $n = N+1$ *nodes,* $2R\lceil \log N \rceil$ *layers and* $2s$ *attention heads. Then, for any instance* T *with Borel measurable local functions, there exists a configuration of* $\mathcal{M}$ *that approximates* T *to any desired degree of accuracy.*

*Proof overview:* We establish this result using a two-step procedure. This procedure involves introducing a proxy computational model which, in contrast to MPC, uses a static network for communication. We call this model Parallel Computation with Oracle Communication (PCOC)[2], and its formal definition is presented in Appendix A. In the first step of the proof, we demonstrate that the computations over a static network model such as PCOC can effectively simulate the dynamic communication of MPC. Specifically, we show that $R$ rounds of an MPC protocol can be simulated using $2R\lceil \log n \rceil$ rounds of PCOC, where $n$ represents the number of processors in MPC. The additional computation rounds and communication coordinated by Beneš networks (Beneš, 1965) are sufficient to simulate MPC.

In the second step of the proof, we show that PCOC can be, in turn, simulated by positional Transformers. Specifically, the attention mechanism is shown to simulate communication between nodes, while the computation stage leverages universal approximation results for MLPs (Cybenko, 1989; Hornik et al., 1989a). By demonstrating our architecture's capability to approximate both communication and local computation, we establish that it can simulate any MPC instance of $R$ rounds using $2R \log n$ layers.

By Theorem 3.4 in Sanford et al. (2024), which demonstrates the simulation of standard Transformers by MPC, we can derive the following remark regarding the simulation of standard Transformers by positional Transformers.

*Remark* 5.2. Consider a standard Transformer $\mathcal{M}'$ with $N$ nodes, $L$ layers, and $d_V \cdot H$ sublinear in $N$. Then, there exists a positional Transformer $\mathcal{M}$ with $O(N^2)$ nodes, $O(L \log N)$ layers, and $d_V \cdot H$ sublinear in $N$ that can simulate the computation of $\mathcal{M}'$.

Note that such a simulation result introduces an additional quadratic dependency on the number of nodes in positional Transformers. These dependencies are merely a byproduct of the simulation strategy employed in Sanford et al. (2024), and other, more efficient, simulation strategies may exist. Additionally, this quadratic cost arises only when simulating the exact same sequence of operations of the standard Transformer. In practice, the model may converge to parameter configurations that do not correspond to the standard Transformer. When solving a given problem, positional Transformers may adopt a completely different configuration to achieve the solution without incurring this additional cost, as demonstrated by our experiments in Section 7.

## 6. Learnability of Positional Transformers

In this section, we present a norm-based generalization bound for the positional Transformer architecture, discuss its implications, and compare it with the previously derived bound for the Transformer architecture that employs self-attention (Edelman et al., 2022). In particular, by decoupling the attention weights from the input $X$ and requiring them to depend exclusively on the positional encodings $P$, we remove the exponential dependence (w.r.t. the depth $L$) on the norms of key and query weight matrices.

We consider multi-layer networks obtained by composing individual layers from Equation (1), denoted as

$$F^{1:L}(X) = F^{(L)} \circ \cdots \circ F^{(2)} \circ F^{(1)}(X). \quad (4)$$

The resulting network $F^{1:L}$ has $L$ layers and each layer has $H$ attention heads. For simplicity we assume that the MLP $\Phi^{(\ell)}$ consists of two layers:

$$\Phi^{(\ell)}(Z) = \sigma(ZW_1^{(\ell)})W_2^{(\ell)}, \ \forall \ell \in [L], \quad (5)$$

where $\sigma$ is an $L_\sigma$-Lipschitz activation with $\sigma(0) = 0$. Using MLPs with more layers would lead to a worse dependence on the spectral norm of weight matrices, but does not affect the results in this section.[3] Following prior work, our generalization bound is derived for scalar-valued functions. We extract a scalar output from our architecture as $w^\top X^{\mathsf{out}}[n]$ with trainable weights $w \in \mathbb{R}^{d_{\mathsf{out}}}$, where $X^{\mathsf{out}} = F^{1:L}(X) \in \mathbb{R}^{n \times d_{\mathsf{out}}}$. The index $i$ at which we apply the linear function $x \mapsto w^\top x$ can be arbitrary, and

---

[2]We note that PCOC is only used to establish our expressivity result. It is not meant as a model to compete with established parallel models, such as MPC.

[3]More layers in the MLP would simply increase the constant $c$ in Theorem 6.1 to $c' \geq c$. On the other hand, two-layer MLPs are already sufficient to invoke the universal approximation properties required by the expressivity result of Theorem 5.1.

here we simply set to the last index. This setting captures algorithmic computations whose output is a scalar. For comparison purposes, it aligns with the setup considered previously by Edelman et al. (2022) for the analysis of self-attention Transformers. In general, for computational tasks that require vector or matrix output, one can easily extend our result to parallel architectures using a union bound on the failure probability $\delta$ and incur an additional logarithmic cost with respect to the dimension of the output.

Given $B_{2,1}, B_2 \geq 1$, the class of $L$-layer $H$-head bounded-norm scalar-output positional Transformer networks is:

$$\mathcal{F} = \left\{ w^\top \left( F^{1:L}(X)[n] \right) : \|w\|_2 \leq B_2, \text{ and } \forall \ell, h : \right.$$

$$\|W_K^{(\ell,h)\top} W_Q^{(\ell,h)}\|_{2,1}, \|W_V^{(\ell,h)}\|_{2,1}, \|W_1^{(\ell)}\|_{2,1} \leq B_{2,1},$$

$$\left. \|W_2^{(\ell)}\|_{2,1} \leq B_{2,1}, \|W_V^{(\ell,h)}\|_2, \|W_1^{(\ell)}\|_2, \|W_2^{(\ell)}\|_2 \leq B_2 \right\},$$

with $F^{1:L}$ from Equation (4), $A^{(\ell,h)}$ from Equation (3), and $\Phi^{(\ell)}$ from Equation (5). In order to avoid overloading the result with unnecessary notation that do not add value to the discussion in any meaningful way, we assume that the input $X \in \mathbb{R}^{n \times d_X}$ and positional encodings $P \in \mathbb{R}^{n \times d_P}$ are normalized row-wise so that $\|X_{i,:}\|_2 = \|P_{i,:}\|_2 = 1$ for all $i \in [n]$. We denote $d = \max\{d_X, d_P, d_V, d_{\text{out}}\}$, and therefore $d(H+1)$ is the largest possible size of any weight matrix along any axis.

Let $\mathcal{D}$ be a distribution on $\mathbb{R}^{n \times d_X} \times \mathbb{R}$ and $(X^{(1)}, y^{(1)}), \dots, (X^{(m)}, y^{(m)}) \in \mathbb{R}^{n \times d_X} \times \mathbb{R}$ be a sequence of $m$ samples.

**Theorem 6.1** (Generalization bound). *For any $\delta > 0$, with probability at least $1 - \delta$, simultaneously for all $f \in \mathcal{F}$, it holds that for some $c > 0$,*

$$\left| \mathsf{risk}(f; \mathcal{D}) - \widehat{\mathsf{risk}}(f; (X^{(i)}, y^{(i)})_{i=1}^m) \right|$$

$$\leq \tilde{O}\left( (HL_\sigma B_2)^{cL} B_{2,1} \sqrt{\frac{\log(Hdmn)}{m}} + \sqrt{\frac{\log(1/\delta)}{m}} \right).$$

In the context of learning algorithmic computations, Theorem 6.1 means that the generalization gap $|\mathsf{risk}(f) - \widehat{\mathsf{risk}}(f)|$ goes to 0 as the number of samples tends to infinity. On the other hand, by Theorem 5.1, for any computational task that is computable in an MPC model with Borel measurable local functions, there is $f \in \mathcal{F}$ (where the parameters $L, H, B_2, B_{2,1}$ that specify $\mathcal{F}$ might depend on $n$ but are independent of $m$) such that the empirical risk can be arbitrarily close to 0 for all samples of size $m$ and for all $m \geq 0$, i.e. $\widehat{\mathsf{risk}}(f; (X^{(i)}, y^{(i)})_{i=1}^m) \approx 0, \forall m$. Consequently, in this case, Theorem 6.1 implies that as the number of samples $m$ tends to infinity, we have $\mathsf{risk}(f; \mathcal{D}) \to 0$. This means that positional Transformer learns algorithmic computations through empirical risk minimization.

Compared with the norm-based risk bounds of self-attention Transformer networks in Edelman et al. (2022), the bound in Theorem 6.1 does not depend on the spectral norm of key and query weight matrices. Intuitively, this is due to restricting attention weights to be determined exclusively by positional encodings, as opposed to allowing them to be determined compositionally by the output from the previous layer. If we replace positional attention with self-attention in the definition of $\mathcal{F}$, then the bound in Theorem 6.1 would incur an additional factor $B_{QK}^{O(L)}$ where $\|W_K^{(\ell,h)\top} W_Q^{(\ell,h)}\|_2 \leq B_{QK}, \forall \ell, h$. This shows a potential advantage of positional attention when all other model configurations are the same.

Note that achieving a low empirical risk with self-attention or positional attention can require very different model configurations, which may depend on the computational task. For example, in the simple task of computing the minimum of an $n$-number array, a standard Transformer can solve it approximately using just one attention layer with an error controlled by how well softmax approximates hardmax. In contrast, a positional Transformer can solve it exactly but requires $\log n$ layers with two attention heads per layer (e.g., via binary tree reduction). Here, the positional Transformer's worst-case sample complexity depends on $n$, while the standard Transformer does not. However, the standard Transformer can have a much worse dependence on the magnitude of the input numbers due to the softmax-hardmax approximation. Therefore, one architecture might be better depending on the magnitude of $n$ and the input numbers. For more complex tasks, setting an "optimal" model configuration (e.g., choosing $L, H, d_V$) is difficult.

## 7. Experiments

In this section, we evaluate the performance of positional and standard Transformers across various algorithmic tasks and regimes. We first outline the tasks considered in this work and then describe the experimental setup in detail. Next, we present results for the in-distribution regime across different settings, followed by an analysis of the out-of-distribution performance. Finally, we introduce a more complex task that combines textual and numerical data. For additional analyses, including comparisons with other models and ablation studies, refer to Appendix D.

**Tasks:** We consider the following algorithmic tasks:

1. *Cumulative sum*: Given $x \in \mathbb{R}^n$, output $y \in \mathbb{R}^n$ where each element $y_i$ is the sum of the first $i$ elements of $x$, i.e. $y_i = \sum_{j=1}^i x_j$.

2. *Cumulative min*: Given $x \in \mathbb{R}^n$, output $y \in \mathbb{R}^n$ where each element $y_i$ is the minimum value among the first $i$ elements of $x$, i.e. $y_i = \min\{x_j \mid 1 \leq j \leq i\}$.

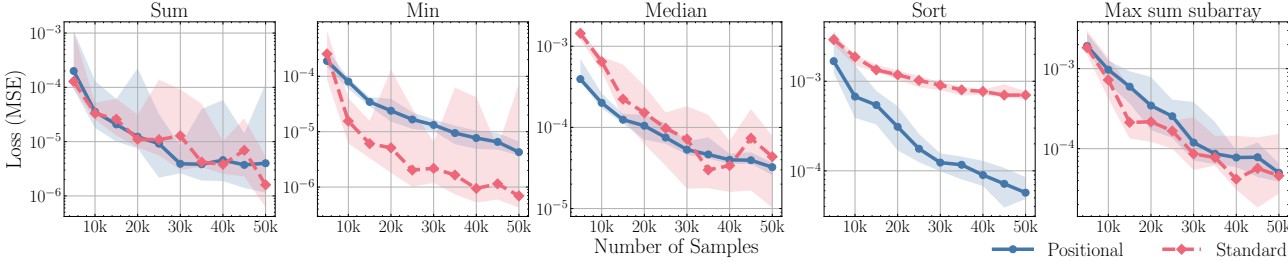

Figure 2: In-distribution loss across all five tasks for standard Transformers (red) and positional Transformers (blue) as a function of the number of training samples (indicated on the x-axis).

3. *Cumulative median*: Given $x \in \mathbb{R}^n$, output $y \in \mathbb{R}^n$ where each element $y_i$ is the median of the first $i$ elements of $x$, i.e. $y_i = \text{median}\{x_j \mid 1 \leq j \leq i\}$.

4. *Sorting*: Given $x \in \mathbb{R}^n$, output $\text{sort}(x)$, a vector containing the entries of $x$ sorted in ascending order.

5. *Cumulative maximum sum subarray*: given $x \in \mathbb{R}^n$, output $y \in \mathbb{R}^n$ where each element $y_i$ is the sum of elements of a maximum sum subarray of the first $i$ elements of $x$, i.e. $y_i = \max_{1 \leq j \leq k \leq i} \left( \sum_{l=j}^{k} x_l \right)$.

The tasks selected were chosen to represent varying levels of complexity and illustrate the strengths and limitations of each architecture. We adopt cumulative versions of algorithms when feasible for several reasons: They naturally provide $n$ to $n$ training settings and are more challenging than non-cumulative versions.

To test the non-numeric capabilities of positional Transformers, we further present two additional tasks that utilize textual data. The first is the $k$-hop inductive heads of Sanford et al. (2024), which represents a higher-order version of the standard inductive heads task by recursively executing the completion of a bigram to determine the subsequent bigram. The second task employs the same computational tasks described earlier in the list, but incorporates additional textual data that serve as categories, which the model must learn to use for conditional reasoning.

Although both tasks involve pattern matching, the nature of pattern matching differs. The $k$-hop induction heads task requires dynamic pattern matching, where each step depends on previous matches, whereas the mixed-type input task involves static pattern matching, which better aligns with our architecture. Despite our expressivity result guaranteeing that this task can be solved using a static network, we *hypothesize* that the inherently dynamic nature of communication in this task is reflected in the loss landscape. Consequently, discovering good parameters for the positional Transformer is difficult. This is evident in our experiments, where we run multiple trials, all of which result in poor out-of-distribution performance.

**Experimental setting:** All tasks employ the same model structure of Equation (1), augmented with linear encoding and decoding layers. We compare the standard Transformer, which utilizes the attention mechanism in Equation (2), and the positional Transformer, which employs the attention defined in Equation (3). Standard Transformers incorporate positional encodings concatenated with the input values, unlike positional Transformers, where positional information is exclusively supplied through the matrix $P$. In Appendix D, we examine other configurations for standard Transformers, including relative positional encodings such as Rotary Positional Embedding (RoPE) (Su et al., 2024b) and find no significant differences in performance. Both variants share the same number of layers and dimensional configurations, with any specific differences explicitly noted. For details on the particular layer configurations and training details, refer to Appendix D. The training data is sampled in the range $[-2, 2]$. To ensure diversity in the data, for each input sample, we first select lower and upper bounds $\gamma_l$ and $\gamma_u$ uniformly in $[-2, 2]$, and then for each of the $n$ elements of the input sample, we select its value uniformly in $[\gamma_l, \gamma_u]$.

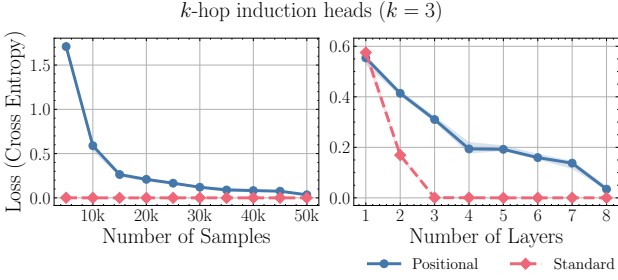

Figure 3: In-distribution loss for standard Transformers (red) and positional Transformers (blue) in the $k$-hop induction heads task of Sanford et al. (2024), plotted as a function of the number of training samples (left, with eight layers) and the number of layers (right, with 50,000 training samples).

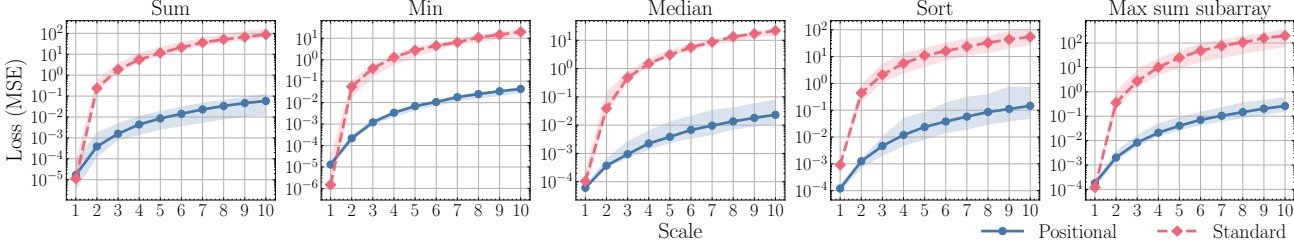

Figure 4: In-distribution and out-of-distribution loss for standard Transformers (red) and positional Transformers (blue) across all five tasks. The x-axis shows the OOD scale factor, with a scale factor of one indicating in-distribution performance.

## 7.1. In-distribution regime

We evaluate our architecture in a fixed input length regime, with a detailed analysis of in-distribution generalization as a function of the number of samples and expressive power. The plots presented show the median over ten runs, with shaded areas representing the $10^{th}$ and $90^{th}$ percentile.

**Sample Size vs. Generalization:** In this setting, we demonstrate the learnability of positional Transformers, established in Theorem 6.1, as a function of the number of samples. To this end, we fix the input length $n = 8$ and examine in-distribution loss as a function of the number of training samples, ranging from 5,000 to 50,000. Figure 2 shows that for all tasks, the in-distribution loss of both architectures consistently decreases with increasing number of samples.

**Expressive power vs. Generalization:** In this experiment, we empirically validate the theoretical results of Remark 5.2, analyzing the in-distribution capabilities of standard and positional Transformers as a function of their expressive power, measured by the number of layers. We evaluate their performance on a generalized version of the induction heads task (Olsson et al., 2022), which reflects the relational capabilities of self-attention. This task, named *k-hop induction heads*, applies a successive $k$ number of bigram completions to determine the next bigram in the sequence. We adapt the setting of Sanford et al. (2024), restricting the number of hops to at most three and the sequence length to 30. For a complete description of the task and additional experimental details, we refer readers to Sanford et al. (2024) and Appendix D, respectively. We chose this task because it is designed to highlight the efficiency of self-attention in terms of the number of required layers. The in-distribution results in Figure 3 show that standard Transformers exhibit near-zero in-distribution loss with as few as three layers. In contrast, we find that positional Transformers generally perform worse. As established in Section 7, this results from the expressivity of the positional Transformers since achieving a level of performance comparable to that of the standard Transformers requires an increasing number of layers, aligning with our theoretical results.

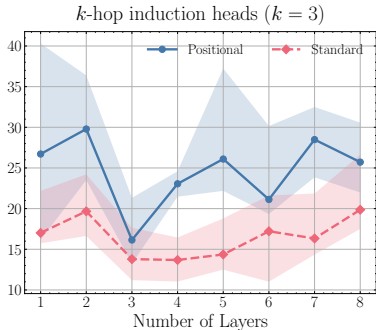

Figure 5: Out-of-distribution loss for standard Transformers (red) and positional Transformers (blue) in the $k$-hop induction heads task of Sanford et al. (2024), plotted as a function of the number of layers (with 50,000 training samples).

## 7.2. Out-of-distribution regime

We further evaluate the capabilities of both architectures in generalizing out-of-distribution (OOD) for the tasks previously outlined. To generate OOD data, we employ a similar sampling strategy as in training but extend the value range to $[-2c, 2c]$, where $c > 1$ is the OOD scale factor. Additionally, during the test sampling process, we apply a rejection step to ensure that either $\gamma_l < -2$ or $\gamma_u > 2$, while maintaining $-2c \leq \gamma_l \leq \gamma_u \leq 2c$. This ensures that a test sample will not lie in the training domain with high probability.[4] This implies that our test results reflect the "true" out-of-distribution performance of both architectures.

The results in Figure 4 show that positional Transformers achieve significantly more stable performance across increasing out-of-distribution ranges than standard Transformers across computational tasks. We *hypothesize* that such differences in performance can be attributed to algorithmic alignment (Xu et al., 2020), a concept that suggests

---

[4]This sampling strategy does not guarantee that every test sample falls outside the training distribution. However, as shown in Appendix E, the probability of a test sample being within the training distribution is at most $O(1/nc^2)$. In practice, most test instances lie outside this domain. For further details, see Appendix E.

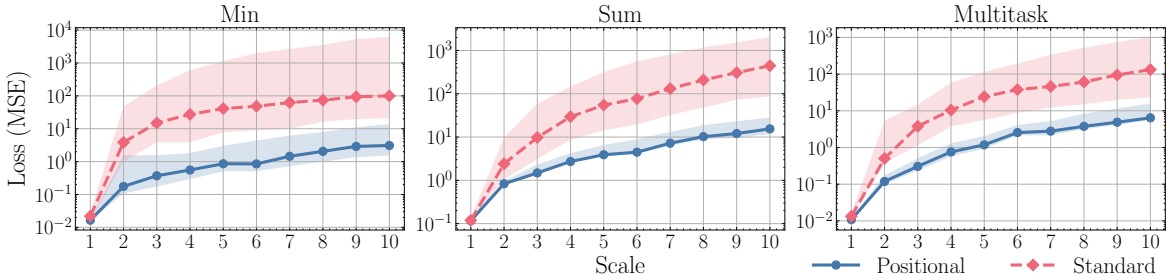

Figure 6: In-distribution and out-of-distribution losses for standard (red) and positional Transformers (blue) across mixed-type input task variations. The x-axis shows the OOD scale factor, where one indicates in-distribution performance.

a particular architecture demonstrates better generalization capabilities if its underlying computational structure aligns more closely with the target task. Specifically, the structure of these tasks aligns well with the core motivation for positional attention, i.e., enabling communication through positional information alone. For example, the minimum task is known to have a solution where the underlying computational graph is a binary tree and independent of the input values, see Appendix A.3. In Appendix F, we discuss potential reasons why standard Transformers fail to match the performance of positional Transformers and present additional experiments that illustrate the differing behaviors of self-attention and positional attention.

For the $k$-hop induction heads task, OOD data is generated by creating sequences sampled exclusively from a portion of the alphabet not used during training. Figure 5 illustrates the OOD performance of both standard and positional Transformers. Note that the positional Transformer performs poorly. We hypothesize that this is because the $k$-hop induction heads task deviates from our core motivation, as its inherent task requirement suggests a need for dynamic communication that changes with every input, see also our comments in the description of the tasks above.

### 7.3. Additional experiments with mixed-type inputs

In this section, we further investigate the OOD performance when the input data is a mixture of tokens and numbers. This task involves conditional reasoning and simple pattern matching on top of computing an aggregation function. We begin by illustrating a training and test sample, followed by high-level overview of the task. For the definition of the task and the details about the setting, see Appendix D.3.

Training sample: `Input = ['Cat2', 3.45, 'Cat+7', 'Cat5', 1.23, 'Cat7', 0.65, 'Cat8', 2.23, 'Cat11', 'Cat-8', 4.10, 'Cat13', 1.10, 'Cat14', 0.10, 'Cat20', 2.75, 'Find min of Cat5, Cat8, Cat11 and Cat20'], Output = 1.23`

Test sample: `Input = ['Cat23', 7.28, 'Cat24',`

`33.5, 'Cat28', 9.17, 'Cat30', 55.90, 'Cat31', 'Cat*24', 23.70, 'Cat33', 12.47, 'Cat34', 8.45, 'Cat_40', 'Cat40', 1.50, 'Find min of Cat28, Cat31, Cat33 and Cat40'], Output = 1.50`

The input prompt presents the model with a number of textual categories and associated values and asks for the result of an aggregation operation on a subset of the categories. Noise is injected by adding irrelevant categories (with no associated value) in the prompt (colored blue in the examples). Note that the category identifiers, the range of values, and the type of noise all change between training and test data. Specifically, the train category identifiers are integers between 1 and 20, whereas the test category identifiers are integers between 21 and 40. The models are trained on category values in $[0, 5]$ and tested on category values in $[0, 5c]$, for $c = 1, 2, \ldots, 10$. We experiment with the following aggregation operations: minimum (min), sum, and a multi-tasking setting that combines minimum (min) and maximum (max). This task involves distinguishing relevant from irrelevant categories and pattern matching across differing train and test distributions. The task also demands basic conditional reasoning since each query involves only a subset of all given categories, and algorithmic computation to derive the correct outputs. The textual input is tokenized and processed through an embedding layer, while the numeric input passes through a linear layer. The results, shown in Figure 6, indicate that the positional Transformer outperforms the Transformer.[5]

## 8. Limitations and future work

Our work provides a theoretical and experimental investigation of learning algorithms with positional attention. However, further research is needed in some directions that

---

[5] In Appendix D.3 we compare the performance of positional Transformer against a fine-tuned version of GPT2 (Radford et al., 2019). Positional Transformer has better OOD performance. Note that this task does not demonstrate/suggest that positional Transformer will outperform other Transformer-based architectures (like LLMs) on general-purpose reasoning on text data.

we believe are important. Theoretically, more precise OOD analyses are necessary to capture finer differences between architectures, as existing OOD generalization bounds are not tight enough (see Appendix G for an extended discussion). Empirically, our approach relies on fixed positional encodings, limiting its applicability to longer inputs. Developing positional encodings that support arbitrary input lengths would enable us to explore the length generalization capabilities of positional attention.

## Acknowledgements

The authors would like to thank Federico Barbero (Google DeepMind) and Csaba Szepesvári (Google DeepMind) for their valuable comments and suggestions on this work.

K. Fountoulakis would like to acknowledge the support of the Natural Sciences and Engineering Research Council of Canada (NSERC). Cette recherche a été financée par le Conseil de recherches en sciences naturelles et en génie du Canada (CRSNG), [RGPIN-2019-04067, DGECR-2019-00147].

S. Yang would like to acknowledge the support of the Natural Sciences and Engineering Research Council of Canada (NSERC).

G. Giapitzakis would like to acknowledge the support of the Onassis Foundation - Scholarship ID: F ZU 020-1/2024-2025.

## Impact Statement

Our theoretical and empirical results demonstrate the potential of neural networks in addressing complex reasoning tasks, which are traditionally deemed exclusive to classical computational methods. This research enhances our understanding of neural network capabilities. While this work is primarily theoretical and relies solely on synthetic data, we must acknowledge that societal implications may emerge from our findings. However, at this stage, we do not identify any specific consequences worth highlighting here.

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

# Appendix

**Table of contents**

## A. Background and definitions

### A.1. Massively Parallel Computation (MPC)

The MPC model is a computational model of the MapReduce framework (Dean & Ghemawat, 2008), widely used for computations over massive datasets. It defines a parallel computing model that jointly executes a function across multiple machines, each constrained by limited memory capacity. The MPC model is capable of representing various parallel algorithms (Im et al., 2023) and is more powerful than other established parallel models, such as parallel random-access machine (PRAM) (Andoni et al., 2018).

For completeness, we provide a simplified version of the definition of the MPC protocol by (Andoni et al., 2018), which makes the connection to our proxy computational model more apparent[6].

**Definition A.1** (MPC protocol, Def. I.1 (Andoni et al., 2018), simplified)**.** Let $s$ be a parameter. There are $p \geq 1$ machines (processors), each with local memory of size $s$. The input is distributed on the local memory of some of the machines. The computation proceeds in rounds. In each round, each machine computes the data in its local memory and sends messages to other machines at the end of the round. The total size of messages sent or received by a machine in a round is bounded by $s$. In the next round, each machine only holds the received messages in its local memory. At the end of the computation, the output is in the memory of some machines.

---

[6]The original definition includes assumptions about the relation between the input size, the memory size, and the number of machines which we omit to better highlight the contrast between MPC and PCOC.

## A.2. Parallel Computation with Oracle Communication (PCOC)

In this section, we present the definition of PCOC used in Section 5 and Appendix B to establish the expressivity of positional Transformers. We emphasize that PCOC is merely a proxy model to establish our expressivity results. It is not intended to serve as a competing parallel computational model, such as MPC.

The PCOC model consists of two steps at each round. The first step is communication, where machines send and receive data from other machines. The communication pattern can change at every round. The second step is computation, where all machines perform some local computation on data stored in their local memory.

**Oracle communication.** For each length $n$ and input data, we assume the existence of an oracle that provides the destination for each machine and message at each round. The oracle executes a communication pattern agnostic to the data being processed. This contrasts with other parallel models, such as the Massively Parallel Computation (MPC) model (Andoni et al., 2018), where it is assumed that the processed data can influence communication patterns. At first glance, introducing such an oracle might not seem particularly useful, mainly because it is fixed for each input data, where the data can be real-valued, which implies potentially unaccountably many communication oracles for a particular task. However, its importance lies in the fact that for a variety of parallel algorithms, the communication pattern at each round depends only on the length of the input and is independent of other input data. For example, in algorithms that compute basic statistics such as the sum or minimum or tasks such as sorting a list of numbers, the communication between machines is determined not by their values but by their positions. This means that if the values at each machine change, the communication pattern between machines for a given task and input length remains the same. In Appendix A.3, we illustrate communication patterns for some of these tasks, which are also addressed in the experiments in Section 7.

**Definition A.2** (PCOC model). The PCOC model is described by a set of $n$ machines labeled from 1 to $n$, the number of rounds $R$, an integer $s$ and an oracle $\mathrm{RCV} : [R] \times [n] \to [n] \times (\mathcal{P}([s]) \setminus \{\emptyset\})$ (which is fixed for a given $n$ and input data) satisfying the following:

1. Each machine $i$ has a local memory $\mathtt{MEM}_i \in \mathbb{T}^s$ of size $s$, where $\mathbb{T}$ is some abstract data-type. The contents of the memory are indexed from 1 to $s$ and we use the notation $\mathtt{MEM}_i[j]$ to refer to the element at position $j$ on the memory of the $i$-th machine.

2. Each machine performs some local computation on the data it receives and overrides the results to its memory. A single machine can perform different local computations on different rounds and different machines (generally) perform different local computations.

3. When $(r, i)$, where $r \in [R]$ and $i \in [n]$, is passed to the oracle it returns a subset $M$ of the set $[n] \times (\mathcal{P}([s]) \setminus \{\emptyset\})$. The oracle essentially returns a (possibly empty) set of machines that machine $i$ has to receive some data from in round $r$ along with the exact positions on the memories of those machines to retrieve.

4. The total size of data sent and received by each machine is at most $s$. Size here is measured in terms of the number of "variables" of data-type $\mathbb{T}$.

The protocol is executed in $R$ rounds. At the start, the input is distributed across the $n$ machines. At the beginning of round $r$, each machine $i$ simultaneously queries the oracle with input $(r, i)$ and receives data from the machines it returns. The machines then simultaneously perform their local computations on the received data.

---

**Input:** $\mathtt{Data} = (\mathtt{Data}_1, \ldots, \mathtt{Data}_n)$ distributed across the memories of $n$ machines, labeled in $[n]$. An oracle $\mathrm{RCV}_{n,\mathtt{Data}} : [R] \times [n] \to [n] \times (\mathcal{P}([s]) \setminus \{\emptyset\})$.

1 **For** each round $r = 1, \ldots, R$ **then**

2      Each machine $i$ simultaneously queries $\mathrm{RCV}_{n,\mathtt{Data}}$ with $(r, i)$ as input and receives

     data from the machines and memory positions returned by the oracle.

3      The machines simultaneously perform local computations on the received data

     and write the results in their local memories.

---

### A.3. Illustration of parallel algorithms

In this section, we further expand on the discussion of Section 5, stating that communication in several parallel algorithms depends only on the identification of the machines rather than their current values. To illustrate this, we provide some concrete examples.

**(Cumulative) sum/min.**       **Sorting (Odd-Even sort)**

Figure 7: Illustration of the computational graph for algorithms such as (cumulative) sum/minimum (on the left) and sorting (on the right) for $n = 4$. Circles indicate the machines, each indexed by the subscript. The superscript indicates each round of the algorithm. At round 0, machine $i$ holds the $i$-th element of the input. The cumulative version of the sum/minimum algorithm includes all arrows (black and orange), while the non-cumulative version is represented only by the orange arrows.

These tasks are examples of those presented in Section 7. Note that these illustrations do not indicate the computational graphs derived by our architecture, as there are multiple ways to achieve the same goal, and they do not necessarily involve the neat graph structures shown in Figure 7. For a more in-depth analysis of the results obtained by our architecture, we refer the reader to Appendix D.

In these computational graphs, we represent each machine by a circle, distinguished by a subscript (from 1 to 4, since $n = 4$). Furthermore, we use superscripts to denote the rounds of the algorithm, with superscript 0 representing the initial stage where no computation or communication is performed. Note that no specific values are provided in these examples. This indicates that the correct results can be achieved by following the computational graph for any set of values. In the subsequent rounds, each machine receives information from other machines (including itself) and performs some computation. For each algorithm in Figure 7, we will briefly describe the computations involved, noting that this is not the main focus of the paper and serves only as motivating examples.

For the computation of the minimum and the summing function, each machine applies the minimum (or sum) operator to all incoming values from the previous round. By iteratively applying this operator over multiple rounds, machine 4 ultimately obtains the global minimum value (or total sum) across all $n$ entries, while the other machines hold cumulative minima (or sums) up to their positions. For the non-cumulative versions of these algorithms, the local computations are the same as the cumulative versions, and the communication paths are denoted in orange and form a binary tree pattern.

For sorting, the graph on the right of Figure 7 represents the odd-even transposition sort algorithm (Habermann, 1972). This algorithm works by alternating communication between adjacent machines, starting with the odd-indexed pairs (machines with indices 1 and 2, 3 and 4, etc.), then switching to even-indexed pairs (machines with indices 2 and 3, in this example). In stages where two machines communicate, the machine with the lower index picks the minimum value among the two, while the machine with the higher index picks the maximum. The procedure runs for a total of $n - 1$ rounds, after which the values are sorted in ascending order.

### A.4. Additional definitions

In this section, we provide important definitions that are used throughout our theoretical results. For $n \in \mathbb{N} := \{1, \dots\}$, we use $[n]$ to refer to the set $\{1, \dots, n\}$. For two nonnegative integers $m, n \in \mathbb{N} \cup \{0\}$ we use $m \veebar n$ to refer to the exclusive or of $m$ and $n$. Next, we define the notion of Hamming neighbors:

**Definition A.3** ($j^{\text{th}}$ Hamming neighbors). Two nonnegative integers $x, y \in [n] \cup \{0\}$ are $j^{\text{th}}$-*Hamming neighbors*, denoted $x \sim_j y$, if:

$$x \sim_j y \iff (\text{bin}(x)_i = \text{bin}(y)_i \; \forall i \neq j) \text{ and } (\text{bin}(x)_j \neq \text{bin}(y)_j),$$

where $\text{bin}(x)$ is the binary representation of $x$ with $\lceil \log_2 n \rceil$ bits ordered from most to least significant bit.

Finally, we give the definition of a Beneš network (Beneš, 1965):

**Definition A.4** (Beneš network). Let $n = 2^k$, $k \in \mathbb{N}$. A Beneš network is a layered directed graph consisting of $L = 2k$ layers with each layer having $n$ nodes and $2n$ directed edges between two sequential layers. Nodes in each layer are labeled with integers from 0 to $n - 1$. We identify each node in the network by a tuple $(\ell, i)$ where $\ell \in [2k]$ is the layer, and $i \in \{0, 1, \ldots, n - 1\}$ is the node label within layer $\ell$. Two nodes $(\ell, i)$ and $(\ell', j)$ are connected by an edge *iff*

$$\ell' = \ell + 1 \text{ and } \left(i = j \text{ or } i \sim_{k - |k - \ell|} j\right).$$

An example of a Beneš network with $n = 2^3$ nodes in each layer is shown in Figure 8.

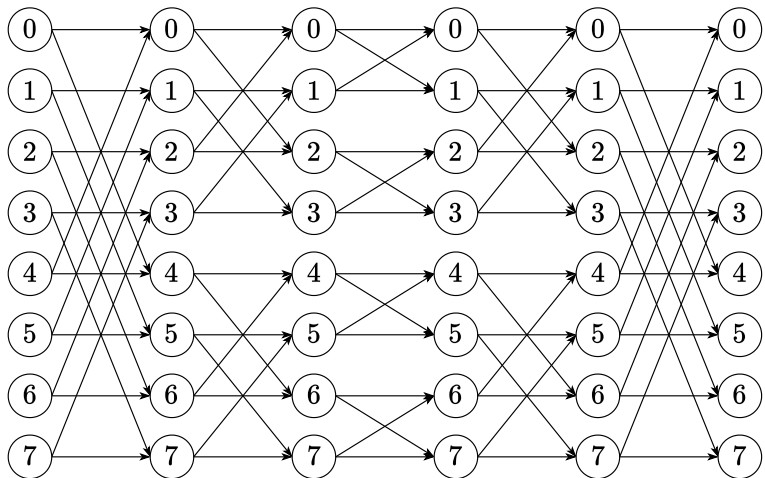

Figure 8: A Beneš network with $n = 2^3$ nodes in each layer.

# B. Expressivity results

### B.1. Proof of Theorem 5.1 (Positional Transformers can simulate MPC)

In this section, we prove our main result that establishes the simulation of Massively Parallel Computation (MPC) model by Positional Transformers. Towards this goal, we present two lemmas that bridge positional Transformers and MPC. More specifically, we utilize a proxy model that has a static communication network. We call this proxy model Parallel Computation with Oracle Communication (PCOC), established in Definition A.2. *We note that PCOC is only used to establish our expressivity result. It is not meant as a model to compete with established parallel models, such as MPC.* We begin by outlining its main features, followed by the two lemmas that lead to our final result.

The PCOC model operates on $n$ machines labeled from 0 to $n - 1$ over $R$ rounds, with each machine having a local memory of size $s$. The model consists of two fundamental steps per round: computation and communication. During computation, machines perform local computations on data stored in their memory. In the communication step, machines exchange data through an oracle that provides destination information for each machine and message, executing a communication pattern that is agnostic to the processed data and depends only on the input length. Critically, the oracle's communication pattern remains fixed for a given input length and task, meaning that if the values at each machine change, the communication pattern between machines remains the same, a property that is observed in many parallel algorithms such as computing basic statistics or sorting, where communication is determined by positions rather than values.

This contrasts with other parallel models, such as MPC, where the communication pattern is more flexible and directly influenced by the data being processed. In the MPC model, each machine has a limited memory size of $s$, and the input data

is distributed among the machines. The computation proceeds in rounds: during each round, a machine processes the data in its memory, exchanges messages with other machines (up to a total size of $s$), and retains only the messages it received for the next round. After all rounds, the final output is stored in the memory of one or more machines. Unlike PCOC, the communication in MPC is not predetermined by the input length or task alone; instead, it can dynamically depend on the data being processed, provided the total communication per machine per round does not exceed the memory limit $s$.

Even with limited communication capabilities, PCOC can perform the same functions as MPC, incurring only a logarithmic cost in rounds relative to the number of machines, $N$. The proof of this lemma is presented in Appendix B.2.

**Lemma B.1.** *Consider an instance of MPC T with $R$ rounds, $N$ machines with local memory $s$. For any instance T, there exists an instance of PCOC P with $2R\lceil \log N \rceil$ rounds, $N$ machines with local memory $2s$ that simulates T.*

An analogy to illustrate the distinction between the two approaches can be drawn from a mailing system. In the MPC model, each processor acts as an individual delivering letters directly to their recipients. Communication is flexible and determined by the specific needs of the task. In contrast, PCOC resembles a structured mailing system with predefined routes and sorting stations. Each individual sends their letters to the nearest sorting station, where the letters are organized into bins and passed through successive stages of sorting and delivery. After several rounds, all letters reach their destinations, following a fixed communication pattern regardless of the specific contents.

Having established the PCOC model, we now show that the positional Transformer model in Equation (1) can simulate it. More specifically, our results show that a $R$-round PCOC protocol can be simulated by a positional Transformer with $R$ layers.

**Lemma B.2.** *Consider a PCOC instance P with $R$ rounds, $N$ machines with local memory $s$, and data type $\mathbb{T} = \mathbb{R}$. Let $\mathcal{M}$ be a model following the architecture in Equation (1) with $n = N + 1$ nodes, $R$ layers and $s$ attention heads. Then, for any instance P with Borel measurable local functions, there exists a configuration of $\mathcal{M}$ that approximates P to any desired degree of accuracy.*

The proof of the above lemma can be found in Appendix B.4, where we show that each round of computation and communication performed by PCOC can be approximated by one layer of positional Transformers. Having established the two necessary lemmas Lemma B.1 and Lemma B.2, we are ready to re-state our main result.

**Theorem 5.1.** *Consider an instance of MPC T with $R$ rounds, $N$ machines with local memory $s$. Let $\mathcal{M}$ be a model following the architecture in Equation (1) with $n = N + 1$ nodes, $2R\lceil \log N \rceil$ layers and $2s$ attention heads. Then, for any instance T with Borel measurable local functions, there exists a configuration of $\mathcal{M}$ that approximates T to any desired degree of accuracy.*

The proof of this result follows directly from the combination of the two preceding results: MPC can be simulated by PCOC, which can be simulated by positional Transformers. Notably, the structure of MPC enforces that local memory can only grow sublinearly relative to the input size. This constraint prevents trivial simulation configurations in which all data is sent to a single node, which would undermine the purpose of parallelism.

### B.1.1. DISCUSSION OF REMARK 5.2

In Sanford et al. (2024), the authors present not only a simulation of MPC using standard Transformers but also the reverse, i.e., a simulation of Transformers by MPC. From the latter, we derive the following remark that establishes an equivalence between standard and positional Transformers.

*Remark* 5.2. Consider a standard Transformer $\mathcal{M}'$ with $N$ nodes, $L$ layers, and $d_V \cdot H$ sublinear in $N$. Then, there exists a positional Transformer $\mathcal{M}$ with $O(N^2)$ nodes, $O(L \log N)$ layers, and $d_V \cdot H$ sublinear in $N$ that can simulate the computation of $\mathcal{M}'$.

An important observation from Sanford et al. (2024) is that the embedding dimension, denoted as $d_V$, also governs other intermediate dimensions, such as the embedding dimension $d_m$ of the matrices $W_K$ and $W_Q$. However, in positional Transformers, these quantities are decoupled. Specifically, the embedding dimension $d_m$ is directly influenced by the positional encodings and scales linearly with the number of nodes.

It is worth noting that the observed quadratic dependency on the number of nodes in positional Transformers is primarily a consequence of the specific simulation strategy employed in Sanford et al. (2024). These dependencies should not be taken too literally, as the results do not provide a sharp characterization of the two computational models, an aspect acknowledged

by Sanford et al. (2024). Moreover, alternative and more efficient simulation strategies may exist. Additionally, this quadratic cost arises only when simulating the same sequence of operations across architectures. In practice, the model may converge to parameter configurations that do not correspond to interpretable algorithms.

## B.2. Proof of Lemma B.1 (PCOC can simulate MPC)

The goal of this section is to show that MPC can be efficiently simulated via PCOC (that is, in a data-agnostic way). For completeness, we re-state the lemma below:

**Lemma B.1.** *Consider an instance of MPC* $\mathsf{T}$ *with $R$ rounds, $N$ machines with local memory $s$. For any instance $\mathsf{T}$, there exists an instance of PCOC* $\mathsf{P}$ *with $2R\lceil \log N \rceil$ rounds, $N$ machines with local memory $2s$ that simulates $\mathsf{T}$.*

We break the proof in two steps: first, we show that MPC routing can be simulated through a fixed communication pattern given by the connectivity of a Beneš network. Subsequently, we show how this routing can be accomplished in a data-agnostic way, consistent with the PCOC definition.

Consider an MPC instance with $p$ machines, each with memory size $s$. Assume messages are encoded in the following way: machine $i$ holds a list $\text{SEND}_i$ containing tuples of the form $(i, \mathsf{b}, \text{pos}, j)$ where $\mathsf{b} \in \mathbb{A}$, $\text{pos} \in \mathbb{N}$ and $j \in [p]$, where $\mathsf{b}$ is the pos-th character in the message machine $i$ sends to machine $j$. Here, $\mathbb{A}$ denotes the alphabet used to encode the messages (for example, $\mathbb{A}$ can be $\{0, 1\}$ if we are representing messages by their bits). As a concrete example, if machine 0 sends 010110 to machine 1 then

$$\text{SEND}_0 = \{(0, 0, 1, 1), (0, 1, 2, 1), (0, 0, 3, 1), (0, 1, 4, 1), (0, 1, 5, 1), (0, 0, 6, 1)\}$$

By the assumptions of the MPC model, $|\text{SEND}_i| \leq s$ for all $i \in \{0, 1, \ldots, p-1\}$. Similarly, denote by $\text{RCVD}_i$ the list of source-character-position-destination tuples that machine $i$ receives. That is:

$$\text{RCVD}_i = \left\{ (k, \mathsf{b}, \text{pos}, j) \in \bigcup_{i'} \text{SEND}_{i'} \,\middle|\, j = i \right\} \quad \text{for all } i \in \{0, 1, \ldots, p-1\}$$

Again, by the assumptions of the MPC model, we have $|\text{RCVD}_i| \leq s$ for all $i \in [p]$. We further impose the following simplifying assumptions:

1. The number of machines $p$ is a power of two, i.e. $p = 2^k$ for some $k \in \mathbb{N}$, and

2. For each pair of machines $i, j$, the number of tuples sent from machine $i$ to machine $j$ is either 0 or at least $2^{k-1}$, where $k$ is as above.

Note that the above assumptions do not limit the MPC model since we can always add a couple dummy machines to reach a power of two and pad shorter messages with a special token (potentially having to scale $s$). The second assumption can be lifted altogether at the expense of a more complicated proof. We chose to work with both assumptions in order to highlight the essence of the proposed routing algorithm. As we mentioned previously, to highlight the routing strategy we split the analysis into two parts: first we show how we can simulate one round of MPC using a Beneš network routing without the need for extra memory, assuming access to destination information. Building on that, we then show how to extend the Beneš routing protocol to the case where message routing is independent of destinations, at the expense of doubling the memory of each machine. The latter shows that PCOC can simulate one round of MPC with the same number of machines and double the memory. The result is then trivially extended to $R$ rounds of MPC by iterating the routing algorithm $R$ times.

**Data-dependent Beneš routing**   We begin by providing a data-dependent routing algorithm that simulates one round of MPC using a Beneš network. Under the two assumptions made above, the routing algorithm executes in $2\log_2(p) = 2k$ steps and is as follows:

- Initially, the memory $M_i$ of each machine holds the list $\text{SEND}_i$ (i.e. the messages it needs to send in the source-character-position-destination format described above).

- At step $L \in \{1, \ldots, k-1\}$, machine $i$ partitions the contents of its memory as

$$\bigcup_{j=1}^{p} M_{i,j}$$

where $M_{i,j} = \{(k, \mathsf{b}, \mathrm{pos}, j') \in M_i \mid j' = j\}$. Let $(L+1, m)$ denote the neighbor of $(L, i)$ at level $L+1$ with a label different from $i$ (so $m \neq i$). Machine $i$ partitions each $M_{i,j}$ into two equal parts (this is possible since every message is a power of two) and sends one part to machine $i$ and the other part to machine $m$. Steps $0$ through $k-1$ are collectively referred to as the "fragmentation phase" of the algorithm.

- At steps $L \in \{k, \ldots, 2k-1\}$, denote by $\mathtt{RCVD}_i^{(L)}$ the source-character-position-destination tuples received by machine $i$ after $L-1$ steps. Also, let $(L+1, m)$ denote the neighbor of $(L, i)$ at level $L+1$ with a label different from $i$. At step $L$, machine $i$ sends all tuples $(k, \mathsf{b}, \mathrm{pos}, j) \in \mathtt{RCVD}_i^{(L)}$ such that there is a path between nodes $(L+1, i)$ and $(2k, j)$ in the Beneš network to machine $i$, and the rest to machine $m$. Notice that by the construction of the Beneš network, exactly one of following holds:

    – There is a path between nodes $(L+1, i)$ and $(2k, j)$, or
    – There is a path between nodes $(L+1, m)$ and $(2k, j)$

  Steps $k$ through $2k-1$ are collectively referred to as the "collection phase" of the algorithm.

- At step $2k$, the nodes in the last layer recombine the received messages using the source and position information and the algorithm terminates.

We show correctness of our algorithm using a two-step argument. First, we show that during the fragmentation phase (first $k-1$ steps of the algorithm) the tuples are evenly distributed ensuring that the at every point no machine receives more than $s$ tuples. Then we proceed by showing that messages are correctly routed and reach their destination.

**Fact 1.** *After $L \in \{0, 1, \ldots, k-1\}$ steps, machine $i$ receives exactly $1/2^L$ of each $\mathtt{SEND}_{i'}$ where $i'$ is such that $i$ and $i'$ agree on the first $k-L$ least significant bits. Furthermore, no two machines receive the same part of the same list.*

*Proof.* The proof is by induction. For $L = 0$ the result trivially holds by the initialization step of the algorithm. Assume it holds for some $L < k-1$. Let $M_i$ be the memory of machine $i$ after $L$ steps. By the connectivity of the Beneš network, after $L+1$ steps, each machine $i$ will have received messages from itself and machine $m := i \veebar 2^{k-L-1}$. From the inductive hypothesis, $M_i$ contains $1/2^L$ of each $\mathtt{SEND}_{i'}$ where $i'$ is such that $i$ and $i'$ agree on the first $k-L$ least significant bits, and similarly, $M_m$ contains $1/2^L$ of each $\mathtt{SEND}_{i'}$ where $i'$ is such that $m$ and $i'$ agree on the first $k-L$ least significant bits. Now $i$ and $i'$ agree on the first $k-L-1$ least significant bits if and only if they agree on the first $k-L$ least significant bits or $m$ and $i'$ agree on the first $k-L$ least significant bits. By construction, machine $i$ will receive half the contents of $M_i$ and $M_m$ and it will hence hold exactly $1/2^{L+1}$ of each $\mathtt{SEND}_{i'}$ where $i'$ is such that $i$ and $i'$ agree on the first $k-(L+1)$ least significant bits. Since there is at most one path between any two nodes in the Beneš network, the second part of the statement is trivially satisfied. This concludes the proof. $\square$

Fact 1 immediately shows that for the first $k-1$ steps of the algorithm, the contents on the memory of each machine never exceed $s$ (since the length of each $\mathtt{SEND}_i$ is bounded by $s$). Similarly, we can show the following fact:

**Fact 2.** *After $L \in \{k, k+1, \ldots, 2k-1\}$ steps of the algorithm, machine $i$ receives exactly $1/2^{2k-L-1}$ of all $\mathtt{RCVD}_j$ where $j$ is such that $i'$ and $i$ agree on the first $L-k+1$ least significant bits. Furthermore, no two machines receive the same part of the same list.*

*Proof.* We shall only prove the base case $L = k$ as the inductive step is completely symmetric to the proof of Fact 1. After the $k$-th step of the algorithm, machine $i$ will receive all tuples for which the destination $j$ is such that $(2k, j)$ is reachable from $(L+1, i)$ in the Beneš network from itself and from machine $m := i \veebar 2^0$. Recall from Fact 1 that before the $k$-th step is executed, the memory of machine $i$, $M_i$, contains $1/2^{k-1}$ of every $\mathtt{SEND}_{i'}$ where $i'$ is such that $i$ and $i'$ agree on the least significant bit. Similarly the memory of machine $m$, $M_m$, contains $1/2^{k-1}$ of every $\mathtt{SEND}_{i'}$ where $i'$ is such that $m$ and $i'$ agree on the least significant bit. But since the least significant bit of $m$ and $i$ differ, the memories of machines $m$ and $i$ combined, $M_i \cup M_m$, contain exactly $1/2^{k-1}$ of $\bigcup_{i'} \mathtt{SEND}_{i'}$ (i.e. a $1/2^{k-1}$ fraction of all messages). Combining the above we see that after the $k$-th step is executed, machine $i$ will receive $1/2^{k-1} = 1/2^{2k-L-1}$ of every $\mathtt{RCVD}_j$ where $j$ is such that $i$ and $j$ agree on the first $L-k+1 = 1$ least significant bits. As in Fact 1, the uniqueness of paths in the Beneš network guarantees that no two machines ever receive the same part of the same list. $\square$

A direct application of Fact 2 shows that after $2k - 1$ steps of the algorithm, machine each machine $i$ will have in memory exactly $\text{RCVD}_i$, showing correctness. Finally, the fact that $|\text{RCVD}_i| \leq s$ for all $i \in [p]$ shows that after every step $L \in \{k, k+1, \ldots, 2k-1\}$ of the algorithm each machine receives no more than $s$ tuples. This construction shows that even if we restrict the communication of MPC to some fixed routing pattern, we can still achieve any and all possible routing patterns by using the Beneš network efficiently, paying the cost of logarithmically more rounds. We now proceed to show how we can further extend this algorithm to be consistent with our PCOC model.

**From data-dependent to data-agnostic Beneš routing**   The algorithm above brings us one step closer to our ultimate goal of data-agnostic simulation since at every step, each machine will send data to at most two other machines, respecting the connectivity of the Beneš network. However, both the fragmentation and collection phases of the algorithm (steps 0 to $k-1$ and $k$ to $2k-1$) implicitly require access to destination information for each message. In terms of PCOC, assuming the oracle mimics the algorithm execution, it would require access to destination information, which is not allowed in the PCOC model. This is because of the following reasons:

1. During the fragmentation phase it needs to know exactly which messages are destined for each machine $j$ to evenly divide them into two parts and route them to the two receiving machines.

2. During the collection phase, destination information is directly used to route messages.

While, at first, these obstacles might seem hard to circumvent, we can do so by using a few simple techniques that enforce a specific structure on the output of the local computation:

1. Observe that we can eliminate the need for destination information during the fragmentation phase if we assume that the tuples in the memory of each machine are always sorted based on their destination. If this is the case each machine can simply send the tuples in the even and odd indices on its memory to the two receiving machines, respectively. This achieves the desired outcome: for each $j$, exactly half of the messages with destination $j$ will get routed to one machine and the rest to the other.

2. As previously discussed, during the collection phase, the oracle would need access to the destination to decide where to route each tuple. However, if we assume that the memory size of each machine is $2s$ (instead of $s$) we can still overcome this issue. Suppose the oracle needs to simulate step $L \in \{k, k+1, 2k-1\}$ of the algorithm. For each machine $i$, let $i_m \neq i$ be such that $(L, i)$ and $(L+1, i_m)$ are connected in the Beneš network. Recall that at step $k$, machine $i$ will send data to itself and machine $i_m$, depending on the final destination. Given the fixed structure of the Beneš network, we can require the local computation to calculate where each tuple needs to be routed ($i$ or $i_m$) and place it either in the first half of the memory slots (if it needs to be routed to $i$) or to the second half (if it needs to be routed to $i_m$). The oracle will then route the first half of the memory to $i$ and the second half to $i_m$. Empty spaces on each half can be padded with special tokens.

## B.3. Hardmax patterns using positional attention

In this section, we show that the positional attention architecture in Equation (3) can approximate any unique hardmax pattern, a concept we define later in this section. This result is used to support the simulation results of Lemma B.2. We begin by stating the definition of the row-wise hardmax transformation for a $p \times q$ matrix $X$ from Section 3:

$$\text{hardmax}(X)_{i,j} = \begin{cases} 1 & \text{if } X_{i,j} = \max_{k \in [q]} X_{i,k} \\ 0 & \text{otherwise} \end{cases} \quad \text{for } i \in [p], j \in [q], \tag{6}$$

where we implicitly extend the definition for vectors in $\mathbb{R}^n$ by viewing them as $1 \times n$ matrices.

We use the term *hardmax pattern* to refer to any matrix in the image of hardmax (i.e. a binary matrix with at least one non-zero element in every row). Furthermore, we use the term *unique hardmax pattern* to refer to hardmax patterns with exactly one non-zero element in every row. Unique hardmax patterns occur when the input matrix has a unique maximum value in every row.

We further define key concepts that will be used for the more formal re-statement of Lemma B.3. Let the input have $n$ rows, and the binary positional encodings be defined by the positional encoding matrix $P = I_n$, therefore having $d_P = n$. Finally, let $T$ be a positive scalar that represents a temperature parameter that controls the approximation of softmax.

**Lemma B.3.** *For any given $n \times n$ unique hardmax pattern $\bar{A}$, there exists a configuration of node positional attention parameters in Equation* (3) *and a temperature parameter $T$ such that the resulting softmax pattern $A$ approximates $\bar{A}$ to any desired level of accuracy. Formally, for any unique hardmax pattern $\bar{A}$ and any $\varepsilon > 0$, there exists some $T = T(\varepsilon) > 0$ such that the inequality $|\bar{A}_{i,j} - A_{i,j}| \leq \varepsilon$ holds for all $i, j \in [n]$.*

*Proof.* Without loss of generality, we may assume $\varepsilon < 1$. We start by setting the node positional attention parameters to be $W_K = I_n$ and $W_Q = T(2\bar{A} - 1)$, where, $T > 0$ is our temperature scalar parameter and $\bar{A}$ is the target pattern. Since, in this construction, node positional encodings are set to be the identity, the inner-product $PW_Q W_K^\top P^\top$ reduces to $W_Q$, where each entry $(i, j)$ is $T$ if $\bar{A}_{i,j} = 1$, and $-T$ otherwise.

This inner product is passed to the softmax operator, resulting in the attention matrix $A$. For each $i, j \in [n]$ we separately analyze the following two cases:

1. Case $\bar{A}_{i,j} = 1$: In that case, the only non-zero element on the $i$-th row of $\bar{A}$ is $\bar{A}_{i,j}$, so we can express the difference as

$$
\begin{aligned}
\bar{A}_{i,j} - A_{i,j} &= 1 - \frac{\exp((W_Q)_{i,i})}{\sum_{k=1}^{n} \exp((W_Q)_{i,k})} = 1 - \frac{\exp(T)}{\exp(T) + \sum_{k \neq j} \exp(-T)} \\
&\leq 1 - \frac{\exp(T)}{\exp(T) + n\exp(-T)} = 1 - \frac{1}{1 + \exp(\ln n - 2T)} \\
&= \frac{\exp(\ln n - 2T)}{1 + \exp(\ln n - 2T)} \leq \exp(\ln n - 2T)
\end{aligned}
$$

2. Case $\bar{A}_{i,j} = 0$: Let $j_0 \neq j$ be the unique index for which $\bar{A}_{i,j_0} = 1$, and we can express the difference as:

$$
\begin{aligned}
A_{i,j} - \bar{A}_{i,j} &= \frac{\exp((W_Q)_{i,j})}{\sum_{k=1}^{n} \exp((W_Q)_{i,k})} = \frac{\exp(-T)}{\exp(T) + \sum_{k \neq j_0} \exp(-T)} \\
&= \frac{1}{\exp(2T) + n - 1} \leq \frac{1}{\exp(2T - \ln n) + 1} \leq \exp(\ln n - 2T)
\end{aligned}
$$

In any case, we have that $|\bar{A}_{i,j} - A_{i,j}| \leq \exp(\ln n - 2T)$. Therefore, by taking $T \geq 1/2 \ln(n/\varepsilon)$, we have $|\bar{A}_{i,j} - A_{i,j}| \leq \varepsilon$. $\square$

## B.4. Proof of Lemma B.2 (Positional Transformers simulate PCOC)

We begin this section by outlining the key concepts utilized in the routing protocol employed in our constructions. First, we describe the general structure of the input matrix $\mathbf{X}$.

B.4.1. ENCODING

**Input matrix:** In alignment with the PCOC model, the input matrix $\mathbf{X}$ represents $N$ machines, where each machine is denoted by a row in the matrix, and its local memory, $\texttt{MEM}_i \in \mathbb{T}^s$, is represented by the corresponding columns. The maximum size of data that any machine can send or receive is $s$ bits, with each bit corresponding to a column in $\mathbf{X}$.

However, the actual number of rows and columns in $\mathbf{X}$ differs from the number of machines and the local memory size for two reasons:

1. **Sink node**: A dummy node is introduced to facilitate all possible communication patterns in PCOC using positional attention. This is necessary because PCOC allows for the possibility of information not being sent to any receiving machine. This scenario is incompatible with the softmax formulation, which requires at least one non-zero entry. The dummy node serves as a sink, collecting all messages that do not have a destination. Consequently, the number of rows in $\mathbf{X}$ is $n = N + 1$.

2. **Unique node identifier:** Each machine also requires a unique identifier to enable element-wise local computations. To achieve this, we encode a unique scalar for each node in the last column of $\mathbf{X}$, resulting in a feature dimension of $d_X = s + 1$ for the input matrix.

As discussed in Section 5, in PCOC, routing is set by an oracle that decides how packets of data should be routed at each round. Under this framework, routing must be performed to prevent multiple data from being sent to the same destination. Since our construction relies on matrix operations, this leads to the following assumption:

**Assumption B.4.** (No-collision). For any layer $\ell \in [L]$, no two different machines $i_1, i_2 \in [N]$ should route data to the same destination $i_3 \in [N]$ for the same column $j \in [s]$, where each column represents a bit of local memory across the nodes.

Note that this assumption does not limit the generality of our PCOC model. It only defines how data should be stored in the memory of each receiving machine, and any valid PCOC routing has a corresponding no-collision configuration of bits that realizes it due to the restriction on the total size of received data. As demonstrated in the constructive proof, this directly influences the sparsity pattern generated by each attention head.

**Positional encodings:** As previously mentioned, although the connectivity at each layer may vary, the positional encodings remain consistent across all layers. Our architecture simulates MPC using node positional encodings with dimension $d_P = n$ by setting $P = I_n$, with each positional encoding functioning as a standard basis vector.

B.4.2. SIMULATION RESULTS

We now demonstrate that, with the established encoding, the architecture provided in Section 4 can simulate the execution of any PCOC instance. Each round of such a PCOC instance can be decomposed into two stages: communication and computation. Our objective is to provide existential results for both stages.

In the communication stage, routing assigns destination machines for each message. In our architecture, this assignment is analogously captured by the attention weights, which determine whether a message should be received by a node using binary values.

The no-collision assumption ensures that all routing patterns can be represented by unique hardmax patterns. As expressed in Lemma B.3, since any unique hardmax pattern can be approximated by our attention layer using softmax, for simplicity, the subsequent proofs use hardmax instead of softmax. With all details now established, we re-state our main simulation result:

**Lemma B.2.** *Consider a PCOC instance* P *with R rounds, N machines with local memory s, and data type* $\mathbb{T} = \mathbb{R}$*. Let* $\mathcal{M}$ *be a model following the architecture in Equation* (1) *with* $n = N + 1$ *nodes, R layers and s attention heads. Then, for any instance* P *with Borel measurable local functions, there exists a configuration of* $\mathcal{M}$ *that approximates* P *to any desired degree of accuracy.*

*Proof.* Despite the desired degree of accuracy being influenced by the number of rounds performed, it suffices to show that one layer of our architecture can simulate one round of PCOC. The same constructive arguments can be extended to more rounds, ensuring the overall degree of approximation is respected[7]. To this end, we begin the proof with the communication stage.

**Communication:** In PCOC, communication is encoded as routing patterns determined by the oracle. At round $\ell \in [R]$, we denote by $H^{(\ell)} = \{((i,j), K) \mid i, j \in [N], K \in \mathcal{P}([s])\}$ the set of valid routing patterns provided by the oracle. This set specifies that the data at positions $K$ in the local memory of machine $i$ must be sent to machine $j$ at the same position. A valid routing pattern requires that no collisions occur (i.e., no two triplets in $H^{(\ell)}$ should have the same destination $j$ and memory position $k \in K$). We further denote by $H_z^{(\ell)} = \{((i,j), z) \mid ((i,j), K) \in H^{(\ell)}, z \in K\}$ the subset of routing patterns corresponding to position $z$ in local memory.

The first part of our result constructively demonstrates that positional attention can reproduce any valid routing set by the oracle that adheres to the PCOC model. We construct $s$ attention heads indexed by $h \in [s]$, which handle routing for the corresponding subset $H_h$.

For clarity in the construction phase, we introduce an augmented set to simplify notation. We begin by extracting the set of source nodes for each set $H_z^{(\ell)}$, denoted as $I_z^{(\ell)} = \{i \mid ((i,j), z) \in H_z^{(\ell)}, j \in [N], z \in [s]\}$. Next, we create a complement

---

[7]To completely align the simulation results with PCOC execution, positional Transformers require an extra initial layer where the attention layer acts as the identity. This is to account for the fact that PCOC first starts with a computation stage rather than communication as in Transformers.

set $\mathcal{H}_z^{(\ell)}$, which routes all unused sources (i.e., those not in $I_z^{(\ell)}$) to the sink node labeled $n = N + 1$. We denote this complement set by $\mathcal{H}_z^{(\ell)} = \{(i, n, z) \mid i \in [n] \setminus I_z^{(\ell)}\}$. Finally, we define the union of these sets as $\hat{H}_z^{(\ell)} = H_z^{(\ell)} \cup \mathcal{H}_z^{(\ell)}$.

The attention parameters are then set as follows:

$$\left(W_K^{(\ell,h)}\right)_{i,j} = \begin{cases} 1 & \text{if } i = j, \\ 0 & \text{otherwise,} \end{cases} \quad \text{and} \quad \left(W_Q^{(\ell,h)}\right)_{i,j} = \begin{cases} 1 & \text{if } (j, i, h) \in \hat{H}_h^{(\ell)} \\ 0 & \text{otherwise,} \end{cases}$$

In this construction, we first observe that both the node positional encodings and the key matrix are identity matrices, reducing the inner product in attention to be solely defined by the query matrix. The query matrix is then designed to encode the source node $i$ as a standard basis vector in the row corresponding to the destination node $j$. This effectively represents the routing set $\hat{H}_h^{(\ell)}$ as a binary matrix, which is also preserved after applying hardmax. Additionally, the no-collision assumption, combined with the sink node strategy, ensures exactly one non-zero entry in the first $N$ rows of the attention weights matrix for each attention head $h$.

For the value and output transformation, we set all value matrices $W_V^{(\ell,h)}$ to be the identity $I_{d_X}$ and define the output matrix $W_O^{(\ell)} \in \mathbb{R}^{(H \cdot d_X) \times (d_X - 1)}$ as follows:

$$\left(W_O^{(\ell)}\right)_{i,j} = \begin{cases} 1 & \text{if } i = k + (h - 1)s, \, j = h \\ 0 & \text{otherwise.} \end{cases}$$

Here, the output matrix $W_O^{(\ell)}$ ensures that only the correct memory position receives the information and places it in the corresponding column. Note that since the outputs of the attention heads are concatenated before being processed by $W_O^{(\ell)}$, the values along the rows of the output matrix also depend on the attention head.

We now focus on the computation stage for the second part of the proof.

**Computation:** At round $\ell \in [R]$ of a PCOC model, let $[\phi_i^{(\ell,z)}]_{i=1}^n$ be the local functions applied by each machine $i \in [N]$ and let $\phi_n^{(\ell,z)}$ correspond to the function of the augmented sink node, which effectively erases all data received. Each function $\phi_i^{(\ell,z)}$ operates on received data in each machine's local memory, which corresponds to the output of the attention layer, denoted by $z$ and outputs a vector of the same dimension $s$, that is, $\phi_i^{(\ell,z)} : \mathbb{R}^s \to \mathbb{R}^s$.

Furthermore, let $\phi^{(\ell,x)} : \mathbb{R}^{d_X} \to \mathbb{R}^{d_X}$ be a function common to all nodes, which operates solely on the residual connection $x$. This function outputs a vector where all entries are zero except the last entry. The value in this last entry corresponds to the unique node identifier extracted from the residual input $x$.

We aim to approximate both $\phi_i^{(\ell,z)}$ and $\phi^{(\ell,x)}$ using neural networks. To this end, we define the combined function $\phi_i^{(\ell)} : \mathbb{R}^{d_X + s} \to \mathbb{R}^{d_X}$ by:

$$\phi_i^{(\ell)}(z_i \oplus x_i) := \phi^{(\ell,x)}(x_i) + \phi_i^{(\ell,z)}(z_i) \oplus 0, \tag{7}$$

where $z_i \oplus x_i$ denotes the concatenation of output of $z_i \in \mathbb{R}^s$ and $x_i \in \mathbb{R}^{d_X}$. We further augment the output of $\phi_i^{(\ell,z)}$ with a zero scalar to match the dimension $d_X = s + 1$.

Let $[\hat{\phi}_i^{(\ell)}]_{i=1}^n$ correspond to multilayer perceptrons (MLPs), each applied to each input $z_i \oplus x_i$. By invoking universal approximation results such as those by (Cybenko, 1989; Hornik et al., 1989b), we assert that as long as the local functions $\phi_i^{(\ell)}$ are Borel measurable, there exist neural networks $\hat{\phi}_i^{(\ell)}$ that can approximate the functions $\phi_i^{(\ell)}$ to any desired degree of accuracy. Additionally, note that the function $\phi_n^{(\ell,z)}$ of the sink node, as well as the function $\phi^{(\ell,x)}$ that operates on the residual connection, are both linear and therefore Borel measurable.

The final step in this argument is to relate these approximations to the proposed architecture in Section 4. Specifically, we use the MLP $\Phi^{(\ell)}$ in Equation (1) and leverage the aforementioned universality results to approximate all the element-wise functions $\phi_i^{(\ell)}$.

A crucial aspect of this step is the need for the input of each machine to be uniquely identifiable. This ensures that a single model can injectively encode multiple functions. Intuitively, it guarantees that each approximation of the local function can identify that it is processing the right row. The unique identification of each machine is guaranteed by the scalar encodings of every node, which, regardless of the contents in local memory, ensure that the input rows are unique. Therefore, the function that $\Phi^{(\ell)}$ has to approximate is a piecewise Borel function with each branch being one of the $\phi_i^{(\ell)}$, based on the unique machine identifier. Such function is Borel measurable, and so the universal approximation results of (Hornik et al., 1989b) hold, guaranteeing the existence of the desired MLP $\Phi^{(\ell)}$.

This demonstrates that our neural network architecture can emulate the computations performed by the local functions $\phi_i^{(\ell,z)}$ acting on the output of the attention layer (with their outputs zero-padded to match the required dimension) and the function $\phi^{(\ell,x)}$ acting on the residual connection, even though they act on distinct parts of the input.

Therefore, we establish that our proposed architecture can approximate the computations in each round of the PCOC model. $\qquad\square$

An important observation is that the computational model and expressive results for the proposed architecture are specific to a fixed input length $n$. Furthermore, one could extend such results to a model with communication and local computations that also consider the input length as an input. For local computations, proof in Lemma B.2 can also cover such cases, provided that the information about the length is also encoded in the input. For communication, we present the following remark.

*Remark* B.5. For any collection of unique hardmax patterns $\{\bar{A}^{(k)}\}_{k=1}^n$, where $\bar{A}^{(k)}$ is $k \times k$, there exists a configuration of node positional attention parameters in Equation (3) and a temperature parameter $T$ such that the resulting softmax patterns $\{A^{(k)}\}_{k=1}^n$ approximate each $\bar{A}^{(k)}$ to any desired level of accuracy. Formally, for any collection of unique hardmax patterns $\{\bar{A}^{(k)}\}_{k=1}^n$ and any $\varepsilon > 0$, there exists a temperature parameter $T = T(\varepsilon) > 0$ and corresponding attention parameters such that for all $i, j \in [n]$ and for all $k \in [n]$, the following inequality holds: $\left| \bar{A}_{i,j}^{(k)} - A_{i,j}^{(k)} \right| \le \varepsilon$.

The proof of this remark relies on a slight modification of the proof of Lemma B.3. However, to cover all possible patterns, the embedding dimension of positional encodings should also encode the input length and be of the order of $O(n^3)$. Although this embedding dimension is theoretically large, in practice, one does not need as many dimensions for positional encodings, as demonstrated in the variable length experiments in Section 7.

## B.5. Softmax patterns using positional attention

We conclude the discussion on expressivity by showing a final, standalone, result, namely that the positional attention architecture in Equation (3) can represent any softmax pattern. This result is further discussed in Appendix F regarding reasons for poor out-of-distribution performance by standard Transformers.

We begin by stating the definition of the row-wise softmax transformation for a matrix $X \in \mathbb{R}^{p \times q}$:

$$\text{softmax}(X)_{i,j} = \frac{\exp(X_{i,j})}{\sum_{k=1}^q \exp(X_{i,k})} \quad \text{for } i \in [p], j \in [q] \tag{8}$$

As with hardmax, the definition is implicitly extended to vectors in $\mathbb{R}^n$ by viewing them as $1 \times n$ matrices. The image of the softmax function is the set of row-stochastic matrices with entries in $(0, 1)$. Indeed, it is easy to see that when softmax is applied to a matrix, the resulting matrix satisfies the above property. On the other hand, for a matrix $\mathbf{B} = (b_{ij})_{i \in [p], j \in [q]}$ with $b_{ij} \in (0, 1)$ and $\sum_{j \in [q]} b_{ij} = 1$ for all $i \in [p]$ we have $\text{softmax}(\tilde{\mathbf{X}}) = B$ where $\tilde{\mathbf{X}}_{i,j} = \ln(b_{ij})$. We use the term *softmax pattern* to refer to any matrix in the image of softmax.

Consider attention weights $A^{(\ell,h)}$ that are defined by positional encodings in Equation (3). Let $B \in (0, 1)^{n \times n}$ be a softmax pattern. We would like to find parameters $W_Q^{(\ell,h)}$ and $W_K^{(\ell,h)}$ that induce $B$, that is $A^{(\ell,h)} = B$. From the properties of softmax described above, it suffices to solve the matrix equation $(PW_Q^{(\ell,h)}) \cdot (PW_K^{(\ell,h)})^\top = \tilde{B}$ where $\tilde{B}_{ij} = \ln(B_{ij})$. This equation always has a solution when $d_P = n$ and $P$ is invertible. We summarize the above observation in the following expressivity remark:

*Remark* B.6 (Positional attention is expressive). Positional attention can realize all softmax patterns at every layer provided that $d_P = n$ and $P$ is invertible. This is not necessarily true in the case of standard attention where, in subsequent layers,

positional encodings are modified and, therefore, not guaranteed to be linearly independent.

# C. Proof of Generalization Bound (Theorem 6.1)

This section provides a complete proof of Theorem 6.1. The high level strategy which inductively builds a cover for the class of multi-layer architectures is similar to that of Edelman et al. (2022). We made a couple of changes to account for positional attention. For the reader's convenience we re-state all relevant definitions below.

## C.1. Preliminaries

### C.1.1. NOTATIONS AND THE NETWORK ARCHITECTURE

For a matrix $Z$ we denote by $Z_{i,:}$ and $Z_{:,i}$ the $i$-th row and column of $Z$, respectively. $\|\cdot\|_2$ denotes the spectral norm for matrices, and $\|\cdot\|_{p,q}$ denotes the $(p,q)$ matrix norm where the $p$-norm is over columns and $q$-norm over rows. For vectors, $\|\cdot\|_p$ denotes the $\ell_p$ norm. For example, $\|Z\|_{2,\infty} := \max_j \|Z_{:,j}\|_2$ and $\|Z\|_{2,1} = \sum_j \|Z_{:,j}\|_2$.

We are interested in the positional transformer architecture of $L$ layers, which is obtained by composing Equation (1) $L$ times. In addition, we parameterize $\Phi^{(\ell)}$ to be a 2-layer MLP for each $\ell$. Given input $X \in \mathbb{R}^{n \times d_X}$ and positional encodings $P \in \mathbb{R}^{n \times d_P}$, the resulting architecture has $L$ layers, each layer has $H$ attention heads, and it is written as

$$F^{(0)}(X; W^{1:0}) := X,$$
$$\forall \ell = 1, 2, \ldots, L:$$
$$F^{(\ell)}(X; W^{1:\ell}) := \sigma\left(\left(\left(\bigoplus_{h=1}^{H} A^{(\ell,h)} F^{(\ell-1)}(X; W^{1:\ell-1}) W_V^{(\ell,h)}\right) \oplus F^{(\ell-1)}(X; W^{1:\ell-1})\right) W_1^{(\ell)}\right) W_2^{(\ell)}, \quad (9)$$
$$\text{with} \quad A^{(\ell,h)} := \text{softmax}(P W_Q^{(\ell,h)} W_K^{(\ell,h)^\top} P^\top), \quad \forall h = 1, 2, \ldots, H.$$

In the above, $\sigma$ is an $L_\sigma$-Lipschitz activation function with $\sigma(0) = 0$. We write

$$W^{(\ell)} := \{W_Q^{(\ell,1)}, W_Q^{(\ell,2)}, \ldots, W_Q^{(\ell,H)}, W_K^{(\ell,1)}, W_K^{(\ell,2)}, \ldots, W_K^{(\ell,H)}, W_V^{(\ell,1)}, W_V^{(\ell,2)}, \ldots, W_V^{(\ell,H)}, W_1^{(\ell)}, W_2^{(\ell)}\}$$

for the set of weight matrices at layer $\ell$, and

$$W^{1:\ell} := (W^{(1)}, W^{(2)}, \ldots, W^{(\ell)})$$

for the set of weight matrices up to layer $\ell$. Note that we may assume without loss of generality that the output projection matrix $W_O^{(\ell)}$ is absorbed into the weight matrix $W_1^{(\ell)}$ of the MLP, and therefore we omit $W_O^{(\ell)}$ in Equation (9). Denote $d = \max\{d_X, d_P, d_V, d_{\text{out}}\}$. Since we are to derive an upper bound which increases with $d_X, d_P, d_V, d_{\text{out}}$, we sometimes replace $d_X, d_P, d_V, d_{\text{out}}$ with their maximum $d$ in the derivations to simplify the notation.

Given the final output $X_{\text{out}} = F^{(L)}(X; W^{1:L}) \in \mathbb{R}^{n \times n}$, we extract a scalar output from the architecture as $w^\top X_{\text{out}}[n]$ with trainable weights $w \in \mathbb{R}^d$. This defines a class of scalar-output architectures as in Edelman et al. (2022). The index $i \in [n]$ on which we apply the linear function $x \mapsto w^\top x$ can be arbitrary, and here we simply set to the last index, $n$. Our generalization bound applies to the following class of scalar-output architectures with bounded-norm parameters:

$$\mathcal{F} := \left\{ X \mapsto w^\top \left(F^{(L)}(X; W^{1:L})[n]\right) : \|w\|_2 \leq B_w, \right.$$
$$\left\|W_V^{(\ell,h)}\right\|_2, \left\|W_1^{(\ell)}\right\|_2, \left\|W_2^{(\ell)}\right\|_2 \leq B_2, \forall \ell \in [L], \forall h \in [H], \quad (10)$$
$$\left.\left\|W_K^{(\ell,h)^\top} W_Q^{(\ell,h)}\right\|_{2,1}, \left\|W_V^{(\ell,h)}\right\|_{2,1}, \left\|W_1^{(\ell)}\right\|_{2,1}, \left\|W_2^{(\ell)}\right\|_{2,1} \leq B_{2,1}, \forall \ell \in [L], \forall h \in [H] \right\}.$$

C.1.2. GENERALIZATION BOUND, RADEMACHER COMPLEXITY, AND THE COVERING NUMBER

**Definition** (Risk). Let $\mathcal{D}$ be a distribution over $\mathcal{X} \times \mathbb{R}$, and let $\ell : \mathbb{R} \times \mathbb{R} \to \mathbb{R}$ be the loss function. For a given class of functions $\mathcal{F} = \{f : \mathcal{X} \to \mathbb{R}\}$ and $f \in \mathcal{F}$, the generalization error, or the risk, and the empirical risk, are defined as

$$\mathsf{risk}(f; \mathcal{D}) := \mathbb{E}_{(x,y) \sim \mathcal{D}} \left[ \ell(f(x), y) \right], \quad \widehat{\mathsf{risk}}(f; (x(i), y^{(i)})_{i=1}^m) := \frac{1}{m} \sum_{i=1}^m \ell(f(x^{(i)}), y^{(i)}).$$

**Definition C.1** (Empirical Rademacher complexity). For a given class of functions $\mathcal{F} = \{f : \mathcal{X} \to \mathbb{R}\}$ and $\{x^{(i)}\}_{i=1}^m \subset \mathcal{X}$, the empirical Rademacher complexity $\widehat{\mathcal{R}}(\mathcal{F}; \{x^{(i)}\}_{i=1}^m)$ is defined as

$$\widehat{\mathcal{R}}(\mathcal{F}; \{x^{(i)}\}_{i=1}^m) = \frac{1}{m} \mathbb{E}_\varepsilon \left[ \sup_{f \in \mathcal{F}} \sum_{i=1}^m \varepsilon_i f(x^{(i)}) \right],$$

where $\varepsilon$ is a vector of $m$ i.i.d. Rademacher random variables, that is, $\mathbb{P}(\varepsilon_i = 1) = \mathbb{P}(\varepsilon_i = -1) = 1/2$.

The following theorem bounds the generalization gap $|\mathsf{risk} - \widehat{\mathsf{risk}}|$ using the Rademacher complexity of a function class.

**Theorem C.2** (Bartlett & Mendelson (2003)). *Let $\mathcal{D}$ be a distribution over $\mathcal{X} \times \mathbb{R}$ and let $\ell : \mathbb{R} \times \mathbb{R} \to \mathbb{R}$ be a b-bounded loss function that is L-Lipschitz in its first argument. Let $\mathcal{F} = \{f : \mathcal{X} \to \mathbb{R}\}$ be a given class of functions. Then for any $\delta > 0$, with probability at least $1 - \delta$, simutaneously for all $f \in \mathcal{F}$, it holds that*

$$\left| \mathsf{risk}(f; \mathcal{D}) - \widehat{\mathsf{risk}}(f; (x(i), y^{(i)})_{i=1}^m) \right| \leq 4L\widehat{\mathcal{R}}(\mathcal{F}; \{x^{(i)}\}_{i=1}^m) + 2b\sqrt{\frac{\log(1/\delta)}{2m}}.$$

We may bound the Rademacher complexity of a function class $\mathcal{F}$ using its covering number.

**Definition C.3** (Covering number). For a given class of (scalar-valued or vector-valued) functions $\mathcal{F}$, the covering number $\mathcal{N}_\infty(\mathcal{F}; \varepsilon; \{x^{(i)}\}_{i=1}^m; \|\cdot\|)$ is the smallest size of a collection (a cover) $\mathcal{C} \subset \mathcal{F}$ such that for all $f \in \mathcal{F}$ there is $\hat{f} \in \mathcal{C}$ satisfying

$$\max_{i \in [m]} \left\| f(x^{(i)} - \hat{f}(x^{(i)}) \right\| \leq \varepsilon.$$

**Lemma C.4** (Rademacher complexity via covering number; Corollary A.3 in Edelman et al. (2022)). *For a class of functions $\mathcal{F} = \{f : \mathcal{X} \to \mathbb{R}\}$ such that $|f| \leq A$ for all $f \in \mathcal{F}$, suppose that $\log \mathcal{N}_\infty(\mathcal{F}; \varepsilon; \{x^{(i)}\}_{i=1}^m) \leq C_\mathcal{F}/\varepsilon^2$, then for some consstant $c > 0$:*

$$\widehat{\mathcal{R}}(\mathcal{F}; \{x^{(i)}\}_{i=1}^m) \leq c \cdot \sqrt{\frac{C_\mathcal{F}}{m}} \cdot \left( 1 + \log \left( A\sqrt{m/C_\mathcal{F}} \right) \right).$$

Combining Theorem C.2 and Lemma C.4 we may bound the generalization gap $|\mathsf{risk} - \widehat{\mathsf{risk}}|$ using the covering number of a function class, as stated in the following lemma.

**Lemma C.5.** *For a class of functions $\mathcal{F} = \{f : \mathcal{X} \to \mathbb{R}\}$ such that $f$ is bounded for all $f \in \mathcal{F}$ and that $\log \mathcal{N}_\infty(\mathcal{F}; \varepsilon; \{x^{(i)}\}_{i=1}^m) \leq C_\mathcal{F}/\varepsilon^2$ for all $\{x^{(i)}\}_{i=1}^m \subset \mathcal{X}$. Then for any $\delta > 0$, with probability at least $1 - \delta$, simultaneously for all $f \in \mathcal{F}$, it holds that*

$$\left| \mathsf{risk}(f; \mathcal{D}) - \widehat{\mathsf{risk}}(f; (x(i), y^{(i)})_{i=1}^m) \right| \leq \tilde{O} \left( \sqrt{\frac{C_\mathcal{F}}{m}} + \sqrt{\frac{\log(1/\delta)}{m}} \right).$$

.

C.1.3. SOME USEFUL LEMMAS

**Lemma C.6** (Corollary A.7 in Edelman et al. (2022)). *For vectors $x, y \in \mathbb{R}^n$ we have that*

$$\|\mathrm{softmax}(x) - \mathrm{softmax}(y)\|_1 \leq 2 \|x - y\|_\infty.$$

**Lemma C.7** (Lemma 4.6 in Edelman et al. (2022)). *Let $\mathcal{W} := \{W \in \mathbb{R}^{d \times d'} : \|W^\top\|_{2,1} \leq B_W\}$, and consider the function class $\mathcal{F}_W := \{x \mapsto Wx : W \in \mathcal{W}\}$. For any $\varepsilon > 0$ and $x^{(1)}, x^{(2)}, \dots, x^{(m)} \in \mathbb{R}^d$ satisfying $\|x^{(i)}\|_2 \leq B_X$ for all $i \in [m]$, we have*

$$\log \mathcal{N}_\infty(\mathcal{F}_W; \varepsilon; x^{(1)}, \dots, x^{(m)}; \|\cdot\|_2) \lesssim \frac{B_W^2 B_X^2}{\varepsilon^2} \log(dm).$$

**Lemma C.8** (Lemma A.8 in Edelman et al. (2022))**.** *For $a_1, b_i \geq 0$, the solution to the following optimization problem*

$$\min_{\varepsilon_1,\ldots,\varepsilon_t} \quad \sum_{i=1}^{t} \frac{a_i}{\varepsilon_i^2}$$

$$\text{s.t} \quad \sum_{i=1}^{t} b_i \varepsilon_i = \varepsilon$$

*is $\nu^3/\varepsilon^2$ and is achieved at $\varepsilon_i = \varepsilon(a_1/b_i)^{1/3}/\nu$ where $\nu := \sum_{i=1}^{t} a_i^{1/3} b_i^{2/3}$.*

### C.2. Lipschitz Property of the Architecture

Denote

$$f(Z; \{W_Q^{(\cdot,h)}\}_{h=1}^{H}, \{W_K^{(\cdot,h)}\}_{h=1}^{H}, \{W_V^{(\cdot,h)}\}_{h=1}^{H}) := \left(\bigoplus_{h=1}^{H} A^{(\cdot,h)} Z W_V^{(\cdot,h)}\right) \oplus Z$$

$$= \left[A^{(\cdot,1)} Z W_V^{(\cdot,1)}, A^{(\cdot,2)} Z W_V^{(\cdot,2)}, \cdots, A^{(\cdot,H)} Z W_V^{(\cdot,H)}, Z\right],$$

where $A^{(\cdot,h)} = \text{softmax}(P W_Q^{(\cdot,h)} W_K^{(\cdot,h)^\top} P^\top)$.

**Lemma C.9.** *For any $Z, \widehat{Z}, \{W_Q^{(\cdot,h)}\}_{h=1}^{H}, \{\widehat{W}_Q^{(\cdot,h)}\}_{h=1}^{H}, \{W_K^{(\cdot,h)}\}_{h=1}^{H}, \{\widehat{W}_K^{(\cdot,h)}\}_{h=1}^{H}, \{W_V^{(\cdot,h)}\}_{h=1}^{H}, \{\widehat{W}_V^{(\cdot,h)}\}_{h=1}^{H}$,*

$$\left\|\left(f\left((Z; \{W_Q^{(\cdot,h)}\}_{h=1}^{H}, \{W_K^{(\cdot,h)}\}_{h=1}^{H}, \{W_V^{(\cdot,h)}\}_{h=1}^{H}\right) - f\left(\widehat{Z}; \{\widehat{W}_Q^{(\cdot,h)}\}_{h=1}^{H}, \{\widehat{W}_K^{(\cdot,h)}\}_{h=1}^{H}, \{\widehat{W}_V^{(\cdot,h)}\}_{h=1}^{H}\right)\right)^\top\right\|_{2,\infty}$$

$$\leq \max\left(\max_{h\in[H]} \|W_V^{(\cdot,h)}\|_2, 1\right)$$

$$\cdot \left(\|Z^\top - \widehat{Z}^\top\|_{2,\infty} + 2\|Z^\top\|_{2,\infty} \|P^\top\|_{2,\infty} \max_{h\in[H]} \left\|\left(W_Q^{(\cdot,h)} W_K^{(\cdot,h)^\top} - \widehat{W}_Q^{(\cdot,h)} \widehat{W}_K^{(\cdot,h)^\top}\right) P^\top\right\|_{2,\infty}\right)$$

$$+ \sqrt{H} \max_{h\in[H]} \left\|\left(W_V^{(\cdot,h)} - \widehat{W}_V^{(\cdot,h)}\right)^\top \widehat{Z}^\top\right\|_{2,\infty}$$

*Proof.* Denote $A^{(\cdot,h)} = \text{softmax}(P W_Q^{(\cdot,h)} W_K^{(\cdot,h)^\top} P^\top)$ and $\widehat{A}^{(\cdot,h)} = \text{softmax}(P \widehat{W}_Q^{(\cdot,h)} \widehat{W}_K^{(\cdot,h)^\top} P^\top)$:

$$\left\|\left(f\left((Z; \{W_Q^{(\cdot,h)}\}_{h=1}^{H}, \{W_K^{(\cdot,h)}\}_{h=1}^{H}, \{W_V^{(\cdot,h)}\}_{h=1}^{H}\right) - f\left(\widehat{Z}; \{\widehat{W}_Q^{(\cdot,h)}\}_{h=1}^{H}, \{\widehat{W}_K^{(\cdot,h)}\}_{h=1}^{H}, \{\widehat{W}_V^{(\cdot,h)}\}_{h=1}^{H}\right)\right)^\top\right\|_{2,\infty}$$

$$= \left\|\begin{bmatrix} W_V^{(\cdot,1)^\top} (A^{(\cdot,1)} Z)^\top \\ \vdots \\ W_V^{(\cdot,H)^\top} (A^{(\cdot,H)} Z)^\top \\ Z^\top \end{bmatrix} - \begin{bmatrix} \widehat{W}_V^{(\cdot,1)^\top} (\widehat{A}^{(\cdot,1)} \widehat{Z})^\top \\ \vdots \\ \widehat{W}_V^{(\cdot,H)^\top} (\widehat{A}^{(\cdot,H)} \widehat{Z})^\top \\ \widehat{Z}^\top \end{bmatrix}\right\|_{2,\infty}$$

$$\leq \left\|\begin{bmatrix} W_V^{(\cdot,1)^\top} \left(A^{(\cdot,1)} Z - \widehat{A}^{(\cdot,1)} \widehat{Z}\right)^\top \\ \vdots \\ W_V^{(\cdot,H)^\top} \left(A^{(\cdot,H)} Z - \widehat{A}^{(\cdot,H)} \widehat{Z}\right)^\top \\ Z^\top - \widehat{Z}^\top \end{bmatrix}\right\|_{2,\infty} + \left\|\begin{bmatrix} \left(W_V^{(\cdot,1)} - \widehat{W}_V^{(\cdot,1)}\right)^\top (\widehat{A}^{(\cdot,1)} \widehat{Z})^\top \\ \vdots \\ \left(W_V^{(\cdot,H)} - \widehat{W}_V^{(\cdot,H)}\right)^\top (\widehat{A}^{(\cdot,H)} \widehat{Z})^\top \\ 0 \end{bmatrix}\right\|_{2,\infty}$$

$$= \left\|\begin{bmatrix} W_V^{(\cdot,1)^\top} & & \\ & \ddots & \\ & & W_V^{(\cdot,H)^\top} \\ & & I \end{bmatrix}\begin{bmatrix} (A^{(\cdot,1)} Z)^\top - \widehat{A}^{(\cdot,1)} \widehat{Z})^\top \\ \vdots \\ (A^{(\cdot,H)} Z)^\top - \widehat{A}^{(\cdot,H)} \widehat{Z})^\top \\ Z^\top - \widehat{Z}^\top \end{bmatrix}\right\|_{2,\infty} + \left\|\begin{bmatrix} \left(W_V^{(\cdot,1)} - \widehat{W}_V^{(\cdot,1)}\right)^\top (\widehat{A}^{(\cdot,1)} \widehat{Z})^\top \\ \vdots \\ \left(W_V^{(\cdot,H)} - \widehat{W}_V^{(\cdot,H)}\right)^\top (\widehat{A}^{(\cdot,H)} \widehat{Z})^\top \end{bmatrix}\right\|_{2,\infty}$$

$$\leq \left\| \begin{bmatrix} W_V^{(\cdot,1)\top} & & \\ & \ddots & \\ & & W_V^{(\cdot,H)\top} \\ & & & I \end{bmatrix} \right\|_2 \left\| \begin{bmatrix} (A^{(\cdot,1)}Z)^\top - \widehat{A}^{(\cdot,1)}\widehat{Z})^\top \\ \vdots \\ (A^{(\cdot,H)}Z)^\top - \widehat{A}^{(\cdot,H)}\widehat{Z})^\top \\ Z^\top - \widehat{Z}^\top \end{bmatrix} \right\|_{2,\infty} + \left\| \begin{bmatrix} \left(W_V^{(\cdot,1)} - \widehat{W}_V^{(\cdot,1)}\right)^\top (\widehat{A}^{(\cdot,1)}\widehat{Z})^\top \\ \vdots \\ \left(W_V^{(\cdot,H)} - \widehat{W}_V^{(\cdot,H)}\right)^\top (\widehat{A}^{(\cdot,H)}\widehat{Z})^\top \end{bmatrix} \right\|_{2,\infty}$$

$$\leq \max\left(\max_{h\in[H]} \|W_V^{(\cdot,h)}\|_2, 1\right) \cdot \left\| \begin{bmatrix} (A^{(\cdot,1)}Z)^\top - \widehat{A}^{(\cdot,1)}\widehat{Z})^\top \\ \vdots \\ (A^{(\cdot,H)}Z)^\top - \widehat{A}^{(\cdot,H)}\widehat{Z})^\top \\ Z^\top - \widehat{Z}^\top \end{bmatrix} \right\|_{2,\infty} + \sqrt{H} \max_{h\in[H]} \left\| \left(W_V^{(\cdot,h)} - \widehat{W}_V^{(\cdot,h)}\right)^\top \left(\widehat{A}^{(\cdot,h)}\widehat{Z}\right)^\top \right\|_{2,\infty}$$

$$\leq \max\left(\max_{h\in[H]} \|W_V^{(\cdot,h)}\|_2, 1\right) \cdot \left( \left\| Z^\top \left[A^{(\cdot,1)\top} - \widehat{A}^{(\cdot,1)\top}, \dots, A^{(\cdot,H)\top} - \widehat{A}^{(\cdot,H)\top}\right] \right\|_{2,\infty} \right.$$
$$\left. + \left\| (Z-\widehat{Z})^\top \left[\widehat{A}^{(\cdot,1)\top}, \dots, \widehat{A}^{(\cdot,H)\top}, I\right] \right\|_{2,\infty} \right) + \sqrt{H} \max_{h\in[H]} \left\| \left(W_V^{(\cdot,h)} - \widehat{W}_V^{(\cdot,h)}\right)^\top \left(\widehat{A}^{(\cdot,h)}\widehat{Z}\right)^\top \right\|_{2,\infty}. \tag{11}$$

Using $\|Pv\|_2 \leq \|P\|_{2,\infty}\|v\|_1$ (and therefore $\|PQ\|_{2,\infty} \leq \|P\|_{2,\infty}\|Q\|_{1,\infty}$):

$$\left\| (Z-\widehat{Z})^\top \left[\widehat{A}^{(\cdot,1)\top}, \dots, \widehat{A}^{(\cdot,H)\top}, I\right] \right\|_{2,\infty} \leq \left\| (Z-\widehat{Z})^\top \right\|_{2,\infty} \left\| \left[\widehat{A}^{(\cdot,1)\top}, \dots, \widehat{A}^{(\cdot,H)\top}, I\right] \right\|_{1,\infty} = \left\| Z^\top - \widehat{Z}^\top \right\|_{2,\infty}, \tag{12}$$

and

$$\left\| \left(W_V^{(\cdot,h)} - \widehat{W}_V^{(\cdot,h)}\right)^\top \left(\widehat{A}^{(\cdot,h)}\widehat{Z}\right)^\top \right\|_{2,\infty} \leq \left\| \left(W_V^{(\cdot,h)} - \widehat{W}_V^{(\cdot,h)}\right)^\top \widehat{Z}^\top \right\|_{2,\infty} \left\| \widehat{A}^{(\cdot,h)\top} \right\|_{1,\infty}$$
$$= \left\| \left(W_V^{(\cdot,h)} - \widehat{W}_V^{(\cdot,h)}\right)^\top \widehat{Z}^\top \right\|_{2,\infty}. \tag{13}$$

Using $\|PQ\|_{2,\infty} \leq \|P\|_{2,\infty}\|Q\|_{1,\infty}$, $\|PQ\|_{\infty,\infty} = \max_{i,j}|P_{i,:}Q_{:,j}| \leq \max_i \|P_{i,:}^\top\|_2 \max_j \|Q_{:,j}\|_2 = \|P^\top\|_{2,\infty}\|Q\|_{2,\infty}$, and Lemma C.6,

$$\left\| Z^\top \left[A^{(\cdot,1)\top} - \widehat{A}^{(\cdot,1)\top}, \dots, A^{(\cdot,H)\top} - \widehat{A}^{(\cdot,H)\top}\right] \right\|_{2,\infty}$$
$$= \max_{h\in[H]} \left\| Z^\top \left(A^{(\cdot,h)\top} - \widehat{A}^{(\cdot,h)\top}\right) \right\|_{2,\infty}$$
$$\leq \max_{h\in[H]} \|Z^\top\|_{2,\infty} \left\| A^{(\cdot,h)\top} - \widehat{A}^{(\cdot,h)\top} \right\|_{1,\infty}$$
$$\leq \max_{h\in[H]} 2\|Z^\top\|_{2,\infty} \left\| A^{(\cdot,h)\top} - \widehat{A}^{(\cdot,h)\top} \right\|_{\infty,\infty}$$
$$= \max_{h\in[H]} 2\|Z^\top\|_{2,\infty} \left\| P\left(W_K^{(\cdot,h)}W_Q^{(\cdot,h)\top} - \widehat{W}_K^{(\cdot,h)}\widehat{W}_Q^{(\cdot,h)\top}\right)P^\top \right\|_{\infty,\infty}$$
$$\leq \max_{h\in[H]} 2\|Z^\top\|_{2,\infty}\|P^\top\|_{2,\infty} \left\| \left(W_Q^{(\cdot,h)}W_K^{(\cdot,h)\top} - \widehat{W}_Q^{(\cdot,h)}\widehat{W}_K^{(\cdot,h)\top}\right)P^\top \right\|_{2,\infty} \tag{14}$$

Combining Equations (11) to (14) gives the result. $\qquad\square$

We introduce one more term to simplify the notation. For $\ell = 1, 2, \dots, L$ denote

$$g^{(\ell)}(X; W^{1:\ell}) = \left( \bigoplus_{h=1}^{H} A^{(\ell,h)} F^{(\ell-1)}(X; W^{1:\ell-1}) W_V^{(\ell,h)} \right) \oplus F^{(\ell-1)}(X; W^{1:\ell-1}) \tag{15}$$

where $F^{(\ell-1)}(X; W^{1:\ell-1})$ and $A^{(\ell,h)}$ are the same as in Equation (9). That is, $g^{(\ell)}(X; W^{1:\ell})$ is the input to the MLP in the $\ell$-th layer of the positional transformer architecture.

**Lemma C.10.** *For any $W^{1:\ell}$ and $\widehat{W}^{1:\ell}$,*

$$\left\| \left( F^{(\ell)}(X; W^{1:\ell}) - F^{(\ell)}(X; \widehat{W}^{1:\ell}) \right)^\top \right\|_{2,\infty}$$

$$\leq \left\| \left( W_2^{(\ell)} - \widehat{W}_2^{(\ell)} \right)^\top \sigma \left( g^{(\ell)}(X; \widehat{W}^{1:\ell}) \widehat{W}_1^{(\ell)} \right)^\top \right\|_{2,\infty}$$

$$+ L_\sigma \| W_2^{(\ell)} \|_2 \left\| \left( W_1^{(\ell)} - \widehat{W}_1^{(\ell)} \right)^\top g^{(\ell)}(X; \widehat{W}^{1:\ell})^\top \right\|_{2,\infty}$$

$$+ L_\sigma \| W_1^{(\ell)} \|_2 \| W_2^{(\ell)} \|_2 \max \left( \max_{h \in [H]} \| W_V^{(\ell,h)} \|_2, 1 \right) \left\| \left( F^{(\ell-1)}(X; W^{1:\ell-1}) - F^{(\ell-1)}(X; \widehat{W}^{1:\ell-1}) \right)^\top \right\|_{2,\infty}$$

$$+ 2 L_\sigma \| W_1^{(\ell)} \|_2 \| W_2^{(\ell)} \|_2 \| F^{(\ell-1)}(X; W^{1:\ell-1})^\top \|_{2,\infty} \| P^\top \|_{2,\infty} \max \left( \max_{h \in [H]} \| W_V^{(\ell,h)} \|_2, 1 \right)$$

$$\cdot \max_{h \in [H]} \left\| \left( W_Q^{(\ell,h)} W_K^{(\ell,h)\top} - \widehat{W}_Q^{(\ell,h)} \widehat{W}_K^{(\ell,h)\top} \right) P^\top \right\|_{2,\infty}$$

$$+ L_\sigma \| W_1^{(\ell)} \|_2 \| W_2^{(\ell)} \|_2 \sqrt{H} \max_{h \in [H]} \left\| \left( W_V^{(\ell,h)} - \widehat{W}_V^{(\ell,h)} \right)^\top F^{(\ell-1)}(X; W^{1:\ell-1})^\top \right\|_{2,\infty}$$

*Proof.* Using the definition of $F^{(\ell)}(X; W^{1:\ell})$ in Equation (9) and the definition of $g^{(\ell)}(X; W^{1:\ell})$ in Equation (15):

$$\left\| \left( F^{(\ell)}(X; W^{1:\ell}) - F^{(\ell)}(X; \widehat{W}^{1:\ell}) \right)^\top \right\|_{2,\infty}$$

$$= \left\| W_2^{(\ell)\top} \sigma \left( g^{(\ell)}(X; W^{1:\ell}) W_1^{(\ell)} \right)^\top - \widehat{W}_2^{(\ell)\top} \sigma \left( g^{(\ell)}(X; \widehat{W}^{1:\ell}) \widehat{W}_1^{(\ell)} \right)^\top \right\|_{2,\infty}$$

(Using triangle inequality):

$$\leq \left\| \left( W_2^{(\ell)} - \widehat{W}_2^{(\ell)} \right)^\top \sigma \left( g^{(\ell)}(X; \widehat{W}^{1:\ell}) \widehat{W}_1^{(\ell)} \right)^\top \right\|_{2,\infty} \tag{16}$$

$$+ \left\| W_2^{(\ell)\top} \left( \sigma \left( g^{(\ell)}(X; W^{1:\ell}) W_1^{(\ell)} \right) - \sigma \left( g^{(\ell)}(X; \widehat{W}^{1:\ell}) \widehat{W}_1^{(\ell)} \right) \right)^\top \right\|_{2,\infty}. \tag{17}$$

The term in (16) gives the first part of the required bound. We focus on the term in (17) below and show that it is bounded by the second to the last parts of the required bound.

$$\left\| W_2^{(\ell)\top} \left( \sigma \left( g^{(\ell)}(X; W^{1:\ell}) W_1^{(\ell)} \right) - \sigma \left( g^{(\ell)}(X; \widehat{W}^{1:\ell}) \widehat{W}_1^{(\ell)} \right) \right)^\top \right\|_{2,\infty}$$

(Using $\| PQ \|_{2,\infty} \leq \| P \|_2 \| Q \|_{2,\infty}$ and $\| P \|_2 = \| P^\top \|_2$):

$$\leq \| W_2^{(\ell)} \|_2 \left\| \left( \sigma \left( g^{(\ell)}(X; W^{1:\ell}) W_1^{(\ell)} \right) - \sigma \left( g^{(\ell)}(X; \widehat{W}^{1:\ell}) \widehat{W}_1^{(\ell)} \right) \right)^\top \right\|_{2,\infty}$$

(Using Lipschitzness of $\sigma$ and $\sigma(0) = 0$):

$$\leq L_\sigma \| W_2^{(\ell)} \|_2 \left\| \left( g^{(\ell)}(X; W^{1:\ell}) W_1^{(\ell)} - g^{(\ell)}(X; \widehat{W}^{1:\ell}) \widehat{W}_1^{(\ell)} \right)^\top \right\|_{2,\infty}$$

(Using triangle inequality):

$$\leq L_\sigma \| W_2^{(\ell)} \|_2 \left\| \left( W_1^{(\ell)} - \widehat{W}_1^{(\ell)} \right)^\top g^{(\ell)}(X; \widehat{W}^{1:\ell})^\top \right\|_{2,\infty}$$

$$+ L_\sigma \| W_2^{(\ell)} \|_2 \left\| W_1^{(\ell)\top} \left( g^{(\ell)}(X; W^{1:\ell}) - g^{(\ell)}(X; \widehat{W}^{1:\ell}) \right)^\top \right\|_{2,\infty}$$

(Using $\| PQ \|_{2,\infty} \leq \| P \|_2 \| Q \|_{2,\infty}$ and $\| P \|_2 = \| P^\top \|_2$):

$$\leq \; L_\sigma \|W_2^{(\ell)}\|_2 \left\| \left( W_1^{(\ell)} - \widehat{W}_1^{(\ell)} \right)^\top g^{(\ell)}(X; \widehat{W}^{1:\ell})^\top \right\|_{2,\infty}$$

$$+ \; L_\sigma \|W_1^{(\ell)}\|_2 \|W_2^{(\ell)}\|_2 \left\| \left( g^{(\ell)}(X; W^{1:\ell}) - g^{(\ell)}(X; \widehat{W}^{1:\ell}) \right)^\top \right\|_{2,\infty}$$

(Using Lemma C.9 and definition of $g^{(\ell)}(X; W^{1:\ell})$):

$$\leq \; L_\sigma \|W_2^{(\ell)}\|_2 \left\| \left( W_1^{(\ell)} - \widehat{W}_1^{(\ell)} \right)^\top g^{(\ell)}(X; \widehat{W}^{1:\ell})^\top \right\|_{2,\infty}$$

$$+ \; L_\sigma \|W_1^{(\ell)}\|_2 \|W_2^{(\ell)}\|_2 \max \big( \max_{h \in [H]} \|W_V^{(\ell,h)}\|_2, 1 \big) \cdot \left( \left\| \left( F^{(\ell-1)}(X; W^{1:\ell-1}) - F^{(\ell-1)}(X; \widehat{W}^{1:\ell-1}) \right)^\top \right\|_{2,\infty} \right.$$

$$+ \; 2 \|F^{(\ell-1)}(X; W^{1:\ell-1})^\top\|_{2,\infty} \|P^\top\|_{2,\infty} \max_{h \in [H]} \left\| \left( W_Q^{(\ell,h)} W_K^{(\ell,h)^\top} - \widehat{W}_Q^{(\ell,h)} \widehat{W}_K^{(\ell,h)^\top} \right) P^\top \right\|_{2,\infty} \Bigg)$$

$$+ \; L_\sigma \|W_1^{(\ell)}\|_2 \|W_2^{(\ell)}\|_2 \sqrt{H} \max_{h \in [H]} \left\| \left( W_V^{(\ell,h)} - \widehat{W}_V^{(\ell,h)} \right)^\top F^{(\ell-1)}(X; W^{1:\ell-1})^\top \right\|_{2,\infty}.$$

This gives the required result. $\qquad\square$

### C.3. Covering Number Upper Bound

Let $X^{(1)}, X^{(2)}, \ldots, X^{(m)} \in \mathbb{R}^{n \times d}$ be a collection of inputs. Let $\mathcal{F}$ be the class of functions $\mathbb{R}^{n \times d} \to \mathbb{R}$ as defined in Equation (10). Given $\epsilon > 0$, let $\mathcal{C} \subset \mathcal{F}$ be such that, for all $f \in \mathcal{F}$ there is $\hat{f} \in \mathcal{C}$ such that

$$\max_{i \in [m]} |f(X^{(i)}) - \hat{f}(X^{(i)})| \leq \epsilon.$$

Our goal is to bound the size of the cover set $\mathcal{C}$. To do this we will inductively construct $\mathcal{C}$. Let us start with a simple bound on the norm of the output.

For $W^{1:1}$ satisfying the norm bounds in Equation (10), using $\|Pv\|_2 \leq \|P\|_2 \|v\|_2$ repeatedly, and the fact that $\sigma$ is $L_\sigma$-Lipschitz with $\sigma(0) = 0$,

$$\left\| F^{(1)}(X; W^{1:1})^\top \right\|_{2,\infty} \leq L_\sigma \|W_2^{(1)}\|_2 \|W_1^{(1)}\|_2 \sqrt{1 + \sum_{h \in H} \|W_V^{(1,h)}\|_2^2} \|X^\top\|_{2,\infty} \leq L_\sigma B_2^2 \sqrt{1 + H B_2^2} \|X^\top\|_{2,\infty},$$

and inductively for each $\ell \in [L]$ and $W^{1:\ell}$ satisfying the norm bounds in Equation (10) we get

$$\left\| F^{(\ell)}(X; W^{1:\ell})^\top \right\|_{2,\infty} \leq B^{(\ell)} \|X^\top\|_{2,\infty}, \text{ where } B^{(\ell)} := L_\sigma^\ell B_2^{2\ell} (1 + H B_2^2)^{\ell/2}.$$

Let us now build a cover $\mathcal{C}^{(1)}$ for the first layer of the architecture. For any $X^{(1)}, X^{(2)}, \ldots, X^{(m)} \in \mathbb{R}^{n \times d}$ satisfying $\|X^{(i)^\top}\|_{2,\infty} \leq B_X$ for all $i$, and $W, \widehat{W} \in \mathcal{W} = \{W \in \mathbb{R}^{d \times d'} : \|W^\top\|_{2,1} \leq B_W\}$, because

$$\max_{i \in [m]} \left\| (W - \widehat{W})^\top X^{(i)^\top} \right\|_{2,\infty} = \max_{i \in [m], j \in [n]} \left\| (W - \widehat{W})^\top X^{(i)^\top}_{j,:} \right\|_2,$$

we may apply Lemma C.7 and get that there is an $\varepsilon_W$-cover $\widehat{\mathcal{W}} \subset \mathbb{R}^{d \times d'}$ such that

$$\log |\widehat{\mathcal{W}}| \lesssim \frac{B_W^2 B_X^2}{\varepsilon_W^2} \log(dmn), \quad \text{and} \quad \max_{W \in \mathcal{W}} \min_{\widehat{W} \in \widehat{\mathcal{W}}} \max_{i \in [m]} \left\| (W - \widehat{W})^\top X^{(i)^\top} \right\|_{2,\infty} \leq \varepsilon_W.$$

Therefore, given inputs $X^{(1)}, X^{(2)}, \ldots, X^{(m)} \in \mathbb{R}^{n \times d}$ such that $\|X^{(i)^\top}\|_{2,\infty} \leq B_X$ for all $i$, using Lemma C.7 we get for each $h \in [H]$ there is an $\varepsilon_V^{(1)}$-cover $\mathcal{C}_V^{(1,h)} \subset \mathbb{R}^{d \times d}$ with

$$\log \left| \mathcal{C}_V^{(1,h)} \right| \lesssim \frac{B_{2,1}^2 B_X^2}{\varepsilon_V^{(1)^2}} \log(dmn)$$

such that for all $h \in [H]$ and $W_V^{(1,h)}$ satisfying $\|W_V^{(1,h)}\|_{2,1} \leq B_{2,1}$ there is $\widehat{W}_V^{(1,h)} \in \mathcal{C}_V^{(1,h)}$ such that

$$\max_{i \in [m]} \left\| \left( W_V^{(1,h)} - \widehat{W}_V^{(1,h)} \right)^\top X^{(i)\top} \right\|_{2,\infty} \leq \varepsilon_V^{(1)}.$$

It follows that there is an $\varepsilon_V^{(1)}$-cover $\mathcal{C}_V^{(1)} \subset \bigotimes_{h=1}^H \mathbb{R}^{d \times d}$ with

$$|\mathcal{C}_V^{(1)}| \leq \prod_{h=1}^H |\mathcal{C}_V^{(1,h)}| \quad \implies \quad \log\left|\mathcal{C}_V^{(1)}\right| \lesssim \frac{H B_{2,1}^2 B_X^2}{\varepsilon_V^{(1)2}} \log(dmn)$$

such that for any $W_V^{(1,1)}, W_V^{(1,2)}, \ldots, W_V^{(1,H)}$ satisfying $\|W_V^{(1,h)}\|_{2,1} \leq B_{2,1}$ for all $h$, there is $(\widehat{W}_V^{(1,1)}, \ldots, \widehat{W}_V^{(1,H)}) \in \mathcal{C}_V^{(1)}$ such that

$$\max_{h \in [H]} \max_{i \in [m]} \left\| \left( W_V^{(1,h)} - \widehat{W}_V^{(1,h)} \right)^\top X^{(i)\top} \right\|_{2,\infty} \leq \varepsilon_V^{(1)}.$$

Similarly, there is an $\varepsilon_{QK}^{(1)}$-cover $\mathcal{C}_{QK}^{(1)} \subset \prod_{h=1}^H \mathbb{R}^{d_P \times d_P}$ with

$$\log\left|\mathcal{C}_{QK}^{(1)}\right| \lesssim \frac{H B_{2,1}^2 B_P^2}{\varepsilon_{QK}^{(1)2}} \log(d_P n)$$

such that for any $W_Q^{(1,1)} W_K^{(1,1)\top}, W_Q^{(1,2)} W_K^{(1,2)\top}, \ldots, W_Q^{(1,H)} W_K^{(1,H)\top}$ satisfying $\|W_Q^{(1,h)} W_K^{(1,h)}\|_{2,1} \leq B_{2,1}$ for all $h$, there is $(\widehat{W}_Q^{(1,1)} \widehat{W}_K^{(1,1)\top}, \ldots, \widehat{W}_Q^{(1,H)} \widehat{W}_K^{(1,H)\top}) \in \mathcal{C}_{QK}^{(1)}$ such that

$$\max_{h \in [H]} \left\| \left( W_Q^{(1,h)} W_K^{(1,h)\top} - \widehat{W}_Q^{(1,h)} \widehat{W}_K^{(1,h)\top} \right)^\top P^\top \right\|_{2,\infty} \leq \varepsilon_{QK}^{(1)}.$$

Given $(\widehat{W}_Q^{(1,1)} \widehat{W}_K^{(1,1)\top}, \ldots, \widehat{W}_Q^{(1,H)} \widehat{W}_K^{(1,H)\top}) \in \mathcal{C}_{QK}^{(1)}$ and $(\widehat{W}_V^{(1,1)}, \ldots, \widehat{W}_V^{(1,H)}) \in \mathcal{C}_V^{(1)}$, we have that $\|g^{(1)}(X; \widehat{W}^{1:1})\|_{2,\infty} \leq B^{(1)} B_X$. Therefore, using Lemma C.7 there is an $\varepsilon_{W_1}^{(1)}$-cover $\mathcal{C}_{W_1}^{(1)} \subset \mathbb{R}^{d(H+1) \times d}$ with

$$\log\left|\mathcal{C}_{W_1}^{(1)}\right| \lesssim \frac{B^{(1)2} B_{2,1}^2 B_X^2}{\varepsilon_{W_1}^{(1)2}} \log(dHmn)$$

such that for any $W_1^{(1)}$ satisfying $\|W_1^{(1)}\|_{2,1} \leq B_{2,1}$ there is $\widehat{W}_1^{(1)} \in \mathcal{C}_{W_1}^{(1)}$ such that

$$\left\| \left( W_1^{(1)} - \widehat{W}_1^{(1)} \right)^\top g^{(1)}(X; \widehat{W}^{1:1})^\top \right\|_{2,\infty} \leq \varepsilon_{W_1}^{(1)}.$$

Similarly, given $(\widehat{W}_Q^{(1,1)} \widehat{W}_K^{(1,1)\top}, \ldots, \widehat{W}_Q^{(1,H)} \widehat{W}_K^{(1,H)\top}) \in \mathcal{C}_{QK}^{(1)}$, $(\widehat{W}_V^{(1,1)}, \ldots, \widehat{W}_V^{(1,H)}) \in \mathcal{C}_V^{(1)}$, and $\widehat{W}_1^{(1)} \in \mathcal{C}_{W_1}^{(1)}$, there is an $\varepsilon_{W_2}^{(1)}$-cover $\mathcal{C}_{W_2}^{(1)} \in \mathbb{R}^{d \times d}$ with

$$\log\left|\mathcal{C}_{W_2}^{(1)}\right| \lesssim \frac{B^{(1)2} B_{2,1}^2 B_X^2}{\varepsilon_{W_2}^{(1)2}} \log(dmn)$$

such that for any $W_2^{(1)}$ satisfying $\|W_2^{(1)}\|_{2,1} \leq B_{2,1}$ there is $\widehat{W}_2^{(1)} \in \mathcal{C}_{W_2}^{(1)}$ such that

$$\left\| \left( W_2^{(1)} - \widehat{W}_2^{(1)} \right)^\top \sigma\left( g^{(1)}(X; \widehat{W}^{1:1}) \widehat{W}_1^{(1)} \right)^\top \right\|_{2,\infty} \leq \varepsilon_{W_2}^{(1)}.$$

Therefore, using Lemma C.10, an $\varepsilon^{(1)}$-cover $\mathcal{C}^{(1)}$ for the first layer of the architecture, where

$$\varepsilon^{(1)} = \varepsilon_{W_2}^{(1)} + L_\sigma B_2 \varepsilon_{W_1}^{(1)} + 2 L_\sigma B_2^3 B_P \varepsilon_{QK}^{(1)} + \sqrt{H} L_\sigma B_2^2 \varepsilon_V^{(1)},$$

has size

$$|\mathcal{C}^{(1)}| \leq |\mathcal{C}_V^{(1)}| \cdot |\mathcal{C}_{QK}^{(1)}| \cdot \max \left\{ \left| \mathcal{C}_{W_1}^{(1)} \left( \widehat{W}_V^{(1,h)}, \widehat{W}_Q^{(1,h)} \widehat{W}_K^{(1,h)\top}, \forall h \right) \right| : \left\{ \widehat{W}_V^{(1,h)} \right\}_{h=1}^H \in \mathcal{C}_V^{(1)}, \left\{ \widehat{W}_Q^{(1,h)} \widehat{W}_K^{(1,h)\top} \right\}_{h=1}^H \in \mathcal{C}_{QK}^{(1)} \right\}$$

$$\cdot \max \left\{ \left| \mathcal{C}_{W_2}^{(1)} \left( \widehat{W}_V^{(1,h)}, \widehat{W}_Q^{(1,h)} \widehat{W}_K^{(1,h)\top}, \forall h; \widehat{W}_1^{(1)} \right) \right| : \left\{ \widehat{W}_V^{(1,h)} \right\}_{h=1}^H \in \mathcal{C}_V^{(1)}, \left\{ \widehat{W}_Q^{(1,h)} \widehat{W}_K^{(1,h)\top} \right\}_{h=1}^H \in \mathcal{C}_{QK}^{(1)}, \widehat{W}_1^{(1)} \in \mathcal{C}_{W_1}^{(1)} \right\}.$$

This means that

$$\log |\mathcal{C}^{(1)}| \lesssim \left( \frac{H B_{2,1}^2 B_X^2}{\varepsilon_V^{(1)2}} + \frac{H B_{2,1}^2 B_P^2}{\varepsilon_{QK}^{(1)2}} + \frac{B^{(1)2} B_{2,1}^2 B_X^2}{\varepsilon_{W_1}^{(1)2}} + \frac{B^{(1)2} B_{2,1}^2 B_X^2}{\varepsilon_{W_2}^{(1)2}} \right) \log(H(d + d_P)mn).$$

We now inductively construct a cover for deeper layers as follows. For each element $\widehat{W}^{1:\ell} \in \mathcal{C}^{1:\ell}$ where $\mathcal{C}^{1:\ell}$ denotes a cover for the architecture up to (and including) layer $\ell$ on inputs $X^{(1)}, \ldots, X^{(m)}$, we construct an $\varepsilon'_{\ell+1}$-cover $\mathcal{C}^{(\ell+1)}(\widehat{W}^{1:\ell})$ on inputs $\left\{ F^{(\ell)}(X^{(i)}; \widehat{W}^{1:\ell}) \right\}_{i \in [m]}$ for the $(\ell+1)$-th layer by following the same steps as above, where

$$\varepsilon'_{\ell+1} = \varepsilon_{W_2}^{(\ell+1)} + L_\sigma B_2 \varepsilon_{W_1}^{(\ell+1)} + 2L_\sigma B^{(\ell)} B_2^3 B_P \varepsilon_{QK}^{(\ell+1)} + \sqrt{H} L_\sigma B_2^2 \varepsilon_V^{(\ell+1)}.$$

It follows that from Lemma C.10 that

$$\mathcal{C}^{1:\ell+1} := \left\{ (\widehat{W}^{1:\ell}, \widehat{W}^{(\ell)}) : \widehat{W}^{1:\ell} \in \mathcal{C}^{1:\ell}, \widehat{W}^{(\ell)} \in \mathcal{C}^{(\ell+1)}(\widehat{W}^{1:\ell}) \right\}$$

is an $\varepsilon^{(\ell+1)}$-cover of the architecture up to (and including) layer $\ell + 1$, where

$$\varepsilon^{(\ell+1)} = L_\sigma B_2^3 \varepsilon^{(\ell)} + \varepsilon_{W_2}^{(\ell+1)} + L_\sigma B_2 \varepsilon_{W_1}^{(\ell+1)} + 2L_\sigma B^{(\ell)} B_2^3 B_P \varepsilon_{QK}^{(\ell+1)} + \sqrt{H} L_\sigma B_2^2 \varepsilon_V^{(\ell+1)},$$

and

$$|\mathcal{C}^{1:\ell+1}| \leq |\mathcal{C}^{1:\ell}| \cdot \max_{\widehat{W}^{1:\ell} \in \mathcal{C}^{1:\ell}} |\mathcal{C}^{(\ell+1)}(\widehat{W}^{1:\ell})|.$$

By induction, we obtain an $\varepsilon^{(L)}$-cover $\mathcal{C}^{1:L}$ of the function $F^{(L)}(X; W^{1:L})$ on inputs $X^{(1)}, \ldots, X^{(m)}$ with

$$\varepsilon^{(L)} = \sum_{\ell=1}^L \alpha^{(\ell)} \left( \varepsilon_{W_2}^{(\ell)} + \beta^{(\ell)} \varepsilon_{W_1}^{(\ell)} + \gamma^{(\ell)} \varepsilon_{QK}^{(\ell)} + \eta^{(\ell)} \varepsilon_V^{(\ell)} \right)$$

where

$$\alpha^{(\ell)} = L_\sigma^{\ell-1} B_2^{3\ell-3}, \quad \beta^{(\ell)} = L_\sigma B_2, \quad \gamma^{(\ell)} = 2L_\sigma B^{(\ell-1)} B_2^3 B_P, \quad \eta^{(\ell)} = \sqrt{H} L_\sigma B_2^2.$$

The size of the cover satisfies

$$\log |\mathcal{C}^{1:L}| \lesssim \sum_{\ell=1}^L \left( \frac{H B_{2,1}^2 B_X^2}{\varepsilon_V^{(\ell)2}} + \frac{H B^{(\ell-1)2} B_{2,1}^2 B_P^2}{\varepsilon_{QK}^{(\ell)2}} + \frac{B^{(\ell)2} B_{2,1}^2 B_X^2}{\varepsilon_{W_1}^{(\ell)2}} + \frac{B^{(\ell)2} B_{2,1}^2 B_X^2}{\varepsilon_{W_2}^{(\ell)2}} \right) \log(H(d + d_P)mn).$$

To build the final cover $\mathcal{C}$ for the class of scalar-output architectures $\mathcal{F}$, note that for any $W^{1:L}, \widehat{W}^{1:L}, w, \widehat{w}$,

$$\left| w^\top \left( F^{(L)}(X; W^{1:L})[n] \right) - \widehat{w}^\top \left( F^{(L)}(X; \widehat{W}^{1:L})[n] \right) \right|$$

$$\leq \left| w^\top \left( F^{(L)}(X; W^{1:L})[n] - F^{(L)}(X; \widehat{W}^{1:L})[n] \right) \right| + \left| (w - \widehat{w})^\top \left( F^{(L)}(X; \widehat{W}^{1:L})[n] \right) \right|.$$

Therefore, we may apply Lemma C.7 and build $\mathcal{C}$ inductively for every element in $\mathcal{C}^{1:L}$, and get that $\mathcal{C}$ is an $\varepsilon$-cover for $\mathcal{F}$ with

$$\varepsilon = \varepsilon_w + B_w \sum_{\ell=1}^L \alpha^{(\ell)} \left( \varepsilon_{W_2}^{(\ell)} + \beta^{(\ell)} \varepsilon_{W_1}^{(\ell)} + \gamma^{(\ell)} \varepsilon_{QK}^{(\ell)} + \eta^{(\ell)} \varepsilon_V^{(\ell)} \right) \tag{18}$$

and that the size of $\mathcal{C}$ is bounded as

$$\log |\mathcal{C}| \lesssim \frac{B^{(L)^2} B_X^2 B_w^2}{\varepsilon_w^2} \log m + \log \left|\mathcal{C}^{1:L}\right|. \tag{19}$$

To determine a good upper bound on $\log |\mathcal{C}|$, we pick $\varepsilon_V^{(\ell)}, \varepsilon_{QK}^{(\ell)}, \varepsilon_{W_1}^{(\ell)}, \varepsilon_{W_2}^{(\ell)}, \forall \ell \in [L]$, and $\varepsilon_w$ that minimize the right-hand size of Equation (19) while satisfying Equation (18). Invoking Lemma C.8, this gives

$$\log |\mathcal{C}| \lesssim \frac{1}{\varepsilon^2} \Bigg( \left(B^{(L)^2} B_X^2 B_w^2 \log m\right)^{1/3} + \left(B_w^2 B_X^2 B_{2,1}^2 \log(H(d+d_P)mn)\right)^{1/3} \cdot$$

$$\sum_{\ell=1}^{L} \left(H^{1/3} \alpha^{(\ell)2/3} \eta^{(\ell)2/3} + H^{1/3} B^{(\ell-1)2/3} B_P^{2/3} \alpha^{(\ell)2/3} \gamma^{(\ell)2/3}\right.$$

$$\left. + B^{(\ell)2/3} \alpha^{(\ell)2/3} \beta^{(\ell)2/3} + B^{(\ell)2/3} \alpha^{(\ell)2/3}\right) \Bigg)^3$$

$$\lesssim \frac{H B_w^2 B_X^2 B_P^2 B_{2,1}^2 B^{(L)^2} \alpha^{(L)^2} \left(\beta^{(L)^2} + \gamma^{(L)^2} + \eta^{(L)^2}\right)}{\varepsilon^2} \log(H(d+d_P)mn)$$

$$\lesssim \frac{L_\sigma^{2L} B_2^{3L} B^{(L)^2} H^2 B_w^2 B_{2,1}^2 B_X^2 B_P^4}{\varepsilon^2} \log(H(d+d_P)mn)$$

$$= (HL_\sigma B_2)^{O(L)} \frac{B_w^2 B_{2,1}^2 B_X^2 B_P^4}{\varepsilon^2} \log(H(d+d_P)mn). \tag{20}$$

Letting $\|w\|_2 \leq B_2$, $\|X^\top\|_{2,\infty} = \|P^\top\|_{2,\infty} = 1$ and using $d \geq d_P$ gives

$$\log |\mathcal{C}| \lesssim \frac{(HL_\sigma B_2)^{O(L)} B_{2,1}^2}{\varepsilon^2} \log(Hdmn). \tag{21}$$

Combining Equation (21) and Lemma C.5 proves Theorem 6.1.

**Remark.** Note that if we additionally apply layer normalization by projecting the output of each layer to the $\ell_2$ unit ball as in Edelman et al. (2022), such that

$$\|g^{(\ell)}(X; W^{1:\ell})^\top\|_{2,\infty}, \|F^{(\ell)}(X; W^{1:\ell})^\top\|_{2,\infty} \leq B^{(\ell)} = 1,$$

then the bound on $\log |\mathcal{C}|$ in Equation (21) becomes

$$\log |\mathcal{C}| \lesssim \frac{(L_\sigma B_2)^{O(L)} H^2 B_{2,1}^2}{\varepsilon^2} \log(Hdmn). \tag{22}$$

# D. Experiments

This section presents detailed results for the experiments reported in Section 7 as well as additional results not present in the main paper. Most reported results combine in-distribution (ID) and out-of-distribution (OOD) in one plot. This is shown by the scale factor on the x-axis, which indicates the level of out-of-distribution. A scale of one indicates in-distribution, whereas a scale greater than one shows an increasing level of out-of-distribution.

## D.1. Numeric tasks

### D.1.1. EXPERIMENTAL SETTING

All numeric tasks employ the same model configuration. The model uses the structure of Equation (1), augmented with linear encoding and decoding layers.

We compare the standard Transformer, which utilizes the attention mechanism in Equation (2), and the positional Transformer, which employs the attention defined in Equation (3). Both variants share the same number of layers and dimensional configurations, with any specific differences explicitly noted.

In all configurations, the total number of layers is set to $\lceil \log_2 n \rceil + 1$, where $n$ denotes the maximum input length, and each layer uses 2 attention heads. Along with each input sequence, we also append an empty scratchpad entry. This extra entry does not count toward the total number of layers and is not used to compute the loss. It is included solely to aid in the computation of the tasks. For the function $\Phi^{(\ell)}$, we employ a 2-layer MLP with ReLU activation functions. The embedding dimension of the encoder and the hidden dimensions of the MLP are both set to $64$.

We use one-hot encoded vectors of dimension $n$ for positional encodings, where the non-zero entry corresponds to the node position. Consequently, the embedding dimensions of $W_Q$ and $W_K$ are set to $n$. A key difference between the models is that standard Transformers concatenate positional encodings to the input, whereas positional Transformers supply positional information exclusively through the matrix $P$. Therefore, in positional Transformers, input values are solely encoded in the input, and positional information is exclusively encoded in the positional encoding matrix.

Both models are trained end-to-end using the squared loss between the predicted and target vectors of size $n$, with no intermediate supervision. We train models with Adam, starting with a learning rate of $5 \cdot 10^{-4}$ and a learning rate scheduler for a total of 2000 epochs.

Our training data consists of samples from the range $[-2, 2]$. To ensure diversity in the data, for each input sample, we first select lower and upper bounds $\gamma_l$ and $\gamma_u$ uniformly in $[-2, 2]$, and then for each of the $n$ elements of the input sample, we select its value uniformly from $[\gamma_l, \gamma_u]$.

### D.1.2. INPUT LENGTH VS. GENERALIZATION EXPERIMENTS

In this section, we validate that our results hold for multiple values of $n$. We train models for each fixed length $n \in \{2, 4, 8, 16, 32\}$ using 30,000 samples across all settings. The model depth varies with the input length $n$, with the number of layers set to $\lceil \log_2 n \rceil + 1$. We report the performance in-distribution and out-of-distribution using 1,000 test samples. As illustrated across Figures 9 to 13, positional Transformers exhibit comparable in-distribution loss to standard Transformers across various sequence lengths. Naturally, for a fixed number of samples, the in-distribution loss slightly increases as the sequence length grows, indicating the need for more samples. For out-of-distribution performance, we observe that the

positional Transformer maintains a more stable performance even with inputs of length $n = 32$. In contrast, the standard Transformer's performance rapidly degrades with the input length.

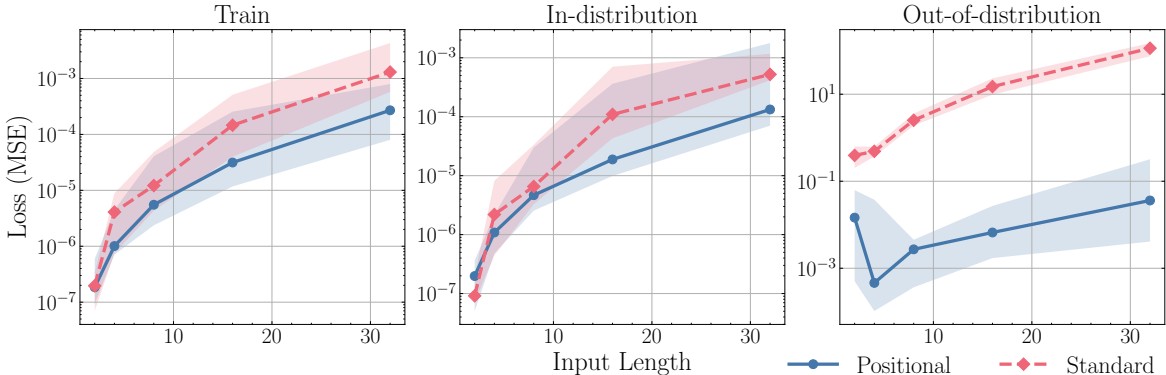

Figure 9: Training, in-distribution, and out-of-distribution test performance for the summing task is shown for standard Transformers (red) and positional Transformers (blue) across different input lengths. The x-axis indicates the fixed input length on which the model was trained. Models are trained on the range $[-2, 2]$ with 30,000 samples, validated on the same domain, and tested on an extended domain, $[-6, 6]$, each with 1,000 samples.

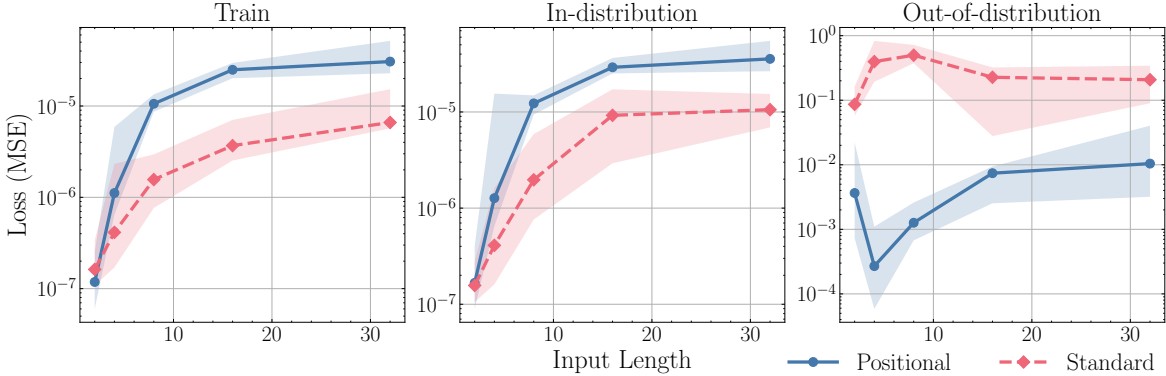

Figure 10: Training, in-distribution, and out-of-distribution test performance for the minimum task is shown for standard Transformers (red) and positional Transformers (blue) across different input lengths. The x-axis indicates the fixed input length on which the model was trained. Models are trained on the range $[-2, 2]$ with 30,000 samples, validated on the same domain, and tested on an extended domain, $[-6, 6]$, each with 1,000 samples.

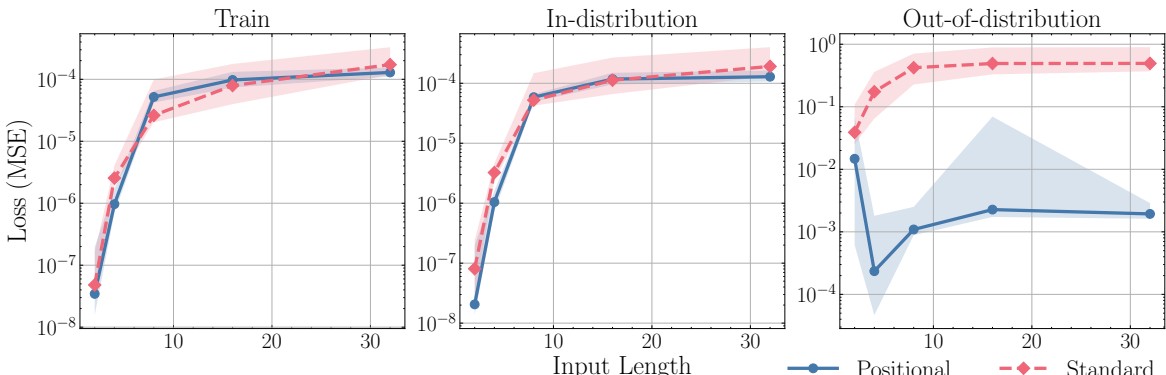

Figure 11: Training, in-distribution, and out-of-distribution test performance for the median task is shown for standard Transformers (red) and positional Transformers (blue) across different input lengths. The x-axis indicates the fixed input length on which the model was trained. Models are trained on the range $[-2, 2]$ with 30,000 samples, validated on the same domain, and tested on an extended domain, $[-6, 6]$, each with 1,000 samples.

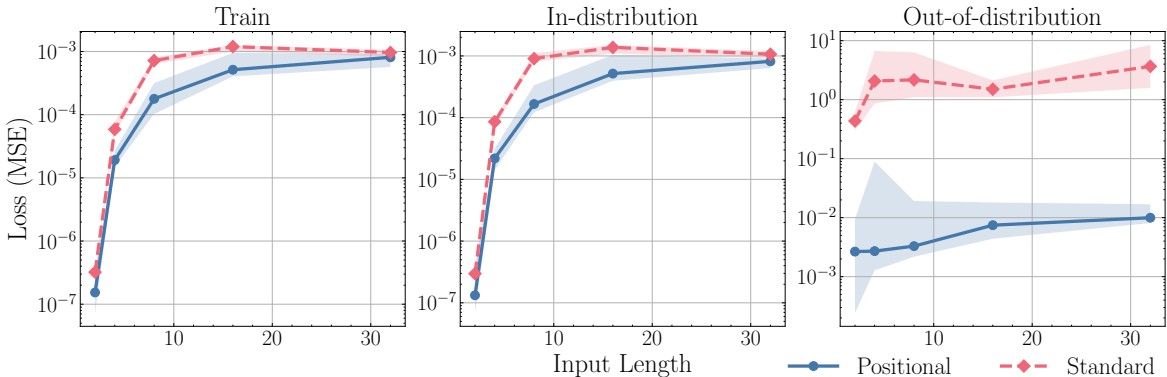

Figure 12: Training, in-distribution, and out-of-distribution test performance for the sorting task is shown for standard Transformers (red) and positional Transformers (blue) across different input lengths. The x-axis indicates the fixed input length on which the model was trained. Models are trained on the range $[-2, 2]$ with 30,000 samples, validated on the same domain, and tested on an extended domain, $[-6, 6]$, each with 1,000 samples.

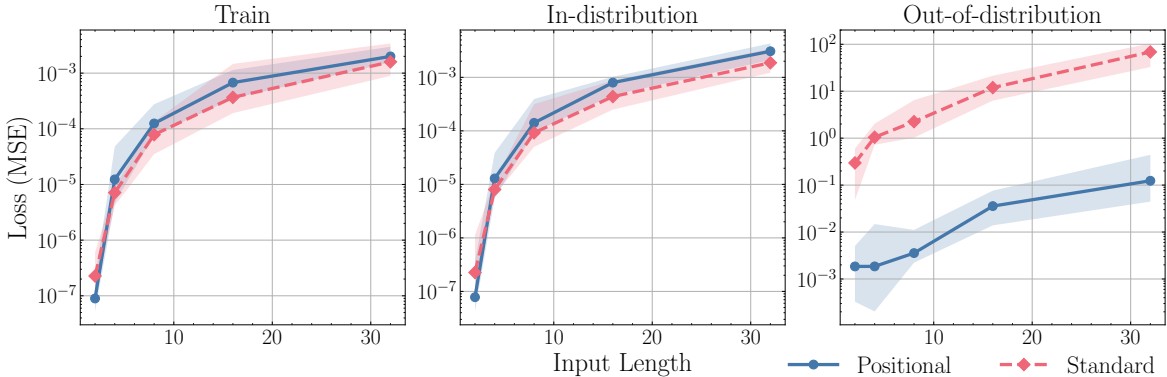

Figure 13: Training, in-distribution, and out-of-distribution test performance for the maximum sum subarray task is shown for standard Transformers (red) and positional Transformers (blue) across different input lengths. The x-axis indicates the fixed input length on which the model was trained. Models are trained on the range $[-2, 2]$ with 30,000 samples, validated on the same domain, and tested on an extended domain, $[-6, 6]$, each with 1,000 samples.

### D.1.3. SAMPLE SIZE VS. GENERALIZATION EXPERIMENTS

In this section, we provide detailed results showcasing the training, in-distribution, and out-of-distribution test performance for each of the five numerical tasks as a function of the number of training samples used. From the results, we can draw two conclusions about the behavior of the models as the number of samples increases. First, both models achieve good in-distribution performance. Second, only the positional Transformer achieves better OOD performance. The results for this experiment are presented in Figures 14 to 18.

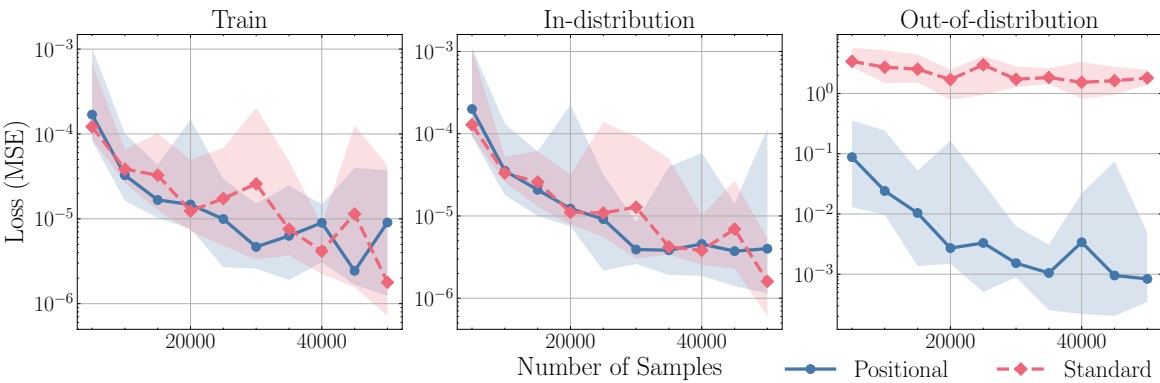

Figure 14: Training, in-distribution, and out-of-distribution test performance for the summing task is shown for standard Transformers (red) and positional Transformers (blue) as a function of the number of training samples (indicated on the x-axis). Models are trained on the range $[-2, 2]$ with varying training set sizes. In-distribution testing is performed on the same domain, and testing is conducted on an extended domain, $[-6, 6]$, each using 1,000 samples.

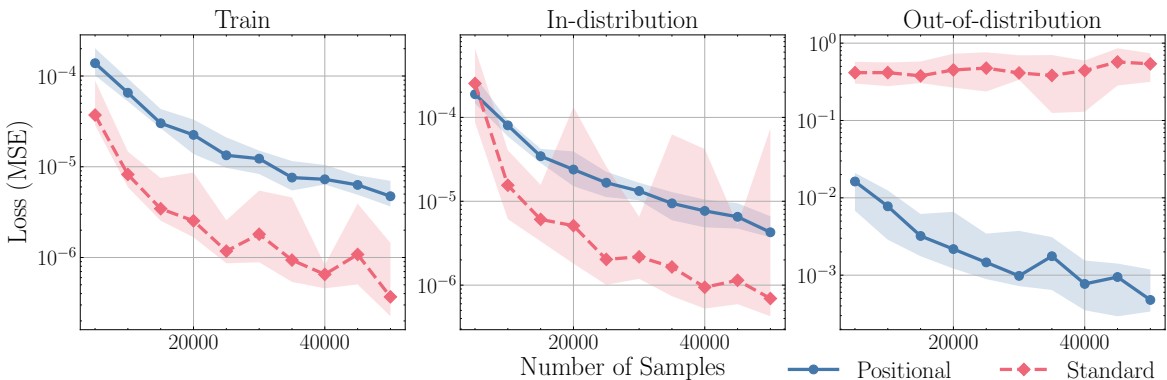

Figure 15: Training, in-distribution, and out-of-distribution test performance for the minimum task is shown for standard Transformers (red) and positional Transformers (blue) as a function of the number of training samples (indicated on the x-axis). Models are trained on the range $[-2, 2]$ with varying training set sizes. In-distribution testing is performed on the same domain, and testing is conducted on an extended domain, $[-6, 6]$, each using 1,000 samples.

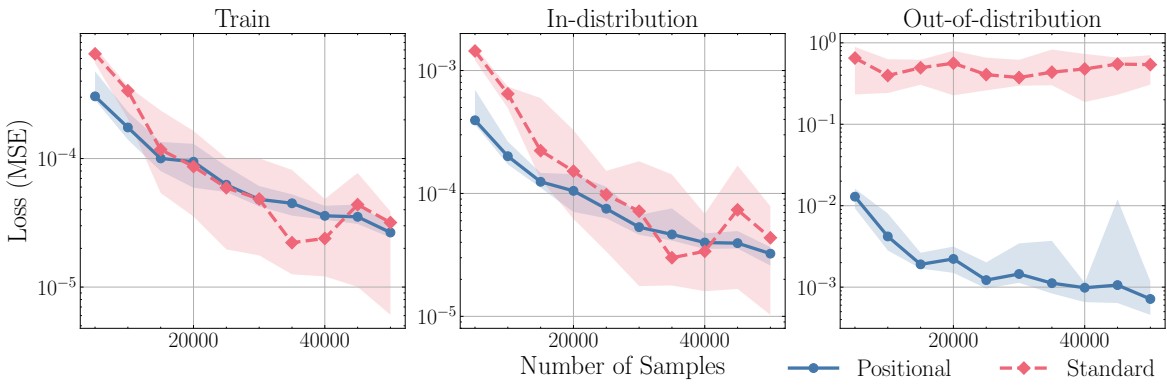

Figure 16: Training, in-distribution, and out-of-distribution test performance for the median task is shown for standard Transformers (red) and positional Transformers (blue) as a function of the number of training samples (indicated on the x-axis). Models are trained on the range $[-2, 2]$ with varying training set sizes. In-distribution testing is performed on the same domain, and testing is conducted on an extended domain, $[-6, 6]$, each using 1,000 samples.

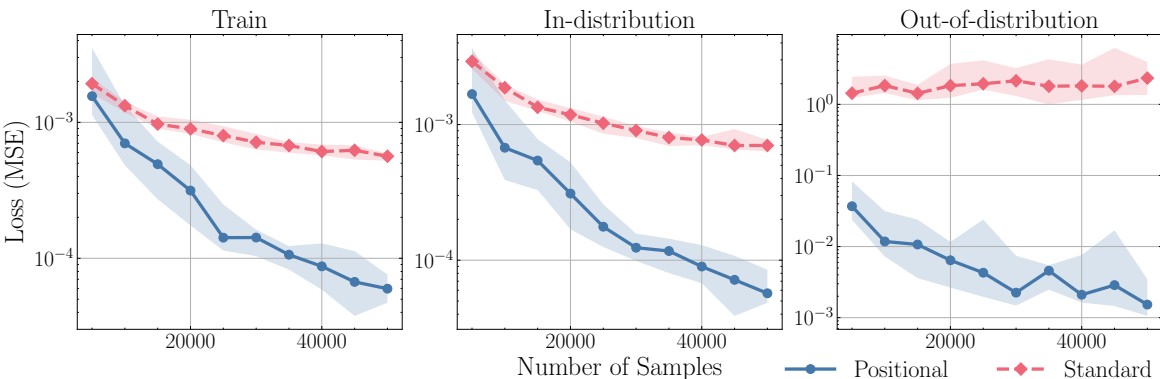

Figure 17: Training, in-distribution, and out-of-distribution test performance for the sorting task is shown for standard Transformers (red) and positional Transformers (blue) as a function of the number of training samples (indicated on the x-axis). Models are trained on the range $[-2, 2]$ with varying training set sizes. In-distribution testing is performed on the same domain, and out-of-distribution testing is conducted on an extended domain, $[-6, 6]$, each using 1,000 samples.

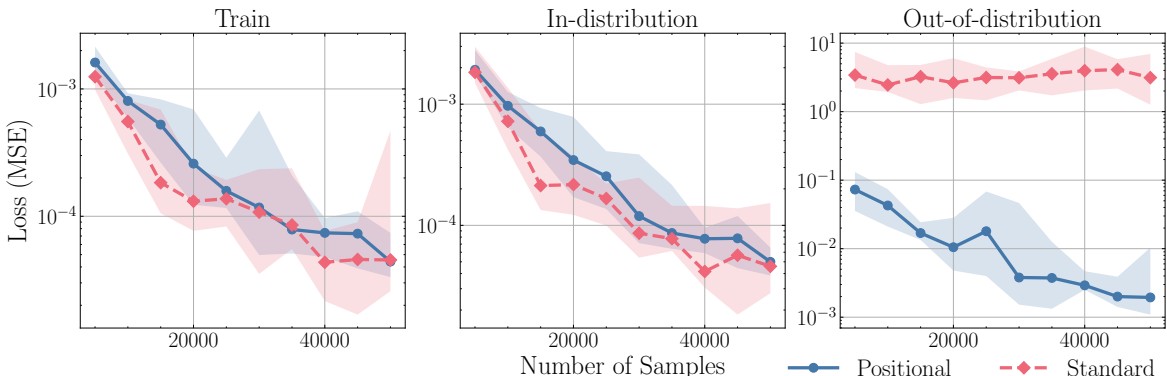

Figure 18: Training, in-distribution, and out-of-distribution test performance for the maximum sum subarray task is shown for standard Transformers (red) and positional Transformers (blue) as a function of the number of training samples (indicated on the x-axis). Models are trained on the range $[-2, 2]$ with varying training set sizes. In-distribution testing is performed on the same domain, and out-of-distribution testing is conducted on an extended domain, $[-6, 6]$, each using 1,000 samples.

### D.1.4. PERFORMANCE OVER VARIABLE INPUT LENGTHS

In this section, we present generalization results for models operating on variable-length inputs. This setting aims to verify the models' ability to generalize across different scales while maintaining the flexibility to handle inputs of varying lengths.

We evaluate the models' ability to process sequences of varying lengths up to a maximum size of $n = 8$ Specifically, the model is required to perform tasks on input sequences with lengths ranging from 1 to 8. Due to the more challenging setting, we train models with 500,000 samples and ensure that all input lengths are equally represented.

We then evaluate the in-distribution (ID) and out-of-distribution (OOD) loss across different scale factors $c \in \{1, 2, \ldots, 10\}$. Note that when $c = 1$, the setting actually corresponds to ID generalization. The losses reported are calculated using 3,000 samples for each scale. As shown in Figure 19, positional Transformers maintain the same in-distribution performance while consistently outperforming standard Transformers across all out-of-distribution scales and tasks. Additionally, our architecture maintains robust OOD performance even in tasks where the output can exceed the input magnitude (e.g., sum and maximum sum subarray).

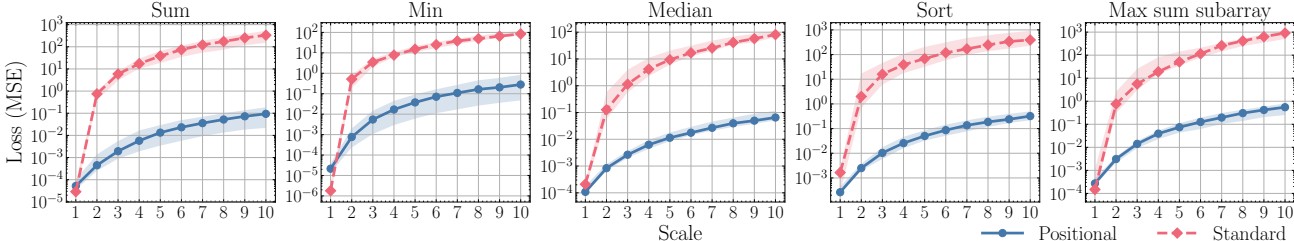

Figure 19: ID/OOD loss (measured as mean squared error, MSE) for standard Transformers (red) and positional Transformers (blue) across all five tasks for *variable* lengths (up to $n = 8$). The x-axis represents the ID/OOD scale factor. The solid line and shaded area denote the median and the region between the 10th and 90th percentiles over ten trials, respectively.

### D.1.5. PERFORMANCE OF COMPACT TRANSFORMERS

This section presents additional generalization results for simpler models, aiming to rule out potential overfitting caused by the excessive complexity of the standard Transformer. We examine two configuration variants: one with $\log n + 1$ layers (4 layers) and another with a single layer. The plots also illustrate the outcomes for different hidden dimensions in the MLP. We report the generalization results for the cumulative sum (Figure 20) and cumulative minimum (Figure 21) tasks. As observed, in both cases, the OOD performance deteriorates as the network size decreases. Although single-layer networks exhibit slightly better performance, they remain inferior to the performance of positional attention reported in the main paper.

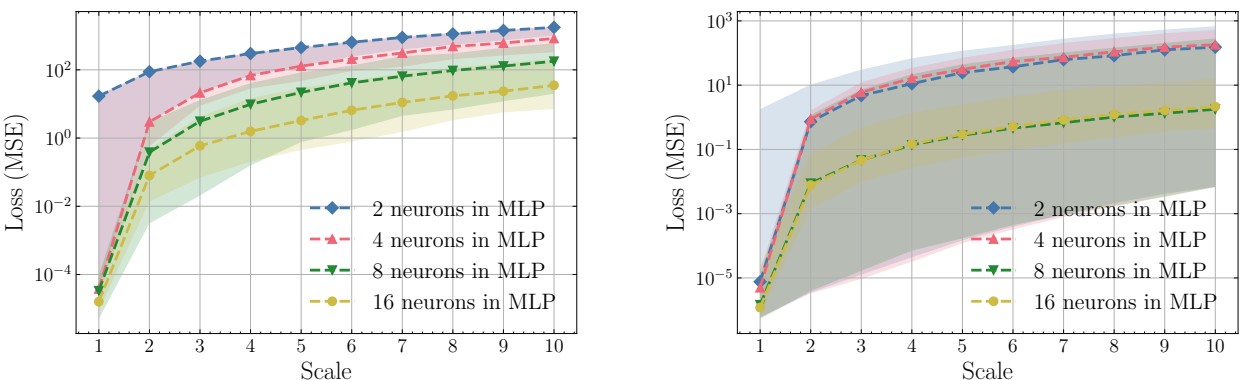

Figure 20: ID/OOD loss for various standard Transformer models on the cumulative sum task with fixed length ($n = 8$). The left plot displays results for models with 4 layers, while the right plot shows results for single-layer models, both featuring varying hidden dimensions in the MLPs. The x-axis represents the out-of-distribution scale factor, indicating the distance from the training distribution. The solid lines and shaded areas denote the median and the regions between the 10th and 90th percentiles across ten trials, respectively.

### D.1.6. PERFORMANCE WITH ALTERNATIVE POSITIONAL ENCODINGS

We compare the performance of positional Transformers and standard Transformers using alternative positional encodings, such as binary and sinusoidal encodings (Vaswani et al., 2017) and Rotary Positional Embedding (RoPE) (Su et al., 2024a), a widely adopted technique in natural language processing contexts that has also been applied to algorithmic tasks (Bounsi et al., 2024).

For binary positional encodings, we use $\lceil \log_2 n \rceil$ dimensions, where each entry represents the binary encoding of its index in $\lceil \log_2 n \rceil$ bits, with zeros encoded as $-1$. The result for binary encodings is shown in Figure 22. For sinusoidal positional encodings, we follow the strategy outlined in (Vaswani et al., 2017), with the encoding dimension set to $\lceil n/2 \rceil$. The result for sinusoidal encodings is shown in Figure 23. From an expressivity perspective, while these encodings are less expressive than one-hot positional encodings, they maintain consistent out-of-distribution (OOD) performance across all ranges. Furthermore, positional Transformers outperform standard Transformers in every task tested.

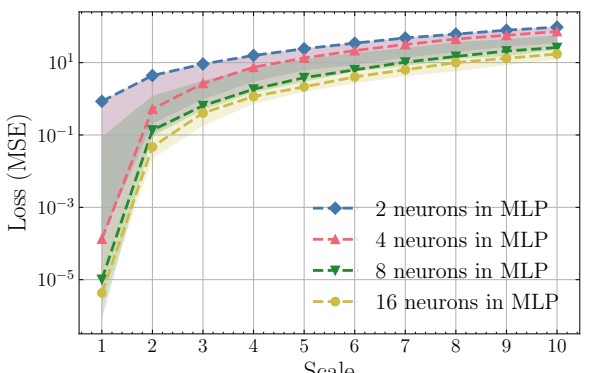 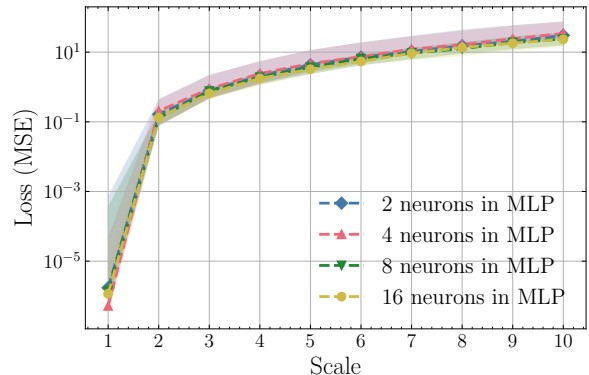

Figure 21: ID/OOD loss for various standard Transformer models on the cumulative minimum task with fixed length ($n = 8$). The left plot displays results for models with 4 layers, while the right plot shows results for single-layer models, both featuring varying hidden dimensions in the MLPs. The x-axis represents the out-of-distribution scale factor, indicating the distance from the training distribution. The solid lines and shaded areas denote the median and the regions between the $10^{\text{th}}$ and $90^{\text{th}}$ percentiles across ten trials, respectively.

As for RoPE, while it manages to decrease the OOD test loss, this improvement is far from what is observed in positional Transformers across every task. The results of this experiment are presented in Figure 24.

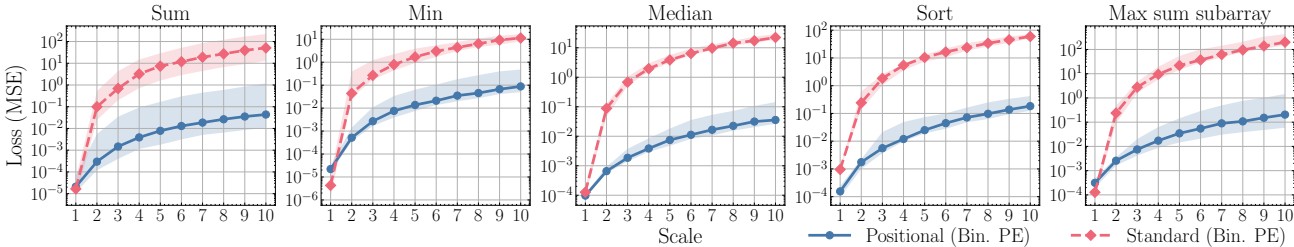

Figure 22: ID/OOD loss (measured as mean squared error, MSE) for standard Transformers (red) and positional Transformers (blue) using binary positional encodings across all five tasks for fixed length ($n = 8$). The x-axis represents the OOD scale factor, indicating the distance from the training distribution. The solid line and shaded area denote the median and the region between the $10^{\text{th}}$ and $90^{\text{th}}$ percentiles over ten trials, respectively.

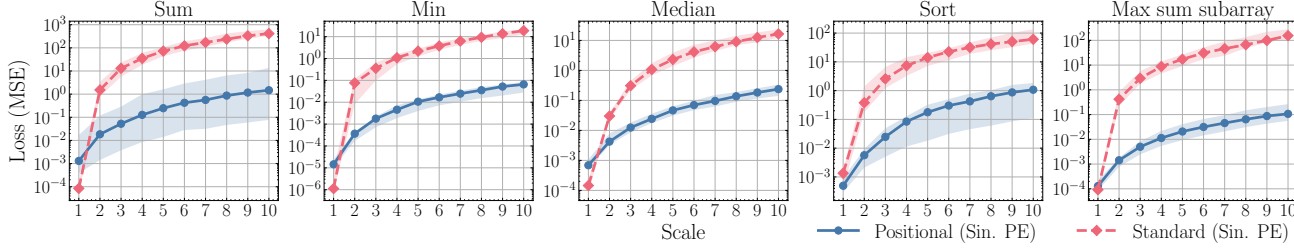

Figure 23: ID/OOD loss (measured as mean squared error, MSE) for standard Transformers (red) and positional Transformers (blue) using sinusoidal positional encodings across all five tasks for fixed length ($n = 8$). The x-axis represents the OOD scale factor, indicating the distance from the training distribution. The solid line and shaded area denote the median and the region between the $10^{\text{th}}$ and $90^{\text{th}}$ percentiles over ten trials, respectively.

Figure 24: ID/OOD loss (measured as mean squared error, MSE) for standard Transformers with RoPE (red) and positional Transformers (blue) across all five tasks for fixed length ($n = 8$). The x-axis represents the OOD scale factor, indicating the distance from the training distribution. The solid line and shaded area denote the median and the region between the $10^{th}$ and $90^{th}$ percentiles over ten trials, respectively.

### D.1.7. ABLATION STUDY ON ARCHITECTURAL DESIGN CHOICES

We carried out additional experiments to explore variations in the input data format and the placement of positional encodings. This leads to the following seven architectural choices (including two from the main paper and five additional variations):

1. Standard Transformers: Input numbers and positional encodings are fed to MLPs, value, query, and key matrices.

2. Standard Transformers with positional input (but no positional encodings): Input numbers are placed in one-hot positions, that is, the input is $\text{Diag}(X)$ where $X$ is the list of input numbers we give to other architectures. No additional positional encodings are used.

3. Positional Transformers: Input numbers are fed to the MLPs and value matrix; positional encodings are fed to query and key matrices.

4. Misaligned Positional Transformers: Input numbers are fed to the MLPs and value matrix; positional encodings are fed to the MLPs, value, query, and key matrices. That is, compared with Positional Transformers, we add positional encodings to the input.

5. Input-regularized Standard Transformers: Input numbers are fed to the MLPs, value, query, and key matrices; positional encodings are only used in query and key matrices.

6. No Positional Encodings: input numbers are fed to the MLPs, value, query, and key matrices. No positional encodings are used.

7. Using RoPE Only: Input numbers are fed to MLPs, value, query, and key matrices, removing absolute positional encodings, and using only RoPE in standard transformers.

We test the architectures for the cumulative sum task and the sorting task as representative tasks, for fixed input length $n = 8$. We keep the train/valid/test setup the same as before. The results are shown in Figure 25 and Figure 26. We note that our proposed positional Transformer architecture has all of the following three important factors that enabled OOD generalization: (1) use positional encodings, (2) do not use positional encodings in the value matrix of the attention, (3) use only fixed positional encodings in the key and query matrices. This design principle aligns with algorithms that are typically used to solve algorithmic tasks.

### D.1.8. EXPERIMENTS ON FINE-TUNING V PROJECTION MATRICES ONLY

Fine-tuning pre-trained models is very common in practice. In this case, one takes a model that has been trained on a different (and usually larger) dataset, and re-trains a small subset of layers on a new dataset, while keeping all other model parameters frozen. To further study the potential advantage of positional attention, we consider the following fine-tuning experiment. We take both positional Transformer and standard Transformer models that have been trained over in-distribution data (cf. Section 7.2 where $c = 1$, and Appendix D.1.1) for the numeric algorithmic tasks, fix all their model parameters except the weight parameters in the V projection matrices, and fine-tune the V projection matrices over OOD data with 10x scale factor

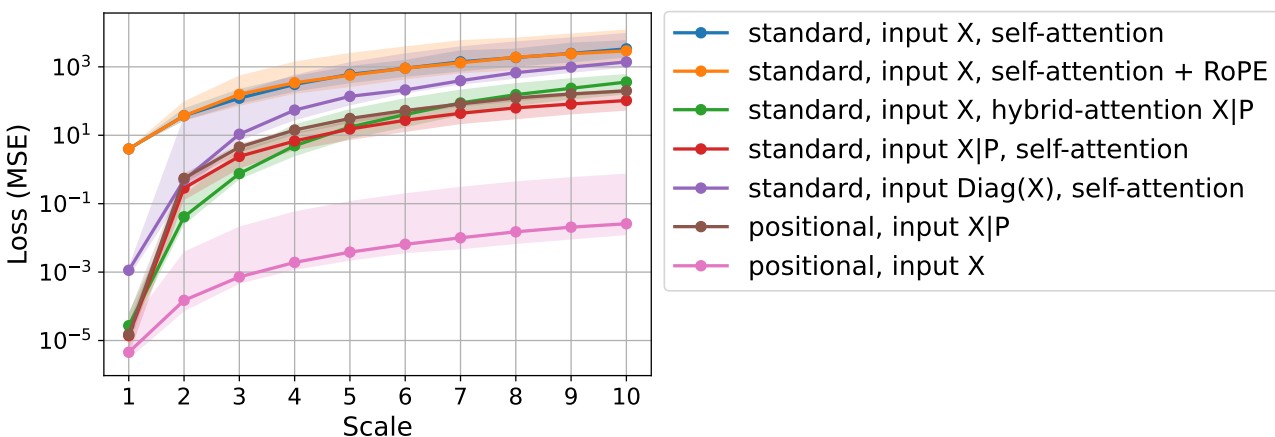

Figure 25: ID/OOD loss for the cumulative sum task under various architectural choices. We fix input length $n = 8$. The x-axis represents the OOD scale factor. The solid line and shaded area denote the median and the region between the $10^{th}$ and $90^{th}$ percentiles for 10 trials, respectively.

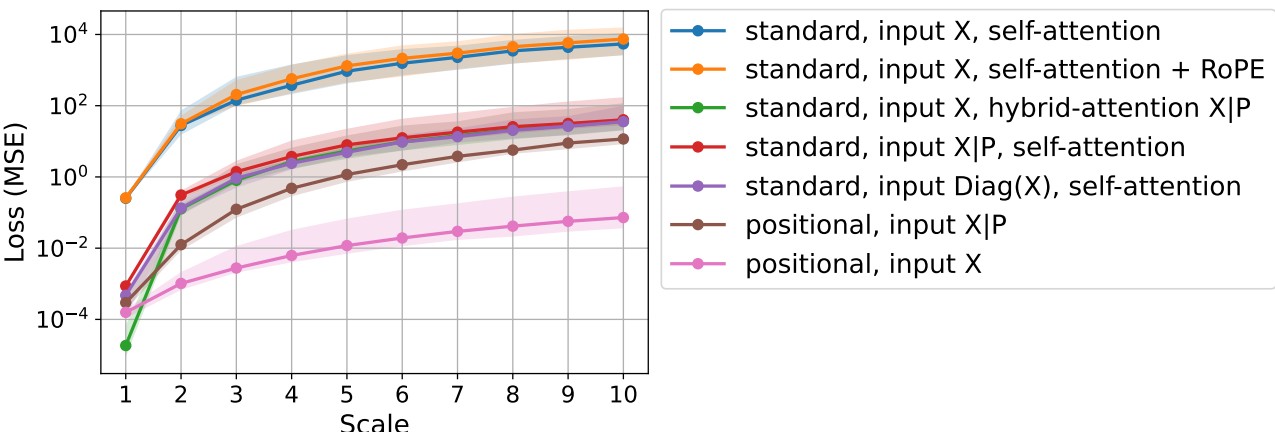

Figure 26: ID/OOD loss for the sorting task under various architectural choices. We fix input length $n = 8$. The x-axis represents the OOD scale factor. The solid line and shaded area denote the median and the region between the $10^{th}$ and $90^{th}$ percentiles for 10 trials, respectively.

(cf. Section 7.2 where $c = 10$). We consider three algorithmic tasks: cumulative sum, cumulative minimum, and sorting. For these tasks, we find that fine-tuning only the V projection matrices in the OOD case allows the positional Transformer to achieve the same level of generalization as in the in-distribution case, while, on the other hand, the same does not hold for the standard Transformer.

Table 1 reports the median test error (MSE) for both positional and standard transformer models when (i) they are trained and tested on in-distribution data (in-dist), (ii) they are trained on in-distribution data but tested on numbers 10x larger than the original in-distribution data (10x OOD), (iii) they are trained on in-distribution data, fine-tuned V projection matrices on numbers 10x larger than the original in-distribution data, and then tested on numbers 10x larger than the original in-distribution data (10x OOD after fine-tuning). We use the same empirical setting as before, where we train both models for 2000 epochs on in-distribution data. For fine-tuning, we adopt the common practice and train for only 20 epochs. We normalize the MSE by the scale factor so that the results indicate relative accuracy. The results in Table 1 indicate that fine-tuning only the V projection matrices brings the accuracy of the positional Transformer to the same level as the in-distribution case. However, for the standard Transformer, even after fine-tuning the V projection matrices, the accuracy on 10x test data is still very high compared to that of the positional Transformer. The comparison highlights another potential advantage of the positional Transformer for executing algorithmic tasks: one might efficiently fine-tune a

pre-trained positional Transformer on OOD data and achieve good performance.

Table 1: Performance (MSE test loss) of fine-tuned positional and standard transformer models

| Task | Architecture | in-dist | 10x OOD | 10x OOD after fine-tuning |
|---|---|---|---|---|
| Cumulative Sum | Positional | 6.07e-06 | 1.31e-03 | 1.32e-05 |
| | Standard | 5.20e-06 | 1.09e+00 | 1.23e-01 |
| Cumulative Minimum | Positional | 1.03e-05 | 3.92e-04 | 2.19e-05 |
| | Standard | 1.74e-06 | 1.39e-01 | 4.00e-02 |
| Sorting | Positional | 1.20e-04 | 7.37e-04 | 1.23e-04 |
| | Standard | 9.05e-04 | 5.18e-01 | 6.90e-02 |

### D.1.9. EMPIRICAL RESULTS USING GRAPH NEURAL NETWORKS (GNNS)

Graph Neural Networks are a popular choice for solving algorithmic tasks on graphs. We tested the performance of Graph Convolutional Network (GCN) and Graph Attention Network (GAT) in solving the cumulative sum and sorting tasks. Since the tasks tested have no underlying native graph, we tested these models on complete and star graphs. Notably, the original GAT architecture on a complete graph is similar to a standard transformer but differs in that the value, key, and query weights are shared in GAT. In Table 2 we present results reporting the median MSE loss. As the results indicate, neither GCN nor GAT works very well even for in-distribution data (OOD scaling factor = 1), let alone achieving OOD generalization. We believe this is because these tasks are inherently unsuitable for standard message-passing architectures.

Table 2: ID/OOD loss of Graph Convolutional Network (GCN) and Graph Attention Network (GAT)

| Task | Graph | Architecture | OOD scale factor | | | | | |
|---|---|---|---|---|---|---|---|---|
| | | | 1 | 2 | 3 | 4 | 5 | 10 |
| Cumulative Sum | Complete | GCN | 3.9e+0 | 2.1e+1 | 4.3e+1 | 7.4e+1 | 1.2e+2 | 4.8e+2 |
| | | GAT | 3.7e+0 | 2.0e+1 | 4.3e+1 | 7.5e+1 | 1.2e+2 | 5.4e+2 |
| | Star | GCN | 4.0e+0 | 2.0e+1 | 4.2e+1 | 7.4e+1 | 1.2e+2 | 4.6e+2 |
| | | GAT | 3.8e+0 | 2.0e+1 | 4.1e+1 | 7.5e+1 | 1.2e+2 | 5.0e+2 |
| Sorting | Complete | GCN | 1.9e-1 | 9.8e-1 | 2.2e+0 | 3.9e+0 | 6.1e+0 | 2.7e+1 |
| | | GAT | 1.9e-1 | 9.6e-1 | 2.1e+0 | 3.7e+0 | 5.6e+0 | 2.4e+1 |
| | Star | GCN | 1.9e-1 | 1.0e+0 | 2.1e+0 | 3.7e+0 | 6.1e+0 | 2.6e+1 |
| | | GAT | 1.9e-1 | 9.9e-1 | 2.0e+0 | 3.5e+0 | 5.6e+0 | 2.3e+1 |

### D.1.10. ACCURACY MEASURES

Given the regressive nature of the tasks considered in this work, the notion of model accuracy is not properly defined. However, we propose the following transformation strategies that assign binary labels ("correct"/"incorrect") to the models' outputs allowing us to measure their accuracy (with respect to these transformations):

- Rounding transformation: We evaluate the model on lists containing integers (while training is still done using real numbers) and round the model's output to the nearest integer (or nearest $0.5$ for the median task). A prediction is considered "correct" if the rounded and ground truth lists are the same. The generalization accuracies for all arithmetic tasks in the main paper using this transformation strategy are presented in Figure 27.

- Closeness transformation: We evaluate the model on lists of real numbers, considering a prediction "correct" if each entry in the predicted list is within an absolute precision of $0.05$ and a relative precision of $5\%$ compared to the corresponding entry of the ground truth list. The generalization accuracies for all arithmetic tasks in the main paper using this transformation strategy are presented in Figure 28.

It is important to note that the above metrics are quite unforgiving, as even a single element in the predicted list failing to meet the corresponding criterion results in the entire list being classified as "incorrect". It is therefore expected that

increasing the OOD scale factor causes the model's accuracy to decrease rapidly. Nevertheless, the positional Transformer significantly outperforms the standard Transformer in terms of accuracy in all five tasks.

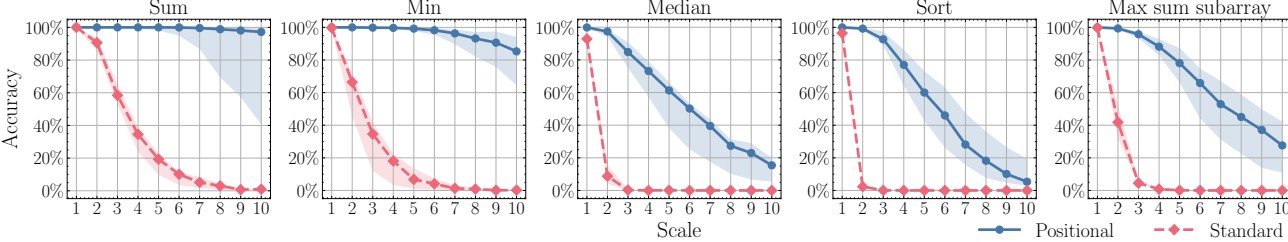

Figure 27: OOD accuracy when using "rounding transformation" across all five tasks for standard Transformers (red) and positional Transformers (blue) as a function of the scale factor. The solid line and shaded area denote the median and the region between the 10th and 90th percentiles over ten trials, respectively.

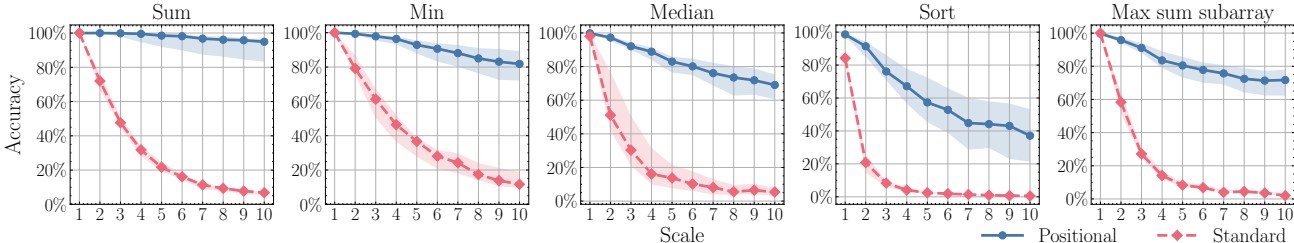

Figure 28: OOD accuracy when using "closeness transformation" across all five tasks for standard Transformers (red) and positional Transformers (blue) as a function of the scale factor. The solid line and shaded area denote the median and the region between the 10th and 90th percentiles over ten trials, respectively.

## D.2. $k$-hop induction heads

### D.2.1. EXPERIMENTAL SETTING

We adopt a slight modification of the configuration used in the numeric task for the $k$-hop induction heads task of Sanford et al. (2024). Specifically, since the input consists of tokens, we use an embedding layer as the encoding layer, with an embedding dimension of 60. The decoding step remains a linear layer to produce the logits, whose output dimension is set to the vocabulary size. Additionally, due to the sequential nature of the task, we apply a causal mask over the attention coefficients in both architectures.

The dataset creation is adopted from Sanford et al. (2024) and consists of the following types of characters: numbers from 0 to $k$, which indicate the number of hops required for each sample (as mentioned in Section 7, we set $k = 3$ for our experiments). The sequences themselves are constructed from specific character sets: $\{$a,b,c,d,e$\}$ for in-distribution and $\{$f,g,h,i,j$\}$ for out-of-distribution. An additional character ($\sim$) denotes cases where a $k$-hop character cannot be found. To ensure that most output tokens can be reached within the specified number of hops (thus avoiding $\sim$ as an output), we set the sequence length to 30.

### D.2.2. SAMPLE COMPLEXITY EXPERIMENT

In this section, we provide detailed results showcasing the training, in-distribution, and out-of-distribution test performance for the $k$-hop induction heads task as a function of the number of training samples used. From the results on Figure 29, we can observe a sharp contrast between the learnability of the standard and positional Transformers, indicating that, for this particular task, the sample complexity of positional Transformers is higher than that of standard Transformers. As discussed in Section 7, this is likely a result of the nature of the induction heads task, which is purposely designed to showcase the relational capabilities of self-attention. As a result, positional Transformers require more data to replicate this same behavior.

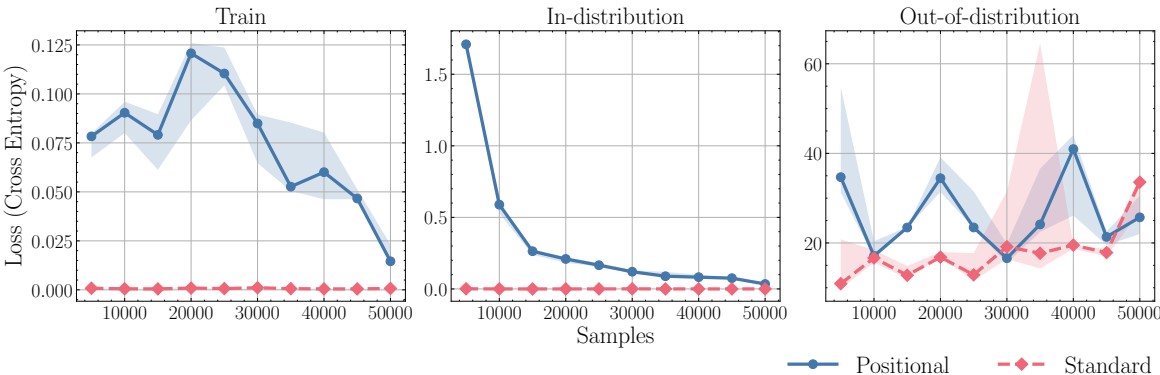

Figure 29: Training, in-distribution, and out-of-distribution test performance for the $k$-hop induction heads task are shown for standard Transformers (red) and positional Transformers (blue) as a function of the number of training samples (indicated on the x-axis).

### D.2.3. EXPRESSIVE POWER EXPERIMENT

In this section, we provide detailed results showcasing the training, in-distribution, and out-of-distribution test performance for the $k$-hop induction heads task as a function of the expressive power of the architectures, measured by the number of layers. From the results presented in Figure 30, we can conclude that, for this particular task, positional Transformers require many more layers to achieve a similar performance as standard Transformers, which empirically validates our findings established in Remark 5.2.

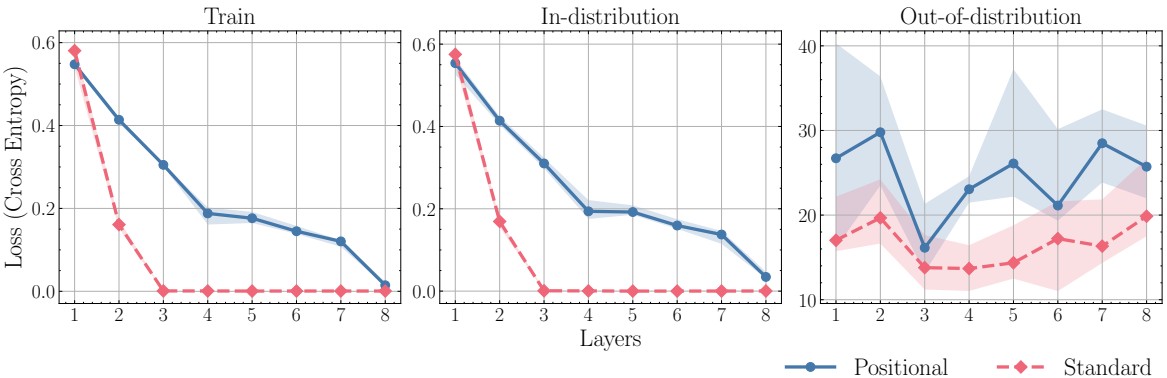

Figure 30: Training, in-distribution, and out-of-distribution test performance for the $k$-hop induction heads task are shown for standard Transformers (red) and positional Transformers (blue) as a function of the number of layers (indicated on the x-axis).

### D.3. Mixed-type input task

In this section, we provide details on the task of Section 7.3. Specifically, we give a formal description of the task, provide details on our experimental setting, and present additional experiments using a fine-tuned version of GPT2-large from HuggingFace (Radford et al., 2019). Lastly, we present additional experiments with a varying training sample size for both standard and positional Transformer.

### D.3.1. TASK DESCRIPTION

Let $n, k, m \in \mathbb{N}$ with $n \geq k$ and $n \geq m$. The input to the model is a list consisting of text and numerical values of the following form:

```
['Cati₁', v_{i_1}, 'Cati₂', v_{i_2}, ..., 'Catiₙ', v_{i_n}, 'Find {type} of Catj₁, Catj₂, ..., Catj_{k-1} and
                                    Catj_k']⁸
```

augmented by adding $m$ more alphanumeric elements at random positions in the list which we call irrelevant structure. Here, $i_1, \ldots, i_n \in \mathbb{N}$, $v_{i_1}, \ldots, v_{i_n} \in \mathbb{R}_{\geq 0}$, $j_1, \ldots, j_k \in \{i_1, \ldots, i_n\}$, and $\{\texttt{type}\}$ is one of $'\texttt{min}'$, $'\texttt{max}'$, $'\texttt{sum}'$. The answer to the query is either the minimum, maximum, or sum of the set $\{v_{j_1}, \ldots, v_{j_k}\}$ depending on $\{\texttt{type}\}$. The irrelevant structure is generated by concatenating the string $'\texttt{Cat}'$ with one of the characters in $\{'+', '-', '\_', '*'\}$ and a category identifier. Notice how this setting allows for one model to potentially process multiple query types. In fact, we present one such experiment later.

**OOD generalization**  For this task, we measure OOD generalization in three ways (simultaneously):

1. We test using category identifiers (i.e $i_1, i_2, \ldots, i_n$) that the models haven't encountered during training, and

2. The range of values $v_1, v_2, \ldots, v_n$ used for testing is larger than the range used for training.

3. The type of irrelevant structure used for testing is different than the type used for training. We detail this difference below.

D.3.2. EXPERIMENTAL SETTING

We fix $n = 8$, $k = 4$ and $m = 2$. Both standard and positional Transformers consist of 3 layers, two attention heads per layer, and an embedding dimension of 32. All characters of the non-numeric parts of the input are tokenized and passed through an embedding layer, while the numeric ones are passed through a linear layer. The category identifiers $i_1, i_2, \ldots, i_8$ are sampled randomly from $\{1, 2, \ldots, 20\}$ for training and $\{21, 22, \ldots, 40\}$ for testing. The query category identifiers are then sampled randomly from $\{i_1, i_2, \ldots, i_8\}$. The values $v_{i_1}, \ldots, v_{i_8}$ are sampled using the technique of Section 7 from $[0, 5]$ for training and from $[0, 5c]$ where $c \in \{1, 2, \ldots, 10\}$ is the scaling factor for testing. We also apply the rejection step of Section 7.2 when generating testing samples. Furthermore, when sampling a training list we form irrelevant structure by concatenating the string $'\texttt{Cat}'$ with either one of $'+'$ or $'-'$ (chosen at random) and a category identifier that is present in the actual input. For testing, irrelevant structure is formed similarly by concatenating the string $'\texttt{Cat}'$ with either one of $'\_'$ or $'*'$ (chosen at random) and a category identifier that is present in the actual (test) input. In both cases, irrelevant structure is injected into the "true" list at random positions. We run experiments for the following three variants of the task:

1. One where training and testing prompts consist exclusively of prompts where $\{\texttt{type}\}$ is $'\texttt{min}'$.

2. One where training and testing prompts consist exclusively of prompts where $\{\texttt{type}\}$ is $'\texttt{max}'$.

3. One where training and testing prompts consist exclusively of prompts where $\{\texttt{type}\}$ is either $'\texttt{min}'$ or $'\texttt{max}'$. We refer to this experiment as "multitasking" since it allows a single trained model to process different query types. In fact, the choice of minimum and maximum as the two query types is, in some sense, an "extreme" case, given the "opposite" nature of minimum and maximum operations.

Both positional and standard Transformer models are trained using Adam for 2000 epochs with a sample size of 50,000, a batch size of 1024, and an initial learning rate of $5 \cdot 10^{-4}$ which is controlled by a learning rate scheduler.

D.3.3. SAMPLE COMPLEXITY EXPERIMENTS

In Figures 31 to 33 we present the in-distribution and out-of-distribution performance of standard and positional Transformer on the three variations of the mixed-type input task. The conclusions are similar to those of Appendix D.1.3, namely, both models fit the training data but positional Transformer achieves significantly better out-of-distribution performance.

D.3.4. EXPERIMENTS WITH GPT2

For completeness, and given the nature of the task, we also perform experiments with a fined-tuned version of GPT2-large from HuggingFace (Radford et al., 2019) on the above task. For all three variations (min, max, and multitasking), we fix the

---

[8]The ellipses are not part of the prompt

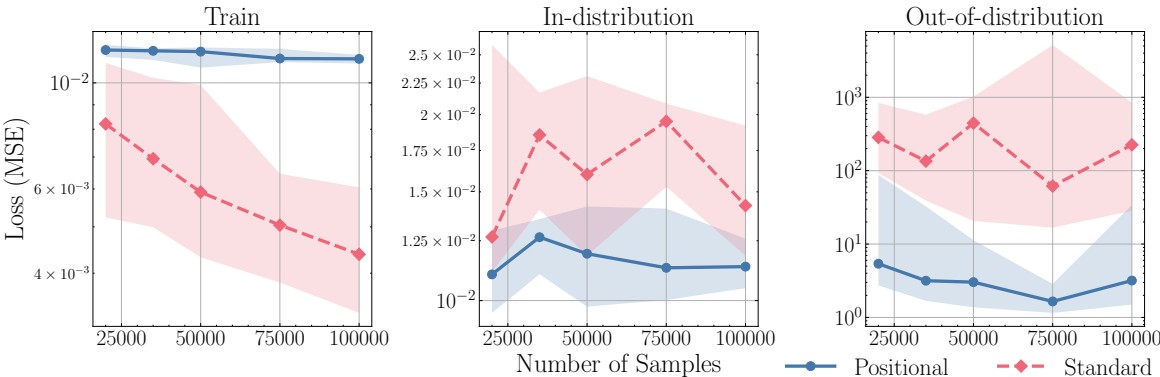

Figure 31: Training, in-distribution, and out-of-distribution test performance for the minimum variation of the mixed-type input task is shown for standard Transformers (red) and positional Transformers (blue) as a function of the number of training samples (indicated on the x-axis). Models are trained on category values in the range $[0, 5]$ with varying training set sizes. In-distribution testing is performed on the same domain, and out-of-distribution testing is conducted on an extended domain of category values, $[0, 50]$, each using 1,000 samples.

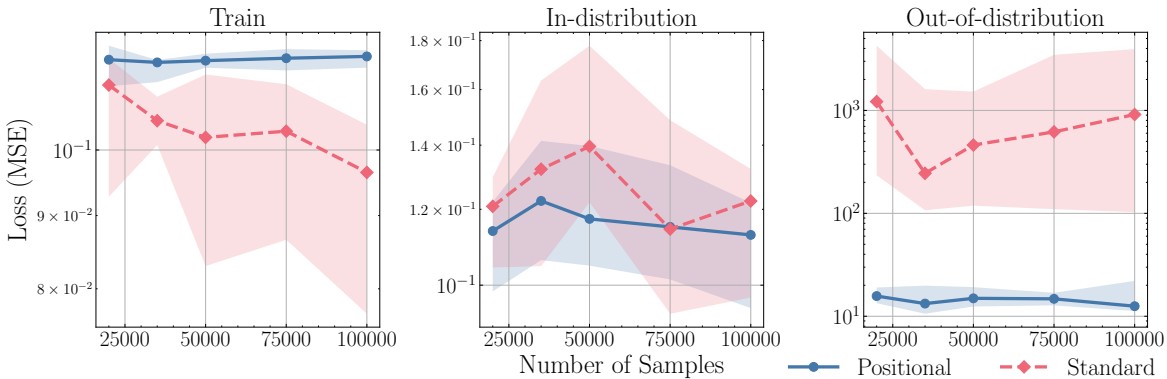

Figure 32: Training, in-distribution, and out-of-distribution test performance for the sum variation of the mixed-type input task is shown for standard Transformers (red) and positional Transformers (blue) as a function of the number of training samples (indicated on the x-axis). Models are trained on category values in the range $[0, 5]$ with varying training set sizes. In-distribution testing is performed on the same domain, and out-of-distribution testing is conducted on an extended domain of category values, $[0, 50]$, each using 1,000 samples.

precision of the input numbers to 4 digits (2 digits for the whole part and 2 digits after the decimal point) and the precision of the output to 4 digits for min and max (2 digits for the whole part and 2 digits after the decimal point) and 5 digits for sum (3 digits for the whole part and 2 digits after the decimal point). This covers all possible numbers that can be sampled. We then use byte pair encoding for tokenization and fine-tuned using 50,000 training samples for 3 epochs. Since this setting is autoregressive (next-token prediction), there were cases where the model's output did not correspond to a real number. As this did not occur frequently enough to be considered a problem, we report the median OOD losses (MSE and MAPE) ignoring samples for which no numeric value could be extracted from the model's output. The results of this experiment are presented in Figure 34. For easier comparison, we also include the losses of the positional Transformer on the same tasks.

## E. Probability of generating OOD test data in our empirical setting

Recall that we sample the training and test data in the following way. The training data consists of i.i.d samples whose values are drawn from the range $[-2, 2]$. To ensure diversity, for each training sample, we first select lower and upper bounds $\gamma_l$ and $\gamma_u$ uniformly in $[-2, 2]$, and then for each of the $n$ elements of the training sample, we select its value uniformly from the interval $[\gamma_l, \gamma_u]$. We employ a similar sampling strategy for testing but extend the value range to $[-2c, 2c]$, where

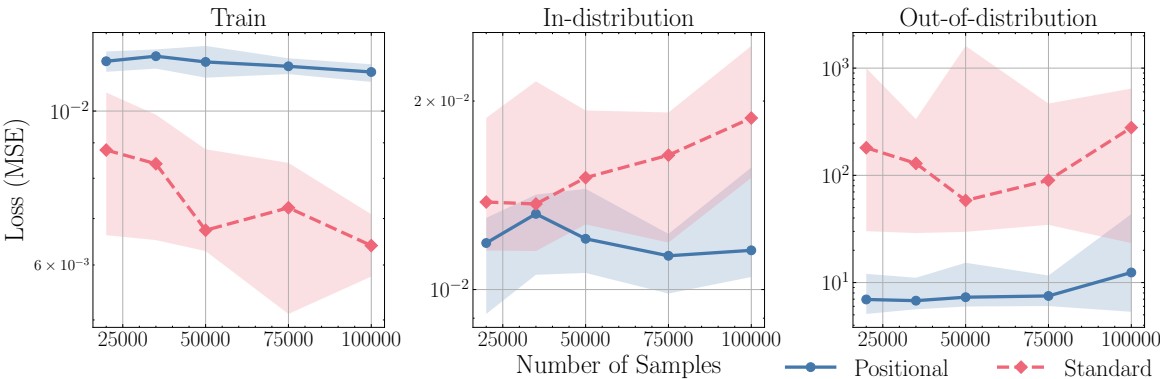

Figure 33: Training, in-distribution, and out-of-distribution test performance for the multitask variation of the mixed-type input task is shown for standard Transformers (red) and positional Transformers (blue) as a function of the number of training samples (indicated on the x-axis). Models are trained on category values in the range $[0, 5]$ with varying training set sizes. In-distribution is performed on the same domain, and out-of-distribution testing is conducted on an extended domain of category values, $[0, 50]$, each using 1,000 samples.

$c > 1$ is the OOD scale factor. Additionally, during the test sampling process, we apply a rejection step to ensure that either $\gamma_l < -2$ or $\gamma_u > 2$, while maintaining $-2c \leq \gamma_l \leq \gamma_u \leq 2c$.

We will compute the probability that a randomly sampled test instance $x \in \mathbb{R}^n$ lies in the domain of the training distribution, i.e., we will compute $\mathbb{P}_{x \sim \mathcal{D}_{\text{test}}(\mathcal{X})}(x \in [-2, 2]^n)$. In particular, we will show that this probability is proportional to $1/nc^2$. Consequently, in our experiments, the majority of the test data lie outside of the domain of the training distribution.

Without loss of generality, let us assume that the training data are sampled within the interval $[-1, 1]$ and the test data are sampled within the interval $[-c, c]$, where $c$ is the OOD scale factor. Note that this does not affect the probability that we want to compute. In the test sampling process, when we sample two uniform numbers $\gamma_\ell$ and $\gamma_u$ from $[-c, c]$, exactly one of the following 3 disjoint events can happen.

- *Event A*. Exactly one of $\gamma_\ell$ and $\gamma_u$ lies in $[-1, 1]$. This happens with probability $2(\frac{c-1}{c})(\frac{1}{c})$.

- *Event B*. Neither $\gamma_\ell$ nor $\gamma_u$ is inside the interval $[-1, 1]$. This happens with probability $(\frac{c-1}{c})^2$.

- *Event C*. Both $\gamma_\ell$ and $\gamma_u$ are inside the interval $[-1, 1]$. This happens with probability $\frac{1}{c^2}$.

Our rejection step rejects the samples generated under Event C. Therefore, in our setting when we sample a pair of $\gamma_\ell$ and $\gamma_u$ in order to generate a single instance of the test list, we have that

$$\mathbb{P}(\text{Event A}|\text{Rejecting Event C}) = \frac{2(c-1)}{c^2 - 1}, \quad \mathbb{P}(\text{Event B}|\text{Rejecting Event C}) = \frac{(c-1)^2}{c^2 - 1}. \tag{23}$$

We analyze the probability of generating OOD test data under each event. First, let us suppose that Event A happens when we sample $\gamma_\ell$ and $\gamma_u$. This means that either $\gamma_\ell \in [-c, -1)$ or $\gamma_u \in (1, c]$ (but not both). More precisely, the following two sub-events partition Event A:

- *Event A.1*. $\gamma_\ell \in [-1, 1]$ and $\gamma_u \in (1, c]$. Given that Event A happens, this sub-event happens with probability 1/2.

- *Event A.2*. $\gamma_\ell \in [-c, -1)$ and $\gamma_u \in [-1, 1]$. Given that Event A happens, this sub-event happens with probability 1/2.

By symmetry of the probability distributions, the probability that we wish to compute remains the same under both of the above sub-events. Therefore, let us focus on Event A.1. Suppose that Event A.1 happens, i.e., $\gamma_\ell \in [-1, 1]$ and $\gamma_u \in (1, c]$.

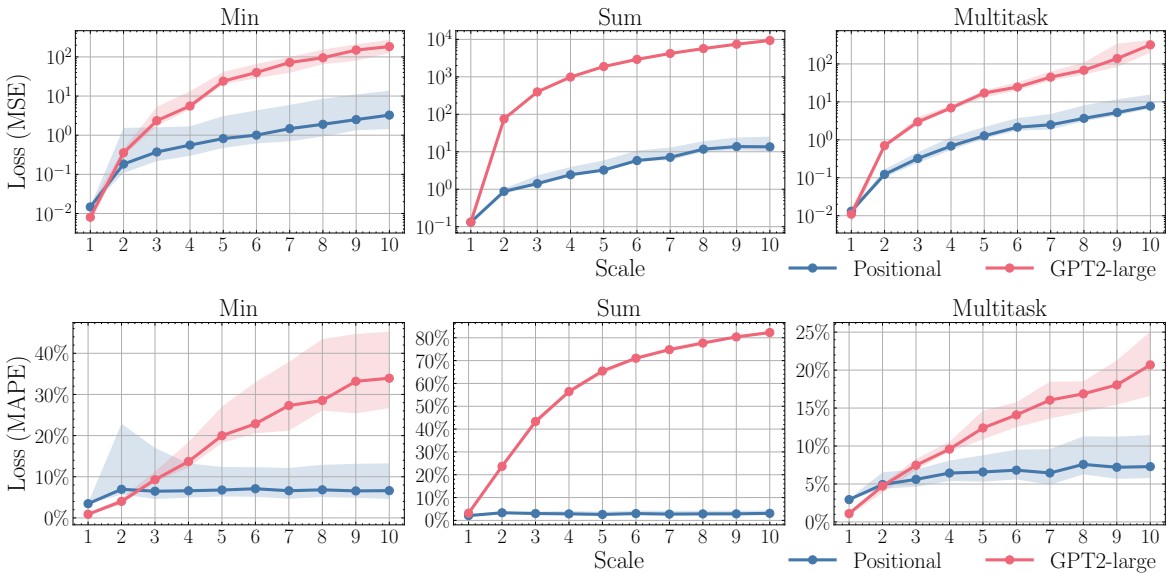

Figure 34: This experiment shows ID/OOD losses (measured as mean squared error, MSE, and mean absolute percentage error, MAPE) for fine-tuned GPT2-large (red) and positional Transformers (blue) across all three variations of the mixed-type input task. The x-axis represents the OOD scale factor. The solid line and shaded area denote the median and the region between the 10$^{\text{th}}$ and 90$^{\text{th}}$ percentiles over ten trials, respectively.

Conditioning on this event, we know that $\gamma_\ell$ is uniform on $[-1, 1]$ and $\gamma_u$ is uniform on $(1, c)$. The probability density functions for $\gamma_\ell$ and $\gamma_u$ are

$$f_{\gamma_\ell}(s) = \frac{1}{2} \cdot \mathbb{I}(-1 \le s \le 1), \quad f_{\gamma_u}(t) = \frac{1}{c-1} \cdot \mathbb{I}(1 < t \le c).$$

Let $X \in \mathbb{R}^n$ be a random vector whose $i$th coordinate $X_i$ is independently and uniformly sampled from the interval $[\gamma_\ell, \gamma_u]$, then the conditional probability density function for $X$ given $\gamma_\ell, \gamma_u$ is

$$f_X(x_1, x_2, \ldots, x_n | \gamma_\ell = s, \gamma_u = t) = \left(\frac{1}{t-s}\right)^n \cdot \mathbb{I}(s \le x_i \le t, \forall i).$$

Therefore, the joint density function is

$$f_{X, \gamma_\ell, \gamma_u}(x_1, \ldots, x_n, s, t) = \frac{1}{2} \frac{1}{(c-1)} \left(\frac{1}{t-s}\right)^n \cdot \mathbb{I}(-1 \le s \le 1, 1 < t \le c, s \le x_i \le t, \forall i).$$

It follows that

$$p_{\text{A,in}} := \mathbb{P}_{X, \gamma_\ell, \gamma_u}\left(X_i \in [-1, 1], \forall i \in [n] \middle| \text{Event A}\right)$$

$$= \int_{s \in [-1,1]} \int_{t \in (1,c)} \int_{x \in [s,1]^n} \frac{1}{2} \frac{1}{(c-1)} \left(\frac{1}{t-s}\right)^n dx\, dt\, ds$$

$$= \frac{1}{2} \frac{1}{(c-1)} \int_{s \in [-1,1]} \int_{t \in (1,c)} \left(\frac{1-s}{t-s}\right)^n dt\, ds$$

$$= \frac{1}{2} \frac{1}{(c-1)} \frac{1}{(n-1)} \int_{s \in [-1,1]} (1-s) \left(1 - \left(\frac{1-s}{c-s}\right)^{n-1}\right) ds$$

$$= \frac{1}{2} \frac{1}{(c-1)} \frac{1}{(n-1)} \left[\int_{s \in [-1,0]} (1-s) \left(1 - \left(\frac{1-s}{c-s}\right)^{n-1}\right) ds\right.$$

$$+ \int_{s \in [0,1]} (1-s) \left( 1 - \left( \frac{1-s}{c-s} \right)^{n-1} \right) ds \Bigg]$$

$$\leq \frac{1}{2} \frac{1}{(c-1)} \frac{1}{(n-1)} \left[ \int_{s \in [-1,0]} (1-s) \left( 1 - \left( \frac{1}{c} \right)^{n-1} \right) ds + \int_{s \in [0,1]} (1-s) ds \right]$$

$$= \frac{3 \left( 1 - 1/c^{n-1} \right) + 1}{4(n-1)(c-1)}. \tag{24}$$

This is the probability that, under Event A, a randomly generated test sample lies within the domain of the training distribution. Again, recall that by scaling down the domain of the test distribution to $[-c, c]$ accordingly, we have assumed that the domain of the training distribution is $[-1, 1]^n$ without loss of generalization.

Now suppose that Event B happens. In this case both $\gamma_\ell$ and $\gamma_u$ are uniformly distributed over $[-c, -1) \cup (1, c]$. The following two sub-events partition Event B:

- *Event B.1.* Either both $\gamma_\ell, \gamma_u > 1$ or both $\gamma_\ell, \gamma_u < -1$. Given that Event B happens, this sub-event happens with probability 1/2.

- *Event B.2.* $\gamma_\ell < -1$ and $\gamma_u > 1$. Given that Event B happens, this sub-event happens with probability 1/2.

Let $X \in \mathbb{R}^n$ be a random vector whose $i$th coordinate $X_i$ is independently and uniformly sampled from the interval $[\gamma_\ell, \gamma_u]$. Note that under Event B.1, one always has that $X \notin [-1, 1]^n$, i.e.,

$$p_{\text{B.1,in}} := \mathbb{P}_{X, \gamma_\ell, \gamma_u} \left( X_i \in [-1, 1], \forall i \in [n] \Big| \text{Event B.1} \right) = 0. \tag{25}$$

Therefore let us consider Event B.2. Conditioning on this event, we know that $\gamma_\ell$ is uniform on $[-c, -1)$ and $\gamma_u$ is uniform on $(1, c]$. The joint density function (conditional on Event B.2) for $X, \gamma_\ell, \gamma_u$ is

$$f_{X, \gamma_\ell, \gamma_u}(x_1, \ldots, x_n, s, t) = \frac{1}{(c-1)^2} \left( \frac{1}{t-s} \right)^n \cdot \mathbb{I}(-c \leq s \leq 1, 1 < t \leq c, s \leq x_i \leq t, \forall i).$$

Therefore, we have that

$$p_{\text{B.2,in}} := \mathbb{P}_{X, \gamma_\ell, \gamma_u} \left( X_i \in [-1, 1], \forall i \in [n] \Big| \text{Event B.2} \right)$$

$$= \int_{s \in [-c,1)} \int_{t \in (1,c]} \int_{x \in [s,1]^n} \frac{1}{(c-1)^2} \left( \frac{1}{t-s} \right)^n dx \, dt \, ds$$

$$= \frac{1}{(c-1)^2} \int_{s \in [-c,1)} \int_{t \in (1,c]} 2^n \left( \frac{1}{t-s} \right)^n dt \, ds$$

$$= \frac{2^n}{(c-1)^2(n-1)} \int_{s \in [-c,1)} \left( \frac{1}{(1-s)^{n-1}} - \frac{1}{(c-s)^{n-1}} \right) ds$$

$$= \begin{cases} \dfrac{4 - 8(\frac{2}{1+c})^{n-2} + 4(\frac{1}{c})^{n-2}}{(c-1)^2(n-1)(n-2)}, & \text{if } n \geq 3, \\[2ex] \dfrac{2 \log(c+1) - \log c - 2 \log 2}{(c-1)^2}, & \text{if } n = 2. \end{cases} \tag{26}$$

Combining Equation (23), Equation (24), Equation (25), Equation (26), we get that, if $n \geq 3$,

$$p_{\text{in}} := \mathbb{P}_{x \sim \mathcal{D}_{\text{test}}(\mathcal{X})}(x \in [-1, 1]^n) \leq \frac{3(1 - 1/c^{n-1}) + 1}{2(c^2 - 1)(n-1)} + \frac{2 - 4(\frac{2}{1+c})^{n-2} + 2(\frac{1}{c})^{n-2}}{(n-1)(n-2)(c^2 - 1)} \tag{27}$$

$$\leq \frac{2}{(c^2 - 1)(n-1)} + \frac{2}{(n-1)(n-2)(c^2 - 1)}$$

$$= O\left( \frac{1}{nc^2} \right)$$

and if $n = 2$,

$$p_{\text{in}} \leq \frac{3(1 - 1/c) + 1 + 2\log(c+1) - \log c - 2\log 2}{2(c^2 - 1)} \leq \frac{3(1 - 1/c) + 9/8}{2(c^2 - 1)}. \tag{28}$$

In the above, $p_{\text{in}}$ is the probability that a randomly sampled test list $x \in \mathbb{R}^n$ has all its elements lie within $[-1, 1]$, that is, the probability that $x$ lies within the domain of the training distribution. This probability is at most $O(1/nc^2)$. Suppose that we generate $N$ test instances, then a straightforward application of the multiplicative Chernoff bound yields that with probability at least $N^{-C}$ for some constant $C > 0$, at most $O(\frac{N}{nc^2})$ samples will lie in the domain of the training distribution.

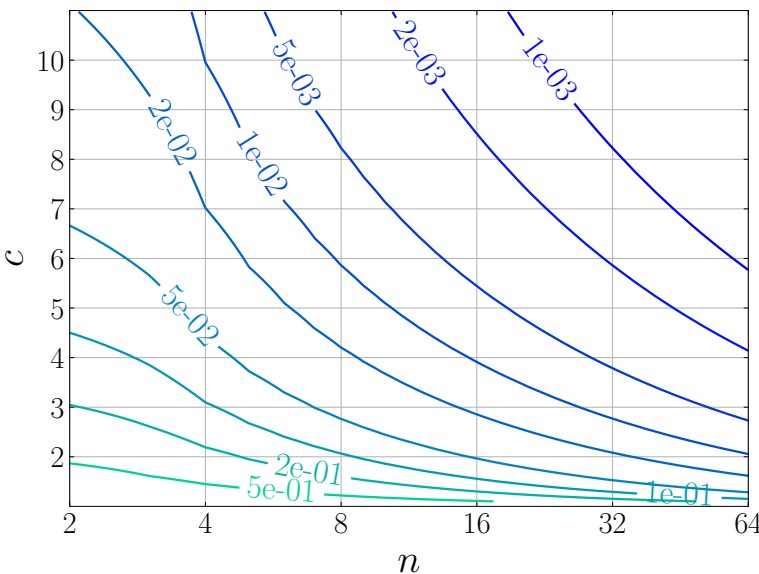

Figure 35: Contour plot of $\mathbb{P}_{x \sim \mathcal{D}_{\text{test}}(\mathcal{X})}(x \in \text{supp}(\mathcal{D}_{\text{train}}(\mathcal{X})))$, i.e., the probability (upper bound in Equation (27) and Equation (28)) that a randomly sampled test instance $x \in \mathbb{R}^n$ lies in the domain of the training distribution.

For $n \in \{2, 4, 8, 16, 32\}$ and $c \in \{1, 2, \ldots, 10\}$ which we consider in our experiments, Figure 35 shows a contour plot of the probability upper bound Equation (27) and Equation (28). This probability is sufficiently small such that the majority of test instances in our test data does not belong to the domain of the training distribution.

For small $n$ and $c$, to determine the fraction of sampled test instances that will be within the domain of the training distribution, it is more informative to directly invoke the additive Chernoff bound with $p_{\text{in}}$. Let $N$ denote the total number of test instances that we sample, and further let $N_{\text{in}}$ denote the number of sampled instances that lie in the domain of the training distribution. Then by the additive Chernoff bound we have that

$$\mathbb{P}(N_{\text{in}} \geq N(p_{\text{in}} + \epsilon)) \leq \exp(-2N\epsilon^2). \tag{29}$$

For example, suppose that we sample $N = 1000$ test instances from the test distribution. Suppose that we generate the test data using list length $n = 2$ and OOD scale factor $c = 2$. Then in this case $p_{\text{in}} \leq 0.4375$. Take $\epsilon = 0.0625$. Then (29) says that with probability at least 0.9995, at least $N/2$ samples do not lie in the domain of the training distribution. For another example with slightly larger $n$ and $c$, suppose that we generate the test data using list length $n = 8$ and OOD scale factor $c = 10$. Then $p_{\text{in}} \leq 0.0034$. Take $\epsilon = 0.0466$. Then (29) says that with probability at least 0.98, more than 95% of test instances do not lie in the domain of the train distribution.

## F. Potential reasons for failure in self-attention for out-of-distribution

In this section, we discuss potential reasons for the shortcomings in out-of-distribution generalization of standard Transformers for algorithmic tasks. While it is more straightforward to elicit reasons for the success of positional attention – motivated by the *algorithmic alignment* (Xu et al., 2020) between positional Transformers and parallel computational models – it is considerably more challenging to pinpoint the causes of failure in standard Transformers.

Firstly, Transformers can simulate parallel algorithms, as demonstrated by (Sanford et al., 2024). Intuitively, a single layer of self-attention should be more powerful than positional attention, as it leverages attention beyond positional encodings and allows for a more flexible structure in response to input variations. However, as discussed in Section 5, executing parallel algorithms does not require using anything beyond positional information in attention.

Assuming that standard Transformers should adopt positional information to effectively execute parallel algorithms, the operations required by standard Transformers become increasingly difficult than positional Transformers for two main reasons:

1. Self-attention layers must learn to ignore input values and exploit positional information.

2. Transformer layers must preserve positional encodings for subsequent layers.

Namely, these desirable properties of positional Transformers present two significant challenges for standard Transformers. The first challenge arises naturally from the differences between standard and positional attention mechanisms. The second challenge highlights the compositional structure of attention layers, which can be detrimental during training. Specifically, the operations performed by each attention and MLP layer can degrade the inputs of subsequent layers.

This issue is further emphasized in Remark B.6, where we state that while positional Transformers can represent any softmax pattern at any layer, standard Transformers may fail to do so due to potential degradation of the attention inputs. Although residual connections can mitigate this issue by preserving input information, they must ensure that no overlaps hinder the use of positional encodings in subsequent layers. Moreover, this problem compounds across layers, making training more difficult as errors in earlier layers adversely affect subsequent computations.

Nevertheless, these remain speculative reasons for the observed failure of standard Transformers. Determining the exact causes and the difficulty of achieving the two aforementioned goals through training requires a thorough analysis of the training dynamics, which is inherently challenging. Future in-depth work within the mechanistic interpretability framework (Nanda et al., 2023) can potentially shed light on these issues by inspecting network parameters at convergence, thereby uncovering the underlying reasons for the failure of standard Transformers.

Along this direction, we present some empirical evidence, in Figure 36 and Figure 37, that self-attention layers in a trained Transformer model can be highly sensitive to input values. In particular, the attention weights change dramatically when the input values of the test data do not necessarily lie in the domain of the training data. This suggests that self-attention potentially overfits the training data and, therefore, offers a plausible explanation for why the standard Transformers exhibit such poor value generalization in our experiments.

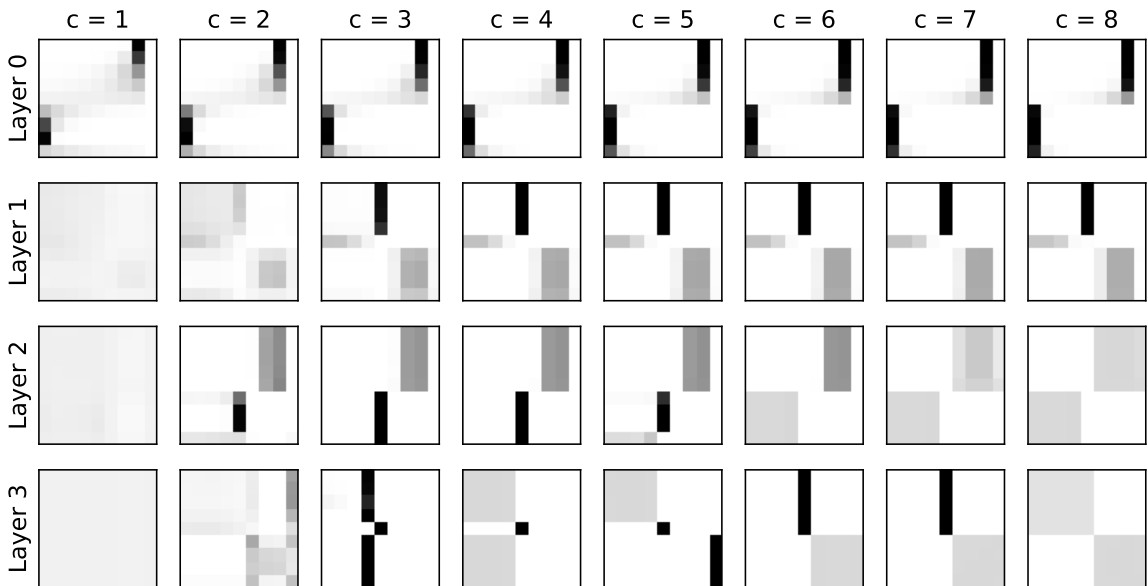

Figure 36: Visualization of learned attention weights in the standard Transformer model trained to solve the sorting task in our experiments. The input list to the model is $cX$ where $X = [1.75, 1.25, 0.75, 0.25, -0.25, -0.75, -1.25, -1.75]$ and $c = 1, 2, \ldots, 8$ is a scaling factor. The model is trained on data whose input values range from -2 to 2. Therefore $c = 1$ gives in-distribution data and larger $c$ yields OOD data. For each layer in the architecture, we plot 1 of the 2 attention heads for illustration purposes. The trend for the other head is similar. Observe that the attention weights change dramatically as we increase the scaling factor of input values, with deeper layers suffering from more radical changes in the attention pattern under even a small change in the scale (e.g. going from $c = 1$ to $c = 2$). This behavior potentially explains why the standard Transformer model performs poorly on OOD test data in our experiments.

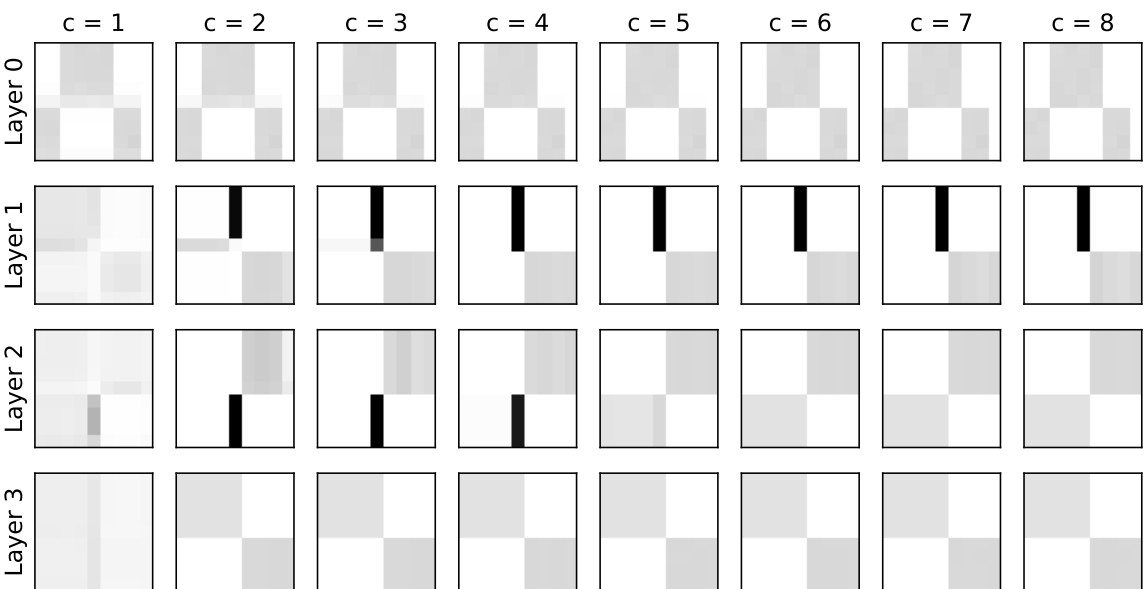

Figure 37: Another visualization of learned attention weights in the standard Transformer model. We use the same setting as described in Figure 36, except that the input list is $cX$ where $X = [2, 2, -2, -2, -2, -2, 2, 2]$. Again, we observe that the attention weights are highly sensitive to the scaling factor $c$, especially those at deeper layers.

## G. Informal Discussion on Out-of-distribution Generalization Bounds and future work

The topic of OOD generalization has been studied extensively from a theoretical perspective. Researchers often focus on bounding the risk on the test distribution using a mixture of terms that go to zero as the number of samples increases, plus a divergence term that does not necessarily go to zero depending on the distributions and the hypothesis class. For an extensive survey, we refer the reader to (Redko et al., 2022). Although, in general, such bounds offer valuable intuition, they might not be tight for our particular setting. In particular, we examine a popular type of bound found in (Mansour et al., 2009) which can be large even if the difference in the support of the train and test distributions is the smallest it can be. Note that there are more types of bounds in (Redko et al., 2022) than the one found in (Mansour et al., 2009). Although we have not conducted an in-depth analysis of all cases, we note that all of them depend in a worst-case on the hypothesis class. We believe that to improve upon such generic bounds, one must consider the dynamics of the training procedure for deep positional attention architectures. We find this topic extremely interesting, but we leave it for future work, as it is highly non-trivial.

In what follows, we use a popular example of one of these bounds (Mansour et al., 2009) and illustrate why it is not tight for a simple task of interest. Briefly, the main issue with this particular OOD bound is that it depends in a worst-case manner on the hypothesis class. We demonstrate this issue using the task of computing the minimum over an array of length $n$. We assume that $n$ is even. For simplicity, we do not work with the cumulative version of the minimum problem, as we did in the main paper. Therefore, the ground truth is simply the minimum of the input array. $\mathcal{D}_{\text{train}}$ is the train distribution over arrays of length $n$, where each component of the array is sampled independently and uniformly at random from integers in the range 0 to $L_{\mathcal{D}\text{train}}$, where $L_{\mathcal{D}\text{train}}$ is a constant. $\mathcal{D}_{\text{test}}$ is the test distribution over arrays of length $n$, where each component of the array is sampled independently and uniformly at random from integers in the range 0 to $L_{\mathcal{D}_{\text{train}}} + z$, where $z \geq 0$ is a constant integer. We use an equal number of samples from the train and test distributions, denoted by $m$. The loss function is $\ell(h(x), y) = |h(x) - y|$, where $h$ is a hypothesis, $x$ is an input array of length $n$, and $y = \min(x)$, which is the minimum function over the array $x$.

The hypothesis class $H$ is the architecture in Equation (1) with $\log_2 n$ layers, 2 heads per layer, and $W_O$ and $W_V$ as the identity matrices for all layers. For positional vectors in general position with dimension $n$, Lemma B.3 implies that there exist key and query matrices of size $n \times n$ that can represent any attention pattern at each layer. The MLP at each layer consists of 2 layers with a hidden dimension equal to 4. We use the ReLU activation function in all MLPs. This allows the MLP to represent the minimum and maximum functions on two input values exactly. This is because the minimum and maximum functions can be written using ReLUs and linear operations:

$$\min(x_1, x_2) = \frac{1}{2}(\text{ReLU}(x_1 + x_2) - \text{ReLU}(-x_1 - x_2) - \text{ReLU}(x_1 - x_2) - \text{ReLU}(x_2 - x_1))$$

and

$$\max(x_1, x_2) = \frac{1}{2}(\text{ReLU}(x_1 + x_2) - \text{ReLU}(-x_1 - x_2) + \text{ReLU}(x_1 - x_2) + \text{ReLU}(x_2 - x_1)).$$

Note that the MLP's ability to represent the minimum function for two inputs exactly is also the reason it can represent the maximum function. In other words, the exact representation of the minimum function comes with the consequence that the MLP can also represent the maximum function for two inputs. This observation is crucial later when we show that an existing popular bound from (Mansour et al., 2009) is not tight for this particular task.

Furthermore, we assume that $|h(x)|$ is constant, which further implies that the magnitude of the loss is bounded above by a constant. Observe that this hypothesis class can represent a binary tree reduction algorithm for the minimum and maximum functions. This is possible because positional attention can represent any attention pattern, and the MLPs can represent the minimum and maximum functions for two inputs exactly. Specifically, the first layer of the positional attention architecture can represent the connections between layers 0 (leaf nodes) and 1 in the binary tree computational graph, the second layer can represent the connections between layers 1 and 2, and so on, up to the $\log_2 n$-th layer. The MLPs are used locally at each node of the binary tree to compute the minimum between two input values. Therefore, the minimum and maximum functions over an array of $n$ elements are in the hypothesis class.

Let us now discuss one of the most popular OOD generalization bound results for regression. We will use the third case of Theorem 8 in (Mansour et al., 2009), which states the following.

**Theorem G.1** (Theorem 8 in (Mansour et al., 2009), repurposed for the minimum function task)**.** *Assume that the loss function $\ell$ is symmetric, it obeys the triangle inequality, and it is bounded above by a constant. If the minimum function is in*

*the hypothesis class H, then, for any hypothesis $h \in H$, the following holds:*

$$R_{\mathcal{D}_{test}}(h) \leq R_{\mathcal{D}_{train}}(h) + disc_\ell(\mathcal{D}_{test}, \mathcal{D}_{train})$$

*where*

$$disc_\ell(\mathcal{D}_{test}, \mathcal{D}_{train}) = \max_{h,h' \in H} |R_{\mathcal{D}_{test}}(h, h') - R_{\mathcal{D}_{train}}(h, h')|$$

*and*

$$R_{\mathcal{D}_{test}}(h, h') = \mathbb{E}_{(x,y) \sim \mathcal{D}_{test}}[\ell(h(x), h'(x))].$$

*and $R_{\mathcal{D}_{train}}(h, h')$ is defined similarly.*

The above theorem states that the difference in risk between the test and train distributions is bounded only by the discrepancy term between the two distributions. It is important to note that the discrepancy term depends on the hypothesis class and measures the worst-case difference in the train and test risks within that class. The fact that the discrepancy considers the worst-case difference is why this bound is not tight for our task of computing the minimum function.

Let us now focus on the discrepancy term. Corollary 7 in (Mansour et al., 2009) states that

$$disc_\ell(\mathcal{D}_{test}, \mathcal{D}_{train}) \leq disc_\ell(\hat{\mathcal{D}}_{test}, \hat{\mathcal{D}}_{train}) + 4(\hat{\mathcal{R}}_{\mathcal{S}_{test}}(H) + \hat{\mathcal{R}}_{\mathcal{S}_{train}}(H)) + \mathcal{O}\left(\sqrt{\frac{1}{m}}\right),$$

where $\hat{\mathcal{D}}_{test}$ and $\hat{\mathcal{D}}_{train}$ are the empirical versions of the test and train distributions, repsectively. We will use the common assumption that these empirical distributions are uniform over the samples. $\mathcal{S}_{test}$ and $\mathcal{S}_{train}$ are the sample sets for the train and test distributions, respectively. Moreover, $\hat{\mathcal{R}}_{\mathcal{S}_{test}}(H)$ and $\hat{\mathcal{R}}_{\mathcal{S}_{train}}(H)$ are the empirical Rademacher complexities for the train and test distributions, respectively. It is well-known that the part of the bound corresponding to the empirical Rademacher complexities goes to zero as the number of samples increases. The same holds for the square-root term in the bound. Therefore, the only term left to understand is the discrepancy between the empirical distributions. Let us try to understand how this term behaves. Its definition is:

$$disc_\ell(\hat{\mathcal{D}}_{test}, \hat{\mathcal{D}}_{train}) = \max_{h,h' \in H} \left| \frac{1}{m} \sum_{x \in \mathcal{S}_{test}} \ell(h(x), h'(x)) - \frac{1}{m} \sum_{x \in \mathcal{S}_{train}} \ell(h(x), h'(x)) \right|.$$

A lower bound of $disc_\ell(\hat{\mathcal{D}}_{test}, \hat{\mathcal{D}}_{train})$ is given by setting $h$ to be the minimum function and $h'$ to the maximum function:

$$disc_\ell(\hat{\mathcal{D}}_{test}, \hat{\mathcal{D}}_{train}) \geq \left| \frac{1}{m} \sum_{x \in \mathcal{S}_{test}} (\max(x) - \min(x)) - \frac{1}{m} \sum_{x \in \mathcal{S}_{train}} (\max(x) - \min(x)) \right|$$

We claim that for polynomial number of samples $m$, e.g., $m = n^c$, where $c$ is a positive integer, there exists $n_0$, such that for all $n \geq n_0$ we have that $\min(x) = 0$ and $\max(x) = L_{\mathcal{D}_{train}}$ for all $x \in \mathcal{S}_{train}$, and $\min(x) = 0$ and $\max(x) = L_{\mathcal{D}_{train}} + z$ for all $x \in \mathcal{S}_{test}$ with probability at least $0.8$. The proof of this claim is trivial and we provide it below. For now, let us discuss the implications of this claim. We have that $disc_\ell(\hat{\mathcal{D}}_{test}, \hat{\mathcal{D}}_{train}) \geq L_{\mathcal{D}_{train}} + z - L_{\mathcal{D}_{train}} = z$ with probability at least $0.8$. Therefore, even if $z$ is the smallest it can be such that there is a difference between the train and test distributions in this particular setting, i.e., $z = 1$, then the empirical discrepancy is going to be at least $1$. This means that the upper bound of Theorem G.1 is at least one, and it is not going to zero as the number of samples increases. This is because the discrepancy definition considers the worst-case scenario without considering the training procedure. In practice, the training procedure may help to discover a hypothesis that is close to the minimum function since the minimum function is part of the hypothesis class. If the hypothesis discovered by the training procedure is close enough to the minimum function, the OOD generalization error may be much smaller than $1$. Therefore, depending on the learning task and the hypothesis class, the bound in Theorem G.1 can be loose.

Consider the following example, $L_{\mathcal{D}_{train}} = 3$ and $z = 1$. Therefore, the largest value in the test distribution is $4$. For $m$ and $n$ as noted above, the bound implies that the loss might be up to $1$ for any hypothesis $h$. This further implies that for any hypothesis $h$ the relative error might be up to $25\%$ with probability at least $0.8$, despite the fact that the hypothesis class includes the true function and the training procedure could converge to a good approximation of it.

Let us prove the above probability claim. For $x \in \mathcal{S}_{\text{train}}$ we have

$$\mathbb{P}(x_i \neq 0 \text{ for all } i \in [n]) = \prod_{j=1}^{n} \mathbb{P}(x_j \neq 0)$$

$$= \prod_{j=1}^{n} \mathbb{P}(x_j \in \{1, 2, \ldots, L_{\mathcal{D}_{\text{train}}}\})$$

$$= \prod_{j=1}^{n} \frac{L_{\mathcal{D}_{\text{train}}}}{L_{\mathcal{D}_{\text{train}}} + 1}$$

$$= \left( \frac{L_{\mathcal{D}_{\text{train}}}}{L_{\mathcal{D}_{\text{train}}} + 1} \right)^{n}.$$

Similarly, we have that

$$\mathbb{P}(x_i \neq L_{\mathcal{D}_{\text{train}}} \text{ for all } i \in [n]) = \left( \frac{L_{\mathcal{D}_{\text{train}}}}{L_{\mathcal{D}_{\text{train}}} + 1} \right)^{n}.$$

Furthermore, we have that

$$\mathbb{P}(\exists \, i, j \in [n] : x_i = 0 \text{ and } x_j = L_{\mathcal{D}_{\text{train}}}) = 1 - \mathbb{P}(x_i \neq 0 \, \forall \, i \in [n] \text{ or } x_i \neq L_{\mathcal{D}_{\text{train}}} \, \forall \, i \in [n])$$

$$\geq 1 - \mathbb{P}(x_i \neq 0 \, \forall \, i \in [n]) - \mathbb{P}(x_i \neq L_{\mathcal{D}_{\text{train}}} \, \forall \, i \in [n])$$

$$= 1 - 2 \left( \frac{L_{\mathcal{D}_{\text{train}}}}{L_{\mathcal{D}_{\text{train}}} + 1} \right)^{n}.$$

Therefore, we conclude that

$$\mathbb{P}(\text{all samples } x \in \mathcal{S}_{\text{train}} \text{ have at least one } 0 \text{ or } L_{\mathcal{D}_{\text{train}}}) \geq \left( 1 - 2 \left( \frac{L_{\mathcal{D}_{\text{train}}}}{L_{\mathcal{D}_{\text{train}}} + 1} \right)^{n} \right)^{m}.$$

and, similarly, we conclude that

$$\mathbb{P}(\text{all samples } x \in \mathcal{S}_{\text{test}} \text{ have at least one } 0 \text{ or } L_{\mathcal{D}_{\text{train}}} + z) \geq \left( 1 - 2 \left( \frac{L_{\mathcal{D}_{\text{train}}} + z}{L_{\mathcal{D}_{\text{train}}} + z + 1} \right)^{n} \right)^{m}.$$

For a polynomial number of samples $m$, e.g., $m = n^c$, where $c$ is a positive integer, there exists some $n_0 \in \mathbb{N}$, such that for all $n \geq n_0$ we have that the latter two probabilities are at least $0.9$ (since for $m = n^c$ both lower bounds tend to $1$ as $n$ tends to infinity). Therefore, our claim about the minimum and maximum over the sampled arrays holds with probability at least $0.81$.

