# OpenReview forum: "Positional Attention: Expressivity and Learnability of Algorithmic Computation"
_ICML.cc/2025/Conference — ICML 2025 poster_

### Official Review · Reviewer_WR9k · 2025-03-10

**Overall Recommendation:** 3

**Summary:**

This paper is a theoretical study on positional transformers, where the attention is determined solely by positional encoding regardless of the content. The paper includes a representation theory that position transformers can simulate MPC, followed by a generalization bound. Empirical study is conducted on several synthetic tasks, showing that positional transformers can indeed complete the tasks, though the model needs more layer and data to achieve so.

**Claims And Evidence:**

yes

**Essential References Not Discussed:**

no

**Experimental Designs Or Analyses:**

yes

**Methods And Evaluation Criteria:**

yes

**Other Comments Or Suggestions:**

If the authors can make the novelty and motivation more convincing, I can raise my score.

**Other Strengths And Weaknesses:**

Strength:
- The MPC representation theory is interesting, which can shed light on how transformers solve problems;
- It seems that a better generalization bound is derived thanks to the positional restriction.
- Positional transformers can be indeed of interest for some tasks, as they are at least as powerful as CNNs.

Weaknesses:
- The novelty is not clear. I think the MPC representation result is also proven for standard transformers.
- The biggest concern I have is, I are not sure *why* we need to study positional transformers. I think the motivation is might the simplification of attention make the theoretical analysis easier and we can obtain more theoretical insights to positional transformers, and more importantly, to standard transformers. However, I don’t quite see the gain of the positional constraints. It seems that (Sanford et al., 2024) already have the MPC simulation result, and the positional version is worse (requires more layers). Why do we want to study  positional transformers then? To show that the content is indeed important, and you will suffer significant representation power loss with only position? Similarly, I do not understand what insight we gain by studying positional transformers rather than standard transformers from the generalization result.

**Questions For Authors:**

see weaknesses

Update:
=

the authors have addressed my concerns on the motivations of studying PT, and the OOD generalization result is interesting. I raised my score.

**Relation To Broader Scientific Literature:**

The contribution of the paper seems to be extending the theoretical analysis of standard transformer to positional transformers.

**Theoretical Claims:**

I checked the soundness of the theorem, but not the proof.

---

> ### Author Rebuttal · Authors · 2025-04-01
>
> We thank the reviewer for allowing us to clarify the motivation of our paper. We will address their questions individually and provide general remarks below.
>
> *High-level motivation:* Generally, there is growing interest in the relationship between neural networks and computational models [1,2,3,4]. Computational models execute algorithms on data. For example, Figure 7 in our paper shows the communication trace of an algorithm for computing a parallel cumulative sum.
>
> *Motivation for positional constraints:* As in Figure 7 and many other parallel algorithms, data does not influence communication among processors within computational models. Instead, communication relies solely on processor IDs (or positions). For standard Transformers (ST), this implies that the computation of attention weights can be independent of the input values for certain algorithms, and that is how we arrive at positional Transformers (PT), which is our object of study. In c., we discuss the trade-offs of these constraints.
>
> We now address specific concerns raised by the reviewer.
>
> > a. The novelty is not clear. I think the MPC representation result is also proven for standard transformers.
>
> PT poses an algorithmic computation-inspired departure from the original attention mechanism, and the theoretical results of one do not directly translate to the other. This fact alone justifies the need for an MPC simulation result. The proof techniques we used are significantly different from the ones used for the ST and cannot be directly borrowed from [1], which we further discuss in response b.
>
> > b. [...] the simplification of attention make the theoretical analysis easier and we can obtain more theoretical insights to PT, and more importantly, to ST.
>
> The reviewer seems to suggest that the goal of introducing PT is to simplify the theoretical analysis of ST. We would like to emphasize that this is not the case. We argue that the expressivity analysis of PT is even more challenging. Simulating MPC communication with ST is facilitated by self-attention’s ability to depend on the input values. In contrast, PTs rely on fixed communication via positional encodings, making simulation more challenging. Consequently, MPC simulation with PT must show dynamic communication over a static network, which we accomplish by using Benes networks, a novel strategy unseen in previous related works.
>
> > c. However, I don’t quite see the gain of the positional constraints. It seems [...] PT is worse (requires more layers).
>
> The comparison between PT and ST is more nuanced than suggested. PT offers a different learnability trade-off, illustrated in theory and practice. While PT requires more layers to simulate MPC, this is a worst-case analysis that accounts for all types of algorithms MPC can execute. However, as shown in Section 6, PT has an advantage in learnability: it removes a dependency on the norms of the query and key parameters when all other factors are fixed. Moreover, our empirical results show the competitiveness of PT: Figure 2 showcases some algorithmic tasks where PT outperforms ST in-distribution (ID) with the same number of layers. Furthermore, PT consistently outperforms ST in OOD scenarios for those tasks (Figure 4). As discussed in Section 7.2, we hypothesize that PT exhibits an OOD advantage in data-independent algorithmic tasks due to its architectural alignment with the target function. Our experiments are designed to test this hypothesis.
>
> > d. Why [...] study PT then? To show [...] significant representation power loss with only position?
>
> The conclusion that PT lacks representation power is incorrect. Our theory demonstrates that with logarithmically more layers, similar to ST, PT can simulate any computation within the MPC protocol. Empirically, we observe that for many algorithmic tasks, PT has comparable ID performance and significantly better OOD performance than ST, with the same number of layers. We refer the reviewer to our response in c. for more context.
>
> > e. Similarly, I do not understand what insight we gain by studying PT rather than standard transformers from the generalization result.
>
> This is highlighted in Sections 1 and 6 of the paper. The interesting takeaway is the tradeoff between a better generalization bound (independent of parameter norms) and the (potential) need for logarithmically more layers needed to maintain expressivity. We refer the reviewer to response c. for more details.
>
> ---
> Once again, we thank the reviewer and welcome any further comments. We hope the clarifications are sufficient for them to consider raising their score.
>
> References:
>
> [1] Sanford et al. Transformers, parallel computation, and logarithmic depth. ICML 2024
>
> [2] Pérez et al. Attention is Turing-Complete. JMLR 2021
>
> [3] Loukas, A. What graph neural networks cannot learn: depth vs width. ICLR 2020
>
> [4] Wei et al. Statistically meaningful approximation: a case study on approximating Turing machines with transformers. NeurIPS 2022

---

### Official Review · Reviewer_USLR · 2025-03-13

**Overall Recommendation:** 3

**Summary:**

This paper introduces and analyzes positional attention in Transformers where attention weights depend exclusively on positional encodings rather than on input data. The authors prove that Transformer with positional attention with logarithmic depth has the same expressive power as MPC, and demonstrate that positional Transformers has a trade-off in-distribution and has certain benefit in OOD tasks when the task relies on positional information.

### Update after rebuttal

The response addresses most concerns so I will keep my positive rating.

**Claims And Evidence:**

Yes the claims made in the submission supported by clear and convincing evidence.

**Essential References Not Discussed:**

Not to my knowledge.

**Experimental Designs Or Analyses:**

I checked the experimental setups for both in-distribution regime and out-of-distribution regime. I found the out-of-distribution regime rather limited since in Transformers people also consider longer lengths as OOD.

**Methods And Evaluation Criteria:**

Yes the proposed methods and/or evaluation criteria (e.g., benchmark datasets) make sense for the problem or application at hand.

**Other Comments Or Suggestions:**

For Weaknesses 1&2, the paper would benefit from further investigation in such directions.

**Other Strengths And Weaknesses:**

Strengths: see contributions.

Weaknesses:
1. The proposed method seems to be heavily dependent on the positional encodings used in the Transformer model and the experiment only uses one type of the positional encodings. How to learn useful positional encodings and how the positional Transformer will behave using other positional encodings remain uncertain.
2. As mentioned in Experimental Designs Or Analyses, length generalization seems more natual to Transformers as OOD and the paper does not investigate it. It also poses extra difficulity since the depth of the model needs to be updated when the length of the input changes. How to solve this problem remains unknown from the paper.

**Questions For Authors:**

1. The benifit of the OOD case in the paper seems to be highly related with the nature of the tested task. Is there any case where the standard Transformer behave better than positional Transformer in OOD? Such results would also help to understand the particular inductive bias that both models favor.

**Relation To Broader Scientific Literature:**

1. The paper proposed to use positional only data to get K, Q matrices in attention as a noval architecture design as opposed to traditional attention mechanism.
2. The paper shows the expressiveness of the positional Transformer respect to MPC, which is new to my best knowledge.
3. The paper honestly presents the trade-offs between positional and standard Transformers, showing where each excels and explaining the underlying reasons.

**Theoretical Claims:**

I briefly checked the proofs in Appendix B.

---

> ### Author Rebuttal · Authors · 2025-04-01
>
> We sincerely thank the reviewer for taking the time to read and review our manuscript. In what follows, we provide a comprehensive response to the weaknesses highlighted by the reviewer.
>
> > The proposed method seems to be heavily dependent on the positional encodings used in the Transformer model and the experiment only uses one type of the positional encodings. How to learn useful positional encodings and how the positional Transformer will behave using other positional encodings remain uncertain
>
> As noted in lines 289-293 (second column) of the main paper, we have experimented with various configurations of positional encodings, including sinusoidal, binary, and rotary (RoPE) positional encodings. In all cases, we observe the effects reported in the main paper. We refer the reviewer to Appendix D 1.6 for more details on those cases.
>
> > As mentioned in Experimental Designs Or Analyses, length generalization seems more natual to Transformers as OOD and the paper does not investigate it. It also poses extra difficulity since the depth of the model needs to be updated when the length of the input changes. How to solve this problem remains unknown from the paper.
>
> *Depth of the model:* We’d like to emphasize that a fixed-depth Positional Transformer can indeed work with inputs of different lengths (up to a predetermined length). This is demonstrated by our experiments, where a PT model is trained to perform algorithmic tasks on multiple input lengths, a setting reminiscent of the notion of context length in Transformers. Appendix D.1.4 presents a single model capable of processing different input lengths up to a fixed upper bound, demonstrating the effectiveness of our approach in processing variable-length inputs while achieving good generalization on unseen input values. We would also like to point out that our in-distribution generalization bound does not require fixing the number of layers; it assumes a fixed upper bound on the length instead.
>
> *Types of OOD generalization:* Algorithmic execution using neural networks involves multiple kinds of OOD generalization. Length generalization refers to executing the same algorithm on input lengths unseen during training, while value generalization applies the algorithm to new input values not encountered during training. Both are crucial, as an ideal model should perform well on unseen lengths and values. In this work, our primary focus is on value generalization, as it ensures that, for a fixed upper bound on the length, the model has learned an algorithmic execution applicable to any combination of numbers.
>
> > The benefit of the OOD case in the paper seems to be highly related with the nature of the tested task. Is there any case where the standard Transformer behave better than positional Transformer in OOD? Such results would also help to understand the particular inductive bias that both models favor.
>
> We thank the reviewer for the insightful question. As mentioned in Section 7.2, we hypothesize that the alignment between the architecture and the underlying problem promotes OOD generalization. While Figure 4 shows instances where Positional Transformers outperform standard Transformers, we also present a case where the opposite occurs. In the k-hop induction heads task, which involves dynamic communication, our hypothesis of a fixed communication graph is violated. As a result, our architecture underperforms compared to standard Transformers, which converge more quickly and with fewer layers, an advantage we attribute to algorithmic alignment. Here, the dynamic nature of the problem aligns better with the flexibility of self-attention compared to the rigidity of positional attention.
>
> ---
>
> Once again, we thank the reviewer and welcome any further comments. We hope the clarifications are sufficient for them to consider raising their score.

---

> > ### Comment · Reviewer_USLR · 2025-04-03
> >
> > Thank you for the response. There is some minor confusion though.
> > > "In this work, our primary focus is on value generalization, as it ensures that, for a fixed upper bound on the length, the model has learned an algorithmic execution applicable to any combination of numbers."
> >
> > I think the value generalization decribed in the paper (L380 left) is extending the range of the input, not extending the combination. Could you clarify this?

---

> > > ### Author Response · Authors · 2025-04-03
> > >
> > > The reviewer’s intuition is correct. Here, “combination” just refers to any selection of numbers from the test set in L380 left. In other words, the input list may include numbers beyond the training scale.
> > >
> > > We appreciate the careful consideration and, if our response has addressed all concerns, would be grateful for a positive reassessment.

---

### Official Review · Reviewer_U2No · 2025-03-14

**Overall Recommendation:** 4

**Summary:**

This paper presents transformer with positional attention (PT), a mechanism that implements data-free query and key inputs for computing attention scores. From the empirical side, this mechanism aims to emulate the massively parallel computation (MPC) model, which the authors show expressivity and learnability results. Experimental results are presented that shows on select algorithmic tasks positional attention can generalize better than standard attention.

**Claims And Evidence:**

Yes. I believe the claims are well articulated. The authors clearly established their aim in establishing equivalence between PT and MPC in Section 5, and discussed potential limitations (in terms of quadratic complexity) from ln 207 onwards. The generalization bound given in Section 6 appears to be correct.

**Essential References Not Discussed:**

I'm not aware of any undiscussed literature.

**Experimental Designs Or Analyses:**

The experiments are designed to assess the generalization performance of PT on algorithmic tasks. The generalization and OOD experiments are done properly.

One of my core concern for this comparison is that the authors directly supplement positional encoding in the form of QK for PT, but it is only fed into the standard transformer at the input layer. One could imagine that for deeper networks, positional encoding may vanish for standard transformers but not the case for PT (though of course, one could argue that this is precisely PT's benefit). I do believe that additional experiments on real datasets, even training small language models could further strengthen this paper.

**Methods And Evaluation Criteria:**

I believe the experiments are designed reasonably well to establish that PT can achieve comparable (in-distribution) or better (out-of-distribution) performance on proposed algorithmic tasks, which are commonly used for evaluating theoretical works that establish transformers as computation models. While I understand that real-world applications is not a focal point of this paper, and I certainly does not lower my review because of it, it'd be great to discuss its limitation (beyond those discussed in Section 8) in modeling natural language data, or discuss how this type of algorithmic transformer could be useful in real world settings.

**Other Comments Or Suggestions:**

NA

**Other Strengths And Weaknesses:**

I believe this is a solid paper with a valid hypothesis, correct theoretical analysis, and reasonable experimental design. OOD (or length) generalization is a challenging task in this line of literature and it's nice to see some positive results from PT.

**Questions For Authors:**

See comments about discussions on natural language datasets and broader impacts in the **Methods And Evaluation Criteria** section.

**Relation To Broader Scientific Literature:**

This work falls into a growing body of work that assess the algorithmic capability of transformers [1][2], investigations of optimal encodings of positional information [3], graph neural nets for algorithmic execution [4].


[1] Neural Networks and the Chomsky Hierarchy, https://arxiv.org/abs/2207.02098

[2] Training Neural Networks as Recognizers of Formal Languages, https://arxiv.org/abs/2411.07107

[3] Round and Round We Go! What makes Rotary Positional Encodings useful?, https://arxiv.org/abs/2410.06205

[4] Everything is Connected: Graph Neural Networks, https://arxiv.org/abs/2301.08210

**Theoretical Claims:**

Due to time constraint, I was only able to check the expressivity results (Appendix B), which seems correct. I was not able to check generalization result in Appendix C carefully but it seems to be a an application of Theorem 5 of Bartlett et. al. (2003), coupled with Lipschitz results of the PT model. And I'm conservatively optimistic about the results.

---

> ### Author Rebuttal · Authors · 2025-04-01
>
> We thank the reviewer for their comments and appreciate their recognition of the soundness of our paper. We appreciate the connections made to other related works, which we will incorporate into the main paper. Below, we address their insightful comments, specifically those written in Methods And Evaluation Criteria:
>
> > [...] it'd be great to discuss its limitation (beyond those discussed in Section 8) in modeling natural language data discuss how this type of algorithmic transformer could be useful in real world settings.
>
> We appreciate the reviewer’s suggestion. Along this line, one relevant computational task we considered in our work is the k-hop inductive heads task discussed in Section 7, which is inspired by the in-context learning abilities of language models and captures some of the pattern matching and higher-order dependencies found in NLP tasks [1]. While we agree that experiments in Natural Language are an interesting direction for studying the effectiveness of positional Transformers (PT), the manuscript is already quite extensive. To ensure a thorough analysis, we believe it is best to reserve these explorations for future work, which we intend to pursue as a follow-up. Regarding its usefulness in real-world settings, one possible direction is applying PT to unstructured tabular data in the form of text, where prompts require algorithmic computation to be answered. In Section 7.3, we present experiments in a simplified setting where PT proves useful and outperforms standard Transformers (ST). Building on this, we are actively preparing further empirical studies in this direction as part of our future work.
>
> > [...] (though of course, one could argue that this is precisely PT's benefit). I do believe that additional experiments on real datasets, even training small language models could further strengthen this paper.
>
> We appreciate the reviewer’s insight and agree that additional real-world experiments could further illustrate PT’s potential. Interestingly, our ablation study in Appendix D 1.7 already captures this intuition but also reveals more: even when positional encodings are incorporated at every layer of ST, its performance remains suboptimal. This suggests that PT’s advantage is not merely due to its handling of positional information but rather to the presence of two distinct computational streams—one for positions and one for input values—which we hypothesize is a key factor behind its effectiveness. Furthermore, the new theoretical learnability trade-off for the in-distribution setting in Section 6 comes from this decoupling property (see comments in line 176).
>
> Again, we thank the reviewer for carefully reading our manuscript and for their thoughtful feedback. We welcome any further questions.
>
> References
>
> [1] Olsson et al. In-context learning and induction heads. 2022

---

> > ### Comment · Reviewer_U2No · 2025-04-01
> >
> > I'd like to thank the authors for their detailed response, which has addressed my concerns. I'd like to keep the rating and recommend acceptance for this paper.

---

### Official Review · Reviewer_DrFP · 2025-03-14

**Overall Recommendation:** 4

**Summary:**

This paper proposes the Positional Transformer architecture for learning algorithmic problems over abstract data structures. In Positional Transformers, the attention maps are computed merely based on the positional embeddings and therefore the learned interaction patterns between different positions in the input sequence are independent of the values at those positions. This property makes Positional Transformers a better candidate compared to regular Transformers for certain class of algorithmic problems, especially in terms of OOD generalization as empirically demonstrated in the paper. The paper has also provided an elegant theoretical analysis proving the expressivity and learnability of Positional Transformers, most notably the generalization bound for Positional Transformers.

**Claims And Evidence:**

The paper has done an excellent job of precisely stating the claims as well as providing both theoretical and empirical evidence for those claims.

**Essential References Not Discussed:**

There's a quite extensive literature around GNNs for algorithmic problems and in particular NP problems (e.g. SAT solving, CSPs, etc.) that is missing from this paper.

**Experimental Designs Or Analyses:**

The paper does a good job of setting up sufficient experiments to evaluate both the merits and the shortcomings of Positional transformers in solving algorithmic problems. I especially liked the fact that different classes of problems were used to pinpoint the scenarios in which Positional Transformers are indeed useful.

**Methods And Evaluation Criteria:**

Regarding evaluating both expressivity and learnability, the authors have provided relevant settings and metrics to empirically evaluate the theoretical results.

**Other Comments Or Suggestions:**

N/A

**Other Strengths And Weaknesses:**

Strengths:

This paper is very well-writte and well-motivated.
The claims are carefully stated and adequately supported.
The authors have done a great job with their empirical study covering both the strength and weaknesses areas for Positional Transformers.

Weaknesses:

In terms of practicality of the proposed models, it'd be interesting to have experimental results on more realistic datasets with much larger problem sizes.

**Questions For Authors:**

- For the OOD case, do you think if we keep the Q and K projection matrices frozen in Positional Transformers and merely fine-tune the V projection matrices, we would be able to achieve the same level of generalization as the original in-distribution case? I believe that'd be an interesting ablation study which can further shows the importance of Positional Attention for certain algorithmic problems.

**Relation To Broader Scientific Literature:**

I believe the findings of this paper can be further relevant useful for other areas of Deep Learning such as neuro-symbolic methods. In particular, it'd be interesting to see how symbolic computation can be further injected as inductive biases into the positional transformers and more generally purely neural models.

**Theoretical Claims:**

I did not check the proofs.

---

> ### Author Rebuttal · Authors · 2025-04-01
>
> We sincerely appreciate the reviewer’s time in evaluating our manuscript. We are grateful for their recognition of the novelty and significance of our work. Below, we address their insightful comments.
>
> > There’s a quite extensive literature around GNNs for algorithmic problems and in particular NP problems (e.g. SAT solving, CSPs, etc.) that is missing from this paper.
>
> We thank the reviewer for pointing out this gap in our literature review section. We have identified key references (original works and highly-cited) like [1,2,3,4,5,6,7] as well as a comprehensive survey [8], which we will include in the camera-ready version. If the reviewer believes specific works need to be mentioned explicitly, we would be happy to add them.
>
> > For the OOD case, do you think if we keep the Q and K projection matrices frozen in Positional Transformers and merely fine-tune the V projection matrices, we would be able to achieve the same level of generalization as the original in-distribution case?
>
> This is a great question! The answer is yes: fine-tuning only the V projection matrix in the OOD case allows the positional Transformer to achieve the same level of generalization as in the in-distribution case. On the other hand, the same does not hold for the standard Transformer. We appreciate the reviewer’s suggestion of this scenario, which closely aligns with the typical approach to “fine-tune a pre-trained foundation model with unseen data” when dealing with new and potentially OOD data in practice.
>
> In the tables below, we consider three simple algorithmic tasks and report the median test error (MSE) for both positional and standard transformer models when
> - they are trained and tested on in-distribution data (in-dist),
> - they are trained on in-distribution data but tested on numbers 10x larger than the original in-distribution data (10x OOD),
> - they are trained on in-distribution data, fine-tuned V projection matrices on numbers 10x larger than the original in-distribution data, and then tested on numbers 10x larger than the original in-distribution data (10x OOD with fine-tuning).
>
> We find that fine-tuning only the V projection matrices brings the accuracy of the positional Transformer to the same level as the in-distribution case. However, for the standard Transformer, even after fine-tuning the V projection matrices, the accuracy on 10x test data is still very high compared to that of the positional Transformer. The comparison highlights another potential advantage of the positional Transformer for executing algorithmic tasks: one might efficiently fine-tune a pre-trained positional Transformer on OOD data and achieve good performance.
>
> | Task  | Positional (in-dist) | Positional (10x OOD) | Positional (10x OOD with fine-tuning) |
> | :--- | ---: | ---: | ---: |
> | sort | 1.20e-04 | 7.37e-04 | 1.23e-04 |
> | min | 1.03e-05 | 3.92e-04 | 2.19e-05 |
> | sum | 6.07e-06 | 1.31e-03 | 1.32e-05 |
>
> | Task | Standard (in-dist) | Standard (10x OOD) | Standard (10x OOD with fine-tuning) |
> | :--- | ---: | ---: | ---: |
> | sort | 9.05e-04 | 5.18e-01 | 6.90e-02 |
> | min | 1.74e-06 | 1.39e-01 | 4.00e-02 |
> | sum | 5.20e-06 | 1.09e+00 | 1.23e-01 |
>
> *We keep the same empirical setting where we train both models for 2000 epochs. For fine-tuning, we adopt the common practice and only train for 20 epochs. We normalize the MSE by the scale factor so that the table results indicate relative accuracy.
>
> ---
>
> Again, we thank the reviewer for carefully reading our manuscript and for their thoughtful feedback. We welcome any further questions.
>
> References:
>
> [1] Vinyals et al. Pointer Networks, NeurIPS 2015
>
> [2] Prates et al. Learning to Solve NP-Complete Problems: A Graph Neural Network for Decision TSP, AAAI 2019
>
> [3] Joshi et al. An Efficient Graph Convolutional Network Technique for the Travelling Salesman Problem, 2019
>
> [4] Bai et al. Learning-Based Efficient Graph Similarity Computation via Multi-Scale Convolutional Set Matching, AAAI 2020
>
> [5] Selsam et al. Guiding High-Performance SAT Solvers with Unsat-Core Predictions, SAT 2019
>
> [6] Karalias and Loukas Erdos Goes Neural: an Unsupervised Learning Framework for Combinatorial Optimization on Graphs, NeurIPS 2020
>
> [7] Gasse et al. Exact Combinatorial Optimization with Graph Convolutional Neural Networks, NeurIPS 2019
>
> [8] Cappart et al. Combinatorial Optimization and Reasoning with Graph Neural Networks, JMLR

---

### Decision · Program_Chairs · 2025-05-01

**Decision:**

Accept (poster)

**Comment:**

This paper introduces Transformer with Positional Attention (PT), a mechanism that computes attention scores using only positional information rather than content. The authors establish the equivalence between PT and the MapReduce-like distributed computing model (aka MPC), provide generalization bounds, and demonstrate through experiments that PTs can generalize better than standard Transformers on select algorithmic tasks, particularly in out-of-distribution scenarios. Even though the near-equivalence between MapReduce and TF has been widely known in the theory community, I believe it is a solid contribution to make this statement more rigorous by proposing a variant of attention.

The reviewers unanimously recommend acceptance based on several strengths. First, the paper presents clear theoretical contributions with expressivity and learnability results. Second, the experimental design effectively demonstrates both the merits and limitations of PTs across different algorithmic problems. Third, the paper presents the trade-offs between positional and standard Transformers, showing where each excels. Fourth, the OOD generalization results seem valuable, as this is a challenging area in transformer research. Overall, the paper is well-written and well-motivated with claims carefully stated and supported.